# Dataset Distillation Efficiently Encodes Low-Dimensional Representations from Gradient-Based Learning of Non-Linear Tasks

**Yuri Kinoshita** [1 2]  **Naoki Nishikawa** [1 3]  **Taro Toyoizumi** [1 2]

## Abstract

Dataset distillation, a training-aware data compression technique, has recently attracted increasing attention as an effective tool for mitigating costs of optimization and data storage. However, progress remains largely empirical. Mechanisms underlying the extraction of task-relevant information from the training process and the efficient encoding of such information into synthetic data points remain elusive. In this paper, we theoretically analyze practical algorithms of dataset distillation applied to the gradient-based training of two-layer neural networks with width $L$. By focusing on a non-linear task structure called multi-index model, we prove that the low-dimensional structure of the problem is efficiently encoded into the resulting distilled data. This dataset reproduces a model with high generalization ability for a required memory complexity of $\tilde{\Theta}(r^2 d + L)$, where $d$ and $r$ are the input and intrinsic dimensions of the task. To the best of our knowledge, this is one of the first theoretical works that include a specific task structure, leverage its intrinsic dimensionality to quantify the compression rate and study dataset distillation implemented solely via gradient-based algorithms.

## 1. Introduction

### 1.1. Background

Over the past few years, deep learning has advanced in tandem with a training paradigm that benefits substantially from systematic increases in data scale. While this approach has yielded unprecedented results across a wide range of domains, it entails fundamental limitations, notably in terms of the costs of training, data storage and its transmission.

*Dataset distillation* (DD), also known as dataset condensation, addresses these challenges by constructing a small set of *trained synthetic* data points that distill essential information from the learning scenario of a given problem so that training with these resulting instances reproduces a high generalization score on the task (Wang et al., 2018; Zhao et al., 2021; Cazenavette et al., 2022; Wang et al., 2022). The effectiveness of DD in reducing the amount of data required to obtain a high-performing model has been reportedly observed and is nowadays attracting increasing attention in various fields that span modalities such as images (Wang et al., 2018), text (Sucholutsky & Schonlau, 2021), medical data (Li et al., 2020), time series (Ding et al., 2024) and electrophysiological signals (Guo et al., 2025). Beyond this efficiency, DD offers a set of condensed training summaries, whose replay can be applied to promote transfer learning (Lee et al., 2024), neural architecture search (Zhao et al., 2021), continual learning (Liu et al., 2020; Masarczyk & Tautkute, 2020; Kong et al., 2024), federated learning (Zhou et al., 2020), data privacy (Dong et al., 2022) and model interpretability (Cazenavette et al., 2025).

In parallel to ongoing empirical development, theoretical work has been led to explain why DD can compress a substantial training effort, arising from the complex interaction between task structure, model architecture, and optimization dynamics, into a few iterations over a small number of synthetic data points. On the one hand, some studies have primarily considered the number of data points sufficient to reconstruct the exact linear ridge regression (LRR) or kernel ridge regression (KRR) solution of the training data (Izzo & Zou, 2023; Maalouf et al., 2023; Chen et al., 2024b). On the other hand, a scaling law for the required number of distilled samples was recently proved (Luo & Xu, 2025). Despite these advances, existing analyses either focus on essentially linear models or do not explicitly characterize how the intrinsic task structure mediates distillation. Especially, leveraging low-dimensionality of the problem is known to be central to the efficiency and adaptivity of gradient-based algorithms (Damian et al., 2022). Similar considerations appear to apply in DD, where more challeng-

---

[1]The University of Tokyo, Tokyo, Japan [2]Laboratory for Neural Computation and Adaptation, RIKEN Center for Brain Science, Wako, Japan [3]RIKEN Center for Advanced Intelligence Project, Wako, Japan. Correspondence to: Yuri Kinoshita <yuri-kinoshita111@g.ecc.u-tokyo.ac.jp>.

*Proceedings of the 43rd International Conference on Machine Learning*, Seoul, South Korea. PMLR 306, 2026. Copyright 2026 by the author(s).

ing tasks tend to require larger distilled sets (Zhao et al., 2021). In short, the rigorous mechanisms through which DD operates in complex practical learning regimes, encompassing task structure, non-linear models, and gradient-based optimization dynamics, remain underexplored.

Therefore, in this paper, we precisely theoretically study DD in such a framework, focusing on how task structure can be leveraged to achieve low memory complexity of distilled data that realizes high generalization ability when used at training. Particularly, we analyze two state-of-the-art DD algorithms, performance matching (Wang et al., 2018) and gradient matching (Zhao et al., 2021), applied to the training of two-layer ReLU neural networks under a nontrivial non-linear task endowed with a latent structure called *multi-index models*. All optimization procedures follow finite time gradient-based algorithms.

## 1.2. Contributions

Our major contributions can be summarized as follows:

- To the best of our knowledge, this is one of the first theoretical works to study DD (gradient and performance matching) implemented solely via gradient-based algorithms and to include a specific task structure with low intrinsic dimensionality.

- We prove that DD applied to two-layer ReLU neural networks with width $L$ learning a class of non-linear functions called *multi-index models* efficiently encode latent representations into distilled data.

- We show that this dataset reproduces a model with high generalization ability for a memory complexity of $\tilde{\Theta}(r^2 d + L)$, where $d$ and $r$ are the input and intrinsic dimensions of the task. See Table 1 for comparison.

- Theoretical results and its application to transfer learning are discussed and illustrated with experiments.

## 1.3. Related Works

This work connects perspectives and insights from three different lines of work.

Empirical investigations of DD have been applied to both regression (Ding et al., 2024; Mahowald et al., 2025) and classification tasks (Wang et al., 2018), and depending on which aspect of training information the distillation algorithm prioritizes, existing methods can be grouped into several categories (Yu et al., 2024). Among them, *performance matching* (PM) directly focuses on minimizing the training loss of a model trained on the distilled data (Wang et al., 2018; Sucholutsky & Schonlau, 2021; Nguyen et al., 2021a,b; Zhou et al., 2022), while *gradient matching* (GM) learns synthetic instances so that gradients of the loss mimic those induced

by real data (Zhao et al., 2021; Liu et al., 2022; Jiang et al., 2023). Other methods such as distribution matching (Zhao & Bilen, 2023), trajectory matching (Cazenavette et al., 2022) and diffusion-based distillation (Gu et al., 2024) are out of the scope of this work as PM and GM serve as the core building blocks of some of them and provide a regime where one can mathematically trace the most fundamental interaction between task structure, non-linear models and optimization dynamics. Such algorithms constitute the ingredients of DD and can be incorporated into a wide variety of strategies. They have been developed to improve scalability (Cui et al., 2023; Chen et al., 2024a), incorporate richer information (Son et al., 2025; Zhao & Bilen, 2021; Deng & Russakovsky, 2022; Liu et al., 2022), and be tailored to pre-trained models (Cazenavette et al., 2025). In this paper, we follow the procedure of Chen et al. (2024a) called *progressive dataset distillation*, which learns a distilled dataset for each partitioned training phase.

Current theory of DD is mainly formulated as the number of synthetic data points required to infer the exact final parameter optimized for the training data. Izzo & Zou (2023) showed that this number equals the input dimension for LRR, and the size of the original training set for KRR with a Gaussian kernel. For shift-invariant kernels (Maalouf et al., 2023) or general kernels (Chen et al., 2024b), this amount becomes the dimension of the kernel. Under LRR and KRR with surjective kernels, Chen et al. (2024b) refined the bounds to one distilled point per class. Izzo & Zou (2023) showed that one data point suffices for a specific type of model called generalized linear models. As for recent work, Luo & Xu (2025) characterize the generalization error of models trained with distilled data over a set of algorithm and initialization configurations. Nevertheless, these prior theoretical works do not take into account any kind of task structure. Gradient-based algorithms of DD such as GM for non-linear neural networks are not considered either. We provide a framework that both fills this gap and reveals the high compression rate of DD.

Our theory builds on analytical studies of feature learning in neural networks via gradient descent. Their optimization dynamics are rigorously inspected, yielding lower bounds on the sample complexity of training data required for low generalization loss of non-linear tasks with low-dimensional structures, such as single-index (Dhifallah & Lu, 2020; Gerace et al., 2020; Ba et al., 2022; Oko et al., 2024b; Nishikawa et al., 2025), multi-index (Damian et al., 2022; Abbe et al., 2022; Bietti et al., 2023), and additive models (Oko et al., 2024a; Ren et al., 2025). Typically, these results provide theoretical support for the effective feature learning of neural networks observed in practice beyond neural tangent kernel (NTK) regimes. This framework is thus suited to elucidate the intricate mechanism of DD beyond the NTK and KRR settings of previous works on this topic. We will

*Table 1.* Comparison of our contributions with prior work. They all treat regression problems of the form $\{x_n, f^*(x_n) + \epsilon_n\}$ with noise $\epsilon_n$. We cite only the most relevant results for KRR with non-linear kernels. $N$ is the training data size, $d$ the input dimension, $r$ the intrinsic dimension of multi-index model, $L$ the width of two-layer neural networks, $q$ the feature dimension of the kernels which equals to $L$ if the model is a two-layer neural network.

| | $f^*$ form | Optimizations | Trained Model | Memory Cost |
|---|---|---|---|---|
| Izzo & Zou (2023) | Unspecified | Exact | Linear (Gaussian kernel) | $\Theta(Nd)$ |
| Maalouf et al. (2023) | Unspecified | Exact | Linear (shift invariant kernels) | $\Theta(qd)$ |
| Chen et al. (2024b) | Unspecified | Exact | Linear (any kernel) | $\Theta(qd)$ |
| **Ours** | **Multi-index** | **Gradient-based** | **Non-Linear** | $\tilde{\Theta}(r^2 d + L)$ |

indeed prove that feature learning happens in DD as well, leading to a compact memory complexity that leverages the low-dimensionality of the problem.

**Organization** In Section 2, we will explain the basic formulation of DD and its algorithms. In Section 3, we will dive into a detailed clarification of our problem setting. Section 4 will be devoted to the description of our theoretical analysis. Section 5 will illustrate our theoretical results, and their discussion will be provided in Section 6.

**Notation** Throughout this paper, the Euclidean norm of a vector $x$ is defined as $\|x\|$, and the inner product as $\langle \cdot, \cdot \rangle$. For matrices, $\|A\|$ denotes the $l_2$ operator norm $A$ and $\|A\|_F$ its Frobenius norm. $S^{d-1}$ corresponds to the unit sphere in $\mathbb{R}^d$. When we state for two algorithms $\mathcal{A} := \mathcal{A}'$, $\mathcal{A}$ is defined as $\mathcal{A}'$ with the input arguments inherited from $\mathcal{A}$. $[i]$ is the set $\{1, \ldots, i\}$ and $a \vee b = \max\{a, b\}$. $\tilde{O}(\cdot)$ and $\tilde{\Omega}(\cdot)$ represent $O(\cdot)$ and $\Omega(\cdot)$ but with hidden polylogarithmmic terms.

## 2. Preliminaries

In this section, we explain the mathematical background of DD and its algorithms.

### 2.1. General Strategy

Let us consider a usual optimization framework $\mathcal{O}^{Tr}$ of a model $f_\theta$, where $\theta$ is optimized based on a training data $\mathcal{D}^{Tr} = \{(x_n, y_n)\}_{n=1}^N$ and a loss $\mathcal{L}$ so that the resulting $\theta^*$ realizes a low generalization error $\mathbb{E}_{(x,y)\sim\mathcal{P}}[\mathcal{L}(\theta^*, (x, y))]$, where $\mathcal{P}$ is the data distribution. The goal of DD is to create a *synthetic* dataset $\mathcal{D}^S = \{(\tilde{x}_m, \tilde{y}_m)\}_{m=1}^M$ and, occasionally, an alternative optimization framework $\mathcal{O}^S$, so that training $f_\theta$ with $\mathcal{D}^S$ along $\mathcal{O}^S$ returns a parameter $\tilde{\theta}^*$ that achieves low $\mathbb{E}_{(x,y)\sim\mathcal{P}}[\mathcal{L}(\tilde{\theta}^*, (x, y))]$. Ultimately, we expect that the memory complexity of $\mathcal{D}^S$ is smaller than that of $\mathcal{D}^{Tr}$ and, preferably, than the storage cost of the whole model; otherwise, saving $f_\theta$ may be sufficient in some settings. DD thus compares two training paradigms, *teacher training* $\mathcal{O}^{Tr}$ tuned for the original training data and *student training* $\mathcal{O}^S$ that employs instead the distilled data. Then, it *distills* information from this comparison so

that *retraining* the model on distilled data can reproduce the performance or optimization dynamics of the teacher. These three phases are summarized in Algorithm 1. Here, $\mathcal{A}$ and $\mathcal{M}$ are algorithms, $\xi$ is the number of iteration, $\eta$ the step size, and $\lambda$ the $L_2$ regularization coefficient. We opt for the strategy of *progressive DD* proposed by Chen et al. (2024a), which separates training into multiple phases and applies DD within each of them. This allows DD to cover and distill the whole training procedure and enables us to precisely characterize what information is distilled at each time step. Although Chen et al. (2024a) reuse distilled data from previous intervals in subsequent phases for retraining, in this work, we adopt a simplified variant where we retrain the model with each distilled data consecutively.

### 2.2. Distillation Algorithms $\mathcal{M}$

One of the primary interests of research in DD lies in identifying which properties of teacher training the distilled data should encode and in building an objective function according to this. The direct approach to achieve a low generalization error $\mathbb{E}_{(x,y)\sim\mathcal{P}}[\mathcal{L}(\tilde{\theta}^*, (x, y))]$ is to minimize the training loss $\mathcal{L}(\tilde{\theta}^*, D^{Tr})$ so that by definition, training on $\mathcal{D}^S$ captures the information of the whole training (Wang et al., 2018). This method is called *performance matching* (PM). While PM tries to find the best $\mathcal{D}^S$, this requires a complex bi-level optimization (Vicol et al., 2022). A simple one-step version learning $\mathcal{D}^S$ that trains in one step a model with low training loss can be defined based on Wang et al. (2018) as follows:

**Definition 2.1.** Consider a one-step student training $\mathcal{A}^S$ that outputs $\tilde{\theta}^{(1)}(\mathcal{D}^S) = \theta - \eta^S (\nabla_\theta \mathcal{L}(\theta, \mathcal{D}^S) + \lambda^S \theta)$ for a given initial parameter $\theta$, data $\mathcal{D}^S$ and loss $\mathcal{L}(\theta, \mathcal{D}^S)$. The *one-step PM* with input $\tilde{\theta}^{(1)}(\mathcal{D}^S)$ is defined as $\mathcal{M}^P(\mathcal{D}^S, \tilde{\theta}^{(1)}(\mathcal{D}^S), \mathcal{D}^{Tr}, \xi, \eta, \lambda)$ so that $\mathcal{D}_\tau^S = \mathcal{D}_{\tau-1}^S - \eta(\nabla_{\mathcal{D}^S}\mathcal{L}(\tilde{\theta}^{(1)}(\mathcal{D}_{\tau-1}^S), \mathcal{D}^{Tr}) + \lambda \mathcal{D}_{\tau-1}^S)$ for $\tau = 1, \ldots \xi$ with $\mathcal{D}_0^S = \mathcal{D}^S$ and $\mathcal{M}^P$ outputs $\mathcal{D}_\xi^S$.

On the other hand, *gradient matching* (GM) focuses on local representative information, namely, the gradient information, and optimizes $\mathcal{D}^S$ so that the gradient with respect to $\theta$ of the loss evaluated on $\mathcal{D}^S$ matches that induced by

the training data (Zhao et al., 2021). Notably, a one-step GM can be inferred by their implementation and subsequent works (Jiang et al., 2023), defined as follows:

**Definition 2.2.** Consider sets of gradients from $T$ iterations of teacher training $G^{Tr} = \{G_t^{Tr}\}_{t \in [T]}$ and one iteration of student training $G^S(\mathcal{D}^S) = \{G_1^S\}$. Each gradient is divided by layers with index $l \in [L]$ and by random initialization of the training with $j \in [J]$. The *one-step GM* is defined as $\mathcal{M}^G(\mathcal{D}^S, G^{Tr}, G^S(\mathcal{D}^S), \eta, \lambda)$ which provides as outputs $\mathcal{D}_1^S = \mathcal{D}^S - \eta(\nabla_{\mathcal{D}^S} m(G^{Tr}, G^S(\mathcal{D}^S)) + \lambda \mathcal{D}^S)$ where $m(G^{Tr}, G^S) = 1/J \sum_j (1 - 1/L \sum_l \langle \sum_t G_{t,j,l}^{Tr}, G_{1,j,l}^S \rangle)$.

In the original work of Zhao et al. (2021), $m$ was defined as the cosine similarity. Here, we omitted the normalization factor. This is acceptable in our case because we consider only a single update step, which does not risk diverging, and our main concern is the *direction* of $\mathcal{D}^S$ after one update. We now make the following important remark for GM.

*Remark* 2.3. GM is not well-defined for ReLU activation function, as the gradient update leads to second derivatives of the activation function through $\nabla_{\mathcal{D}^S} G_{1,j,l}^S$. Therefore, we need to slightly adjust the student training and propose two approaches. We either replace the ReLU activation function in the student training with a surrogate $h$ where $h''$ is well-defined, and prove a rather strong result that applies to any well-behaved $h$, or, we propose a well-defined one-step GM update for ReLU. The former is treated in the main paper with Assumption 3.8, and the latter in Appendix C. Both lead to the same result qualitatively.

## 3. Problem Setting and Assumptions

In this section, we provide details on the problem setting, including task structure, model and algorithms.

### 3.1. Task Setup

It has been repeatedly reported that, for simple tasks such as MNIST, as little as one distilled image per class can produce strong performance after retraining, whereas on CIFAR-10 even 50 images per class is insufficient (Zhao et al., 2021). More precisely, Pope et al. (2021) estimates the intrinsic dimensions of MNIST, SVHN, and CIFAR10 to be approximately 13, 19, and 26, respectively. In parallel, the dataset distillation (DD) results reported in Zhao et al. (2021) with 500 distilled images decrease from MNIST to SVHN to CIFAR10 (roughly $98\%$, $82\%$, and $54\%$) for convolutional networks (ConvNet). Based on the *manifold hypothesis*, which posits that real-world data distributions concentrate near a low-dimensional manifold (Tenenbaum et al., 2000; Fefferman et al., 2016), we hypothesize that the intrinsic structure of the task plays an important role in DD[1]. We formalize and analyze this phenomenon by considering a

task, called *multi-index model*, that captures the essence of the complex interaction between latent structure and optimization procedure. This is a common setup in analyses of feature learning of neural networks trained under gradient descent (Damian et al., 2022; Abbe et al., 2022; Bietti et al., 2023).

**Assumption 3.1.** Training data $\mathcal{D}^{Tr}$ is given by $N$ i.i.d. points $\{(x_n, y_n)\}_{n=1}^N$ with, $x_n \sim N(0, I_d) \in \mathbb{R}^d$, $y_n = f^*(x_n) + \epsilon_n \in \mathbb{R}$ where $\epsilon_n \sim \{\pm \zeta\}$ with $\zeta > 0$. $f^* : \mathbb{R}^d \to \mathbb{R}$ is a normalized degree $p$ polynomial with $\mathbb{E}_x[f^*(x)^2] = 1$, and there exist a $B = (\beta_1, \ldots, \beta_r) \in \mathbb{R}^{d \times r}$ and a function $\sigma^* : \mathbb{R}^r \to \mathbb{R}$ such that $f^*(x) = \sigma^*(B^\top x) = \sigma^*(\langle \beta_1, x \rangle, \ldots, \langle \beta_r, x \rangle)$. Without loss of generality, we assume $B^\top B = I_r$.

When $r = 1$, we call it *single index model* (Dhifallah & Lu, 2020). We define its principal subspace and orthogonal projection.

**Definition 3.2.** $S^* := \text{span}\{\beta_1, \ldots, \beta_r\}$ is the principal subspace of $f^*$ and $\Pi^*$ the orthogonal projection onto $S^*$.

Understanding how and when DD captures this principal subspace constitutes one of the main focuses of this analysis. We impose the following additional condition on the structure of $f^*$. This guarantees that the gradient information is non-degenerate.

**Assumption 3.3.** $H := \mathbb{E}_x[\nabla_x^2 f^*(x)]$ has rank $r$ and satisfies $\text{span}(H) = S^*$. $H$ is well-conditioned, with maximal eigenvalue $\lambda_{\max}$, minimal non-zero eigenvalue $\lambda_{\min}$ and $\kappa := \lambda_{\max}/\lambda_{\min}$.

### 3.2. Trained Model

The task defined above is learned by a two-layer ReLU neural network $f_\theta$ with width $L$ and activation function $\sigma(x) = \max(0, x)$, i.e., for $a \in \mathbb{R}^L$, $W = (w_1, \ldots, w_L) \in \mathbb{R}^{d \times L}$, $b \in \mathbb{R}^L$, $\theta = (a, W, b)$, $f_\theta(x) := \sum_{i=1}^L a_i \sigma(\langle w_i, x \rangle + b_i)$. For a dataset $\mathcal{D} = \{(x_n, y_n)\}_{n=1}^N$, the empirical loss is defined as $\mathcal{L}(\theta, \mathcal{D}) := \frac{1}{2N} \sum_{n=1}^N (f(x_n) - y_n)^2$ (MSE loss). While this model may be simple, it is more complex than previous analyses of linear layers and embodies the complex interaction of task, model, and optimization we are interested in. We use a symmetric initialization. As mentioned in other work (Damian et al., 2022; Oko et al., 2024b), small random initializations should not change our statements qualitatively.

**Assumption 3.4.** $L$ is even. $f_\theta$ is initialized as $\forall i \in [L/2]$, $a_i \sim \{\pm 1\}$, $a_i = -a_{L-i}$, $w_i \sim N(0, I_d/d)$, $w_i = w_{L-i}$ and $b_i = b_{L-i} = 0$.

Damian et al. (2022) showed that $N \geq \tilde{\Omega}(d^2 \vee d/\epsilon \vee 1/\epsilon^4)$ data points are required for a gradient-based training (See Definition 3.5) to achieve a generalization error below $\epsilon$. The preprocessing of Algorithm 1 is inspired by their work.

---

[1]Please also refer to Table 5 for additional illustration

**Algorithm 1** Dataset Distillation

---

**Input:** Training dataset $\mathcal{D}^{Tr}$, model $f_\theta$, loss function $\mathcal{L}$, reference parameter $\theta^{(0)}$
**Output:** Distilled data $\mathcal{D}^S$
Initialize distilled data $\mathcal{D}_0^S$ randomly, $\alpha = \frac{1}{N}\sum_{n=1}^N y_n$, $\gamma = \frac{1}{N}\sum_{n=1}^n y_n x_n$, preprocess $y_n \leftarrow y_n - \alpha - \langle \gamma, x_n \rangle$
Sample initial states $\{\theta_j^{(0)}\}_{j=0}^J$.
**for** $t = 1$ **to** $T_{\mathrm{DD}}$ **do**
  If we change distilled data (progressive step): keep old $\mathcal{D}^S \leftarrow \mathcal{D}^S \cup \mathcal{D}_{t-1}^S$, prepare new $\mathcal{D}_{t-1}^S$
  *I. Training Phase*
  **for** $j = 0$ **to** $J$ **do**
    Teacher Training $I_j^{Tr} = \{\theta_j^{(t)}, G_j^{(t)}\}$ where
    $I_j^{Tr} \leftarrow \mathcal{A}_t^{Tr}(\theta_j^{(t-1)}, \mathcal{D}^{Tr}, \xi_{t-1}^{Tr}, \eta_{t-1}^{Tr}, \lambda_{t-1}^{Tr})$
    Student Training $I_j^S = \{\tilde{\theta}_j^{(t)}, \tilde{G}_j^{(t)}\}$ where
    $I_j^S \leftarrow \mathcal{A}_t^S(\theta_j^{(t-1)}, \mathcal{D}_{t-1}^S, \xi_{t-1}^S, \eta_{t-1}^S, \lambda_{t-1}^S)$
  **end for**
  *II. Distillation Phase*
  $\mathcal{D}_t^S \leftarrow \mathcal{M}_t(\mathcal{D}_{t-1}^S, \mathcal{D}^{Tr}, \{I_j^{Tr}, I_j^S\}, \xi_{t-1}^D, \eta_{t-1}^D, \lambda_{t-1}^D)$
  *III. Retraining Phase*
  $\theta_j^{(t)} \leftarrow \mathcal{A}_t^R(\theta_j^{(t-1)}, \mathcal{D}_t^S, \xi_{t-1}^R, \eta_{t-1}^R, \lambda_{t-1}^R)$
  for all $j = 0, \ldots, J$.
  Resample $\{\theta_j^{(t)}\}_{j=0}^J$ if necessary.
**end for**
**return** $\mathcal{D}$

---

### 3.3. Optimization Dynamics and Algorithms

Each algorithm of Algorithm 1 can be now presented straightforwardly. Please refer to Algorithm 2 in Appendix D.6 for the complete specification.

**Teacher and Student Training** We opt for the gradient-based training from Damian et al. (2022) (Algorithm 1 in their work) as the teacher training since its mechanism is well studied for multi-index models. This is divided into two phases as shown in Definition 3.5 below. For our progressive type of DD, we prepare two different distilled datasets, one for each phase. Student training follows the same update rule but with only one iteration, which is sufficient for one-step PM and one-step GM. Damian et al. (2022) reinitialize $b$ between the two parts which is also incorporated just before $t = 2$. For each step, we output the final parameter and gradient of the loss with respect to the parameter of each iteration as follows:

**Definition 3.5.** Based on the two phases of Algorithm 1 in Damian et al. (2022), we define the following procedures for $\theta = (a, W, b)$. $\mathcal{A}^{(\mathrm{I})}(\theta, \mathcal{D}, \eta, \lambda) := \{-\eta g, \{g\}\}$, where $g = \nabla_W \mathcal{L}(\theta, D)$, and $\mathcal{A}^{(\mathrm{II})}(\theta, \mathcal{D}, \xi, \eta, \lambda) := \{a^{(\xi)}, \{g_\tau\}_{\tau=1}^\xi\}$, where $a^{(\tau)} = a^{(\tau-1)} - \eta g_\tau$ $(\tau \in [\xi])$, $g_\tau = \nabla_a \mathcal{L}((a^{(\tau-1)}, W, b), D) + \lambda a$, and $a^{(0)} = a$.

**Distillation and Retraining** One-step PM (Definition 2.1 and one-step GM (Definition 2.2) are the two DDs we study. For the first phase $t = 1$, we only consider the former as we do not have access to the final state of the model and cannot apply PM. The retraining algorithm follows the teacher training, except for PM which supposes a one step gradient update by definition. To evaluate DD, we retrain the model from a fixed reference initialization $\theta^{(0)}$, and build DD accordingly (see Assumptions 3.6 and 3.7), which is consistent with existing analyses (Izzo & Zou, 2023).[2] Fixing $\theta^{(0)}$ enables controlled evaluation and facilitates isolating fundamental interactions, an essential step towards understanding the mechanism of DD.

In summary, the whole formulation of each optimization dynamics can be presented as follows.

**Assumption 3.6.** Algorithm 1 applied to our problem setting is defined as follows. $T_{\mathrm{DD}} = 2$, and for each $t$ we create a separate synthetic data, $\mathcal{D}_1^S$ and $\mathcal{D}_2^S$. For $t = 1$, $\mathcal{A}_1^{Tr}$, $\mathcal{A}_1^S$, and $\mathcal{A}_1^R$ are all set to be $\mathcal{A}^{(\mathrm{I})}$ with their respective arguments. As for the distillation, $\mathcal{M}_1 = \mathcal{M}^G$ with the gradient information of $\mathcal{A}_1^{Tr}$ and $\mathcal{A}_1^S$ as inputs. For $t = 2$, $\mathcal{A}_2^{Tr} = \mathcal{A}^{(\mathrm{II})}$ and $\mathcal{A}_2^S = \mathcal{A}^{(\mathrm{II})}$, with their respective hyperparameters and $\xi_2^S = 1$. If we use one-step GM at $t = 2$, $\mathcal{M}_2 = \mathcal{M}^G$ with the gradient information of $\mathcal{A}_2^{Tr}$ and $\mathcal{A}_2^S$ as inputs, and $\mathcal{A}_2^R = \mathcal{A}^{(\mathrm{II})}$, otherwise if we use one-step PM at $t = 2$, $\mathcal{M}_2 = \mathcal{M}^P$ with input $\tilde{\theta}_0^{(2)}$, and $\mathcal{A}_2^R = \mathcal{A}^{(\mathrm{II})}$ with $\xi_2^R = 1$. The hyperparameters of $\mathcal{A}_1^{Tr}$ and $\mathcal{A}_2^{Tr}$ are kept the same as Theorem 1 of Damian et al. (2022). and those of $\mathcal{A}_1^S$ is also the same as the former.

The rest of the undefined hyperparameters will be specified later. Batch initializations are defined as follows.

**Assumption 3.7.** For $t = 1$, $\{\theta_j^{(0)}\}_{j=1}^J$ are sampled following the initialization of Assumption 3.4, with $\theta_0^{(0)} = (a^{(0)}, W^{(0)}, 0)$, where the reference $\theta^{(0)} = (a^{(0)}, W^{(0)}, b^{(0)})$ satisfies $\forall i \in [L]$, $a_i \in \{\pm 1\}$, $a_i = -a_{L-i}$, $w_i = w_{L-i}$ and $b^{(0)} \sim N(0, I_L)$. For $t = 2$, we resample $\{\theta_j^{(1)}\}_{j=1}^J$ by $W_j^{(1)} = W_0^{(1)} = W^{(1)}$, $b_j^{(1)} = b^{(0)}$, $a_j^{(1)} \sim \{\pm 1\}$ for $j \in [J]$, and $a_0^{(1)} = 0$.

Finally, based on Remark 2.3, the student training at $t = 1$ is approximated as follows:

**Assumption 3.8.** ReLU activation function $\sigma$ of the student training at $t = 1$, is replaced by a (surrogate) $C^2$ function $h$ so that $h''(t) > 0$ for all $t \in [-1, 1]$.

The last condition is satisfied by a large variety of continuous functions including surrogates of ReLU such as

---

[2] $\theta^{(0)}$ can be viewed as a pre-trained model (Cazenavette et al., 2025), or a structured initialization reducing its storage cost to some constant order. If the pre-trained model $\theta^{(0)}$ is trained more than once, then DD becomes beneficial as it avoids the storage of each fine-tuned model.

softplus.

# 4. Main Result

We now state our main result followed by a proof sketch. Our goal is to 1) explicitly formulate the result of each distillation process (at $t = 1$ and $t = 2$) and 2) evaluate the required size of created synthetic datasets $\mathcal{D}_1^S$ and $\mathcal{D}_2^S$ so that retraining $f_\theta$ with the former in the first phase and with the latter in the second phase leads to a parameter $\tilde{\theta}^*$ whose generalization performance preserves that of the baseline trained on the large dataset $\mathcal{D}^{Tr}$ across both phases. Theorem 4.1 establishes the behavior of the first distilled dataset, Theorem 4.2 then evaluates the sufficient size of $\mathcal{D}_1^S$ to successfully substitute $D^{Tr}$ in the first phase, and Theorem 4.5 analyzes the behavior and the sufficient size of $\mathcal{D}_2^S$ to replace $D^{Tr}$ in the second phase. Please refer to Appendix B for the proof of single index models, and Appendix D for that of multi-index models.

## 4.1. Structure Distillation and Memory Complexity

Our first result is that the low-dimensional intrinsic structure of the task, represented here as $S^*$, is encoded into the first distilled data $\mathcal{D}_1^S$ by Algorithm 1.

**Theorem 4.1** (Latent Structure Encoding). *Under Assumptions 3.1, 3.3, 3.4, 3.6, 3.7 and 3.8, we consider $\mathcal{D}_1^S$ with initializations $\{\tilde{x}_m^{(0)}, \tilde{y}_m^{(0)}\}_{m=1}^{M_1}$ where $\|\tilde{x}_m^{(0)}\| \sim U(S^{d-1})$ and $\tilde{y}_m^{(0)}$ is some constant. Then, with high probability, Algorithm 1 returns $\mathcal{D}_1^S = \{\tilde{x}_m^{(1)}, \tilde{y}_m^{(0)}\}_{m=1}^{M_1}$ with $\tilde{x}_m^{(1)} \propto H\tilde{x}_m^{(0)} +$ (lower order term) for all $m \in [M_1]$.*

Please refer to Appendices B.3 and D.3 for the proofs. Assumption 3.3 directly implies that $\tilde{x}_m^{(0)}$ is effectively projected onto the principal subspace $S^*$ up to lower order terms, and $\mathcal{D}_1^S$ now contains much cleaner information on the latent structure than randomly generated training points. This set can be applied on its own to transfer learning (see Section 5).

$W^{(0)}$ can be trained with $\mathcal{D}_1^S$ from $\theta^{(0)} = (a^{(0)}, W^{(0)}, 0)$, resulting in $\theta^{(1)} = (a^{(0)}, W^{(1)}, b^{(0)})$ (retraining phase of step $t = 1$) where $b^{(0)}$ is the value after reinitialization. Interestingly, only $M_1 \sim \tilde{\Theta}(r^2)$ is sufficient to guarantee that the teacher training of $t = 2$ can train the second layer and infer a model with low population loss.[3]

**Theorem 4.2.** *Under assumptions of Theorem 4.1, consider the teacher training at $t = 2$ of $f_{\theta_0^{(1)}}$ with $\theta^{(1)} = (a^{(0)}, W^{(1)}, b^{(0)})$. If $M_1 \geq \tilde{\Omega}(r^2)$, $d \geq \tilde{\Omega}(r^{8p+3})$ $N \geq \tilde{\Omega}(r^{8p+1}d^4)$, $LJ \geq \tilde{\Omega}(r^{8p+1}d^4)$ and $(\tilde{y}_m^{(0)})^2 \sim \chi(d)$. Then,*

---

[3]We only show the dependence on $r$ and $d$ for the bounds here. The precise formulation of the statement can be found in the appendices.

*there exist hyperparameters $\eta_1^D$, $\lambda_1^D$, $\eta_1^R$, $\lambda_1^R$, $\eta_2^{Tr}$, $\lambda_2^{Tr}$ and $\xi_2^{Tr}$ so that at $t = 2$ the teacher training finds $a^*$ that satisfies with probability at least 0.99 with $\theta^* = (a^*, W^{(1)}, b^{(0)})$,*

$$\mathbb{E}_{x,y}[|f_{\theta^*}(x) - y|] - \zeta \leq \tilde{O}\left(\sqrt{\frac{dr^{3p}}{N}} + \sqrt{\frac{r^{3p}}{L}} + \frac{1}{N^{1/4}}\right).$$

Please refer to Appendix D.4 for the proof. Importantly, Theorem 1 of Damian et al. (2022) states learning $f^*$ requires $\tilde{\Omega}(d^2 \vee d/\epsilon \vee 1/\epsilon^4)$ general training data points. In contrast, thanks to DD, we only need to save $\mathcal{D}_1^S$ and the information of $a^*$, which amounts to a memory complexity of $\tilde{\Theta}(r^2d + L)$. This clearly shows that one of the key mechanisms behind the empirical success of DD in achieving a high compression rate resides in its ability to capture the low-dimensional structure of the task and translate it into well-designed distilled sets for smoother training.

For single index models, we can prove a stronger result which shows that *one* distilled data point $M_1 = 1$ can be sufficient.

**Theorem 4.3.** *Under assumptions of Theorem 4.1, consider the teacher training at $t = 2$ of $f_{\theta_0^{(1)}}$ with $\theta^{(1)} = (a^{(0)}, W^{(1)}, b^{(0)})$. For $r = 1$, if $M_1 = 1$, $N \geq \tilde{\Omega}(d^4)$, $J \geq \tilde{\Omega}(d^4)$, and $\langle \beta_1, \tilde{x}_1^{(0)} \rangle$ is not too small (i.e., with order $\tilde{\Theta}(d^{-1/2})$), then there exist hyperparameters $\eta_1^D$, $\lambda_1^D$, $\eta_1^R$, $\lambda_1^R$, $\eta_2^{Tr}$, $\lambda_2^{Tr}$ and $\xi_2^{Tr}$ so that at $t = 2$ the teacher training finds $a^*$ that satisfies with probability at least 0.99 with $\theta^* = (a^*, W^{(1)}, b^{(0)})$,*

$$\mathbb{E}_{x,y}[|f_{\theta^*}(x) - y|] - \zeta \leq \tilde{O}\left(\sqrt{\frac{d}{N}} + \sqrt{\frac{1}{L}} + \frac{1}{N^{1/4}}\right).$$

Please refer to Appendix B.5 for the proof.

## 4.2. Distillation of Second Phase $t = 2$

We now turn to the second DD (distillation phase $t = 2$) which distills the teacher training that finds $a^*$ of Theorems 4.2 and 4.3. We start by observing that $a^*$ already has a compact memory storage of $\Theta(L)$, and we may just store it to achieve the lowest memory cost. Below, we discuss DD methods that creates $\mathcal{D}_2^S$ with the same order of memory complexity. On the one hand, we can prepare a set of points $\{\hat{p}_m, \hat{y}_m\}_{m=1}^{M_2}$ where $\hat{p}_m$ lies in the *feature space* $\mathbb{R}^L$. Since we only train the second layer at the second step, this training is equivalent to a LRR. We can then employ the result of Chen et al. (2024b) and conclude that $M_2 = 1$ is sufficient. On the other hand, we show that Algorithm 1 also constructs a compact set and reproduces such $a^*$ at retraining under a regularity condition on the initialization of $\mathcal{D}_2^S = \{(\hat{x}_m^{(0)}, \hat{y}_m^{(0)})\}_{m \in [M_2]}$.

**Assumption 4.4** (Regularity Condition). The second distilled dataset $\mathcal{D}_2^S$ is initialized as $\{(\hat{x}_m^{(0)}, \hat{y}_m^{(0)})\}_{m=1}^{M_2}$ so that the kernel of $f_{\theta^{(1)}}$ after $t = 1$, $(\tilde{K})_{im} = \sigma(\langle w_i^{(1)}, \hat{x}_m^{(0)} \rangle + b_i^{(0)})$ has the maximum attainable rank, and its memory cost does not exceed $\tilde{\Theta}(r^2 d + L)$. When $D := \{i \mid w_i^{(1)} \neq 0\}$, the maximum attainable rank is $|D| + 1$ if there exists an $i_0 \in [L] \setminus D$ such that $b_{i_0} > 0$, and $|D|$ otherwise.

Please refer to Appendix B.8 for further discussion.[4] Under this condition, one-step GM and one-step PM can directly create $\mathcal{D}_2^S$ that reconstructs the final layer.

**Theorem 4.5.** *Under assumptions of Theorem 4.2 and regularity condition 4.4, there exist hyperparameters $\lambda_2^S$, $\eta_2^D$, $\lambda_2^D$, $\xi_2^D$, $\eta_2^R$, $\lambda_2^R$, $\xi_2^R$ so that second distillation phase of Algorithm 1 (both one-step PM and one-step GM) finds $\mathcal{D}_2^S = \{(\hat{x}_m^{(0)}, \hat{y}_m^{(\xi_2^D)})\}_{m=1}^{M_2}$ so that retraining with initial state $\theta^{(0)}$ and dataset $\mathcal{D}_1^S \cup \mathcal{D}_2^S$ provides $\tilde{\theta}^*$ that achieves the same error bound as $\theta^*$ of Theorem 4.2.*

In all cases discussed above, we obtain the same memory complexity as follows

**Theorem 4.6.** *The overall memory complexity in terms of training data to obtain an $f_\theta$ with generalization error below $\epsilon$ is reduced by DD from $\Theta(d^3 \vee d^2/\epsilon \vee d/\epsilon^4 \vee dL)$ to $\tilde{\Theta}(r^2 d + L)$, which is lower than the model storage cost of $\Theta(dp)$.*

Please refer to Appendices B.7 and D.5 for the proofs.

### 4.3. Proof Sketch

The key points of our proof can be summarized as the following three parts.

**(i) Feature Extraction at $t = 1$** We show that the teacher gradient extracts information from the principal subspace (Lemma B.14), which follows from Damian et al. (2022), and then that this is transmitted to the population gradient (Theorem B.18 and D.8), leading to $\mathcal{D}_1^S$ projected onto $S^*$.

**(ii) Teacher Learning at $t = 2$** Since $\mathcal{D}_1^S$ now captures the information of the low dimensional structure, we prove by construction that a second layer with a good generalization ability of the whole task exists. Intuitively, as we already have the information of $B$ in $f^*(x) = \sigma^*(B^\top x)$, we look for the right coefficient to reconstruct $\sigma^*$ with the last layer. By equivalence between ridge regression and norm constrained linear regression, the ridge regression of

---

[4]We believe that such a construction can be obtained empirically, since increasing $M_2$ never decreases the rank of $\tilde{K}$, and makes unfavorable event almost unlikely by randomness of $b^{(0)}$. In Appendix B.8 we provide different constructions that work well in our experiments and have a theoretical guarantee and compact memory cost as required.

teacher training at $t = 2$ can find a second layer as good as the one we constructed.

**(iii) Second Distillation Phase** We show that the final direction of the parameters can be encoded into $\mathcal{D}_2^S$ under a regularity condition 4.4, especially into its labels, with both one-step PM and GM. The crux is that under this condition, $a^* \in \text{col}(\tilde{K})$, and we can import information of the last layer to the set $\hat{y}_m^{(1)}$, which reconstructs $a^*$ at retraining.

## 5. Synthetic Experiments

In this section, we provide experiments based on synthetic data. The goal of this section is twofold: (i) to examine the scaling trends that align with the predicted dependence of our main results, and (ii) to illustrate the efficiency of DD in downstream transfer learning tasks. Please refer to Appendix E for experiments using real-world data and practical models.

### 5.1. Theoretical Illustration

Here, we present an illustration of our result in terms of sample complexity of $N$ and $J$ and show that it matches our theory. We set $f^*(x) = \sigma^*(\langle \beta, x \rangle)$ where $\sigma^*(z) = \text{He}_2(z)/2 + \text{He}_4(z)/4!$, $\zeta = 0$, $r = 1$, $d = 10$, and $L = 100$. Since this is a single index model, we prepare one synthetic point for $\mathcal{D}_1^S = \{(\tilde{x}^{(0)}, \tilde{y}^{(0)})\}$ following Theorem 4.3. $\mathcal{D}_2^S$ is initialized following the construction of Appendix B.8. We run Algorithm 1 and compare the generalization error of the network trained according to five different paradigms, namely, the vanilla training with the full training data, that with the obtained distilled data and its corresponding random baseline where we replaced $\mathcal{D}_1^S$ and $\mathcal{D}_2^S$ used in the training with random points of $\mathcal{D}^{Tr}$ (Random II), the result of teacher training at $t = 2$ and its corresponding random baseline where we replaced $\mathcal{D}_1^S$ used in the training by a random point of the original dataset $\mathcal{D}^{Tr}$ (Random I). Results are plotted in Figure 1 for different sizes of $N = \{10, 10^2, 10^3, 10^4, 10^5\}$ and effective random initialization $J^* = LJ/2 = \{10, 10^2, 10^3, 10^4, 10^5\}$, which is the actual number of directions the gradient update can see for the first distillation step (see Corollary B.7).

As we can observe, random baselines cannot reproduce a performance close to the original training, while one distilled point for $\mathcal{D}_1^S$ is enough. Furthermore, the generalization loss of the distilled data decreases as $J$ and $N$ increase. These illustrate the compression efficiency of DD and the necessity of large $N$ and $J$ implied in Theorem 4.5.

### 5.2. Application to Transfer Learning

An implication of Theorem 4.2 is that $\mathcal{D}_1^S$ can be used to learn other functions $f^*(x) = \sigma^*(B^\top x)$ that possess

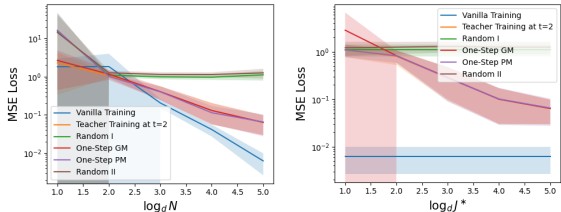

*Figure 1.* Dependence of training data size with $J^* = 10^5$ (left figure) and initialization batch size with $N = 10^5$ (right) with respect to the achieved MSE loss. Mean and standard deviation over five seeds.

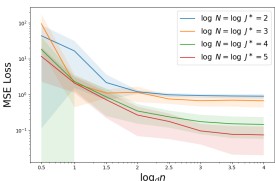

*Figure 2.* MSE loss with respect to the training data size $n$ used to fine-tune a model pre-trained with data distilled from the earlier training of a function with the same principal subspace. $N$ and $J^*$ are the parameters for this training before. Mean and standard deviation over five seeds.

the same principal subspace but different $\sigma^*$. We use $\mathcal{D}_1^S$ obtained from the previous experiment to learn a novel function $g^*(x) = \text{He}_3(\langle \beta, x \rangle)/\sqrt{3!}$ with the identical underlying structure as $f^*$. $\mathcal{D}_1^S$ is computed with $N$ training data of $f^*$ and $J^*$ initializations following Algorithm 1. Weights were then pre-trained with the resulting $\mathcal{D}_1^S$, and we fine-tuned the second layer with $n$ samples from $g^*$. The result is plotted in Figure 2.

Theorem 3 of Damian et al. (2022) states that with such pre-trained weights the number of $n$ does not scale with $d$ anymore. We can observe the same phenomenon in Figure 2 where population loss is already low for smaller $n$ compared to Figure 1. Moreover, spending more resources on computing the distilled set $\mathcal{D}_1^S$ for the first task leads to higher performance on the subsequent task. Note that learning $g^*$ is a difficult problem and neural tangent kernel requires $n \gtrsim d^3$ to achieve non-trivial loss (Damian et al., 2022). Importantly, this result was computed over random initializations of the whole network, showing the robustness of this approach over initial configurations in this kind of problems. This result aligns with general transfer learning scenarios where we assume several tasks share a common underlying structure, and one general pre-trained model can be fine-tuned with low training cost to each of them. Since our distilled dataset needs less memory storage than the pre-trained model, this experiment supports the idea that DD provides a compact summary of previous knowledge that can be deployed at larger scale for applications such as transfer learning (Lee et al., 2024).

## 6. Discussion and Conclusion

One notable feature of our result is that DD exploits the intrinsic dimensionality of the task to realize a high compression rate, primarily during the early stage of distillation, as frequently suggested in prior work (Zhao et al., 2021). This suggests that the necessary distilled size should scale with task complexity rather than ambient dimension alone and provides a concrete theory-supported perspective on how the distilled dataset size should scale in practice. This also confirms prior empirical findings that DD encodes information from the beginning of teacher training, which can be sufficient to attain strong distillation performance on certain tasks (Zhao et al., 2021; Yang et al., 2024). While the first distillation principally captures information from $S^*$ and can be utilized to other tasks such as transfer learning, the second distillation can be interpreted as an architecture-specific distillation that encodes more fine-tuned information of the training. Since a factor of $d$ is unavoidable as soon as we save a point in the input dimension, the obtained memory complexity of $\tilde{\Theta}(r^2 d + L)$ depends only fundamentally on the intrinsic structure of the task and the architecture of the neural network. At the same time, this efficient retraining and compact distilled dataset come at the cost of increased computation relative to standard training. Especially, the initialization batch size $J$ scales with $d$. While we expect the exponents can be improved, this computational overhead appears to be the price for the strong performance of DD. Moreover, our analysis reveals a clear trade-off between PM and GM. The former has lower computational complexity during distillation but higher cost during retraining, and *vice versa*. Finally, our result supports a principled strategy for DD. Distilling earlier training phases may capture intrinsic features that are not captured later. Consequently, aggregating information from the entire training trajectory into a single distilled dataset may be inefficient, and a careful distillation appropriate to each phase of the training constitutes a key factor to improve the performance of DD. From this perspective, the progressive DD approach of Cazenavette et al. (2025) not only facilitates a cleaner analysis of the complex machinery of DD but also contributes to creating effective distilled datasets.

**Comparison with Prior Works**  To the best of our knowledge, we provide the first analysis of DD dynamics that accounts for task structure, non-linear models, and gradient-based optimization. This leads to several novel results that previous research could not obtain. Notably, our theoretical framework shows that DD can identify the intrinsic structure of the task, which helps explain why the performance of DD varies in function of task difficulty, beyond previous KRR settings. By carefully tracking the algorithmic mechanism when all layers of the model are trained, we could quantify the memory complexity of DD as $\tilde{\Theta}(r^2 d + L)$. In compari-

son, for the same model, Chen et al. (2024b) would require a memory complexity $\Theta(Ld)$. $\Theta(Ld)$ is also the storage cost of the entire two-layer neural network. Our result is proved for one-step distillation methods, which also helps to explain why such techniques can be competitive in practice. Furthermore, throughout our proofs, random initializations play a crucial role in isolating the fundamental information to be distilled. This feature was not considered in prior work, and we highlight it accordingly.

**Potential Benefit of Pre-Trained Models** In our theorem, the dependence on $d$ is mainly absorbed into $N$ and $J$. This dependence may be improved by leveraging information from pre-trained models which are also subjected to DD to reduce their fine-tuning cost (Cazenavette et al., 2025). Indeed, as such models have been trained on a wide range of tasks, their weights may concentrate on a lower-dimensional structure that reflects a moderately small effective dimension $r^*$ of the environment which includes the small $r$-dimensional subspace of our problem setting. As a result, rather than using generic isotropic distributions for the initialization of the parameters as in our analysis, one could estimate a task-relevant subspace directly from the pre-trained weights, extracting a principal subspace of dimension $r^*$ and restrict the sampling distribution to that $r^*$-dimensional subspace. This would replace the dependence of $N$ and $J$ on $d$ by a dependence on $r^*$. We believe this is a promising approach for future work.

**Extension to Deeper Networks** Our analysis can be interpreted in the context of deeper neural networks by viewing those models as consisting of a feature-learning layer followed by an output function comprising the remaining layers. Under this perspective, the first distilled dataset of Section 4.1 can be used to retrain the feature-learning layer, while the analysis in Section 4.2 extends to the subsequent layers. This suggests that similar distilled datasets capturing the low-dimensionality of the task exist that can be reused across different deeper architectures, expanding our theory to deeper networks and, at the same time, providing a theoretical explanation for cross-architecture transfer as well. Nevertheless, we do not consider this a full treatment of deeper networks, as it remains unclear what fundamentally new theoretical insights would arise beyond the core mechanism already identified in our paper when moving to deeper architectures, which would require a more involved analysis since learned representation evolves jointly across multiple layers in this broader setting. This is beyond the scope of our analysis and this work.

**Other Limitations** One limitation of our work resides in the function class we analyze, that of multi-index models, which may not fully represent practical settings. Nevertheless, this captures the essence of the complex interaction that happens inside DD, such as the low-dimensional latent structure of the task, which was not considered in prior theoretical investigations of DD. The feature-learning analysis in DD is more involved than in standard supervised training, since DD contains several interacting stages (teacher training, student training, distillation, and retraining) and each stage has its own optimization dynamics rather than a single end-to-end training process as we tried to keep the algorithmic procedure as close as possible to practice using gradient-based formulation. Analyzing feature learning across these coupled stages is therefore substantially more complex, and constituted one of the main technical challenges of the paper. Our work on its own thus advances theoretical understanding and develops tools that can serve as a foundation for even more practice-aligned models. The initialization of $\mathcal{D}_1^S$ and $\mathcal{D}_2^S$ may also look too model-specific. However, the scope of our work does not include cross-architecture generalization, and thus this does not undermine the validity of our results. Analogous construction procedures could be developed for other deeper neural networks, which is also left for future work.

In conclusion, we analyzed DD applied to an optimization framework for two-layer neural networks with ReLU activation functions trained on a task endowed with low-dimensional latent structures. We proved not only that DD leverages such a latent structure and encodes its information into well-designed distilled data, but also that the memory storage necessary to retrain a neural network with high generalization score is decreased to $\tilde{\Theta}(r^2 d + L)$. While the primary goal of this work was to elucidate how DD exploits intrinsic patterns of a training regime, the setting we analyzed remains simplified relative to practical deployments, and extending the theory to more realistic regimes constitutes an essential avenue for future work. Nevertheless, we believe this paper contributes on its own to the development and deeper understanding of DD, ultimately helping advance it towards a reliable methodology for reducing the data and computational burdens of modern deep learning.

## Acknowledgements

YK was supported by JST ACT-X (JPMJAX25CA) and JST BOOST (JPMJBS2418). NN was partially supported by JST ACT-X (JPMJAX24CK) and JST BOOST (JPMJBS2418). TT is supported by RIKEN Center for Brain Science, RIKEN TRIP initiative (RIKEN Quantum), JST CREST program JPMJCR23N2, and JSPS KAKENHI 25K24466. We thank Taiji Suzuki for helpful discussions.

## Impact Statement

This paper presents work whose goal is to advance the field of Machine Learning. There are many potential societal

consequences of our work, none which we feel must be specifically highlighted here.

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

# A. Further Notations, Tensor Computation and Hermite Polynomials

## A.1. Notations

In this paper, we employ the notion of high probability events defined as follows:

**Definition A.1.** Throughout the proofs, $\iota$ is used to denote any quantity such that $\iota = C_\iota \log(NLd)$ for sufficiently large constant $C_\iota$, and will be redefined accordingly. Moreover, an event is said to happen *with high probability* if its probability is at least $1 - \mathrm{poly}(N, L, d)\mathrm{e}^{-\iota}$ where $\mathrm{poly}(N, L, d)$ is a polynomial of $N$, $L$ and $d$ that does not depend on $C_\iota$.

Any union bound of a number of $\mathrm{poly}(N, L, d)$ of high probability events is itself a high probability events. Since all our parameters will scale up to $\mathrm{poly}(N, L, d)$ orders, we can take such union bound safely. For example, the following holds for the Gaussian distribution and the uniform distribution over $S^{d-1}$.

**Lemma A.2.** *If $x \sim N(0, I_d)$, then $\|x\|^2 = \tilde{\Theta}(d)$ with high probability.*

*Proof.* From Theorem 3.1.1 of Vershynin (2018), $\sqrt{d} - t \leq \|x\| \leq \sqrt{d} + t$ with probability at least $1 - 2\mathrm{e}^{-ct^2}$, where $c$ is a universal constant independent of $d$ and $t$. $t = \frac{1}{2}\sqrt{d}$ leads to $\frac{1}{2}\sqrt{d} \leq \|x\| \leq \frac{3}{2}\sqrt{d}$ with probability at least $1 - 2\mathrm{e}^{-cd}$, which is a high probability event. $\square$

**Lemma A.3.** *If $x \sim U(S^{d-1})$ and $\|\beta\| = 1$, then $|\langle x, \beta \rangle| \lesssim 1/\sqrt{d}$ with high probability.*

*Proof.* This follows from Corollary 46 of Damian et al. (2022). $\square$

We also denote for a matrix $K$ the span of its column as $\mathrm{col}(K)$.

## A.2. Tensor Computation

Our proofs for single-index and multi-index models involve tensor computations. Therefore, we present a brief clarification of symbols and notations used in this paper. Those are usual notations also used in other works (Damian et al., 2022, 2023; Oko et al., 2024b).

**Definition A.4.** A $k$-tensor is defined as the generalization of matrices with multiple indices. Let $A \in (\mathbb{R}^d)^{\otimes k}$ and $B \in (\mathbb{R}^d)^{\otimes l}$ be respectively a $k$-tensor and an $l$-tensor. The $(i_1, \ldots, i_k)$-th entry of $A$ is denoted by $A_{i_1, \ldots, i_k}$ with $i_1, \ldots, i_k \in [d]$. Moreover, when $k \geq l$, the tensor action $A(B)$ is defined as

$$A(B)_{i_1, \ldots, i_{k-l}} := \sum_{j_1, \ldots, j_l} A_{i_1, \ldots, i_{k-l}, j_1, \ldots, j_l} B_{j_1, \ldots, j_l}$$

and is thus a $k - l$-tensor. We also remind the following usual definitions.

**Definition A.5.** For a vector $v \in \mathbb{R}^d$, $v^{\otimes k}$ is a $k$-tensor with $(i_1, \ldots, i_k)$-th entry $v_{i_1} \cdots v_{i_k}$.

**Definition A.6.** For differentiable function $f : \mathbb{R}^d \to \mathbb{R}$, $\nabla^k f(x)$ is a $k$-tensor with $(i_1, \ldots, i_k)$-th entry

$$\frac{\partial}{\partial x_{i_1}} \cdots \frac{\partial}{\partial x_{i_k}} f(x).$$

**Definition A.7.** For a matrix $M \in \mathbb{R}^{d \times r}$, we define the action $M^{\otimes k}$ on a $k$-tensor $A \in (\mathbb{R}^r)^{\otimes k}$ as $M^{\otimes k} A \in (\mathbb{R}^d)^{\otimes k}$ with $(i_1, \ldots, i_k)$-th entry

$$\sum_{j_1, \ldots, j_k \in [r]} M_{i_1, j_1} \cdots M_{i_k, j_k} A_{j_1, \ldots, j_k}.$$

**Lemma A.8.** *For a sufficiently smooth function $\sigma^* : \mathbb{R}^r \to \mathbb{R}$ and $B \in \mathbb{R}^{d \times r}$, $\nabla_x^k \sigma^*(B^\top x) = B^{\otimes k} \nabla_z^k \sigma^*(B^\top x)$.*

*Proof.* Let $x \in \mathbb{R}^d$, and set $z = B^\top x \in \mathbb{R}^r$. Then,

$$z_j = \sum_{i=1}^d B_{ij} x_i.$$

By the chain rule,

$$\frac{\partial}{\partial x_i} = \sum_{j=1}^r \frac{\partial z_j}{\partial x_i}\frac{\partial}{\partial z_j} = \sum_{j=1}^r B_{ij}\frac{\partial}{\partial z_j}.$$

As a result,

$$\frac{\partial^k \sigma^*(B^\top x)}{\partial x_{i_1}\cdots\partial x_{i_k}} = \left(\sum_{j_1=1}^r B_{i_1 j_1}\frac{\partial}{\partial z_{j_1}}\right)\cdots\left(\sum_{j_k=1}^r B_{i_k j_k}\frac{\partial}{\partial z_{j_k}}\right)\sigma^*(z) = \sum_{j_1,\dots,j_k=1}^r \left(\prod_{l=1}^k B_{i_l j_l}\right)\frac{\partial^k\sigma^*(z)}{\partial z_{j_1}\cdots\partial z_{j_k}}.$$

$\square$

**Lemma A.9.** *For an orthogonal matrix $B \in \mathbb{R}^{d\times r}$ with $B^\top B = I_r$ and a $T \in (\mathbb{R}^r)^{\otimes k}$, $\|B^{\otimes k}T\|_F = \|T\|_F$.*

*Proof.* By definition of the Frobenius norm,

$$\|B^{\otimes k}T\|_F^2 = B^{\otimes k}T(B^{\otimes k}T) = T((B^\top)^{\otimes k}B^{\otimes k}T) = T((B^\top B)^{\otimes k}T) = T(T) = \|T\|_F.$$

$\square$

We also define the symmetrization as follows.

**Definition A.10.** For a $k$-tensor $T$, its symmetrization $\mathrm{Sym}(T)$ is defined as

$$(\mathrm{Sym}(T))_{i_1,\dots,i_k} := \frac{1}{k!}\sum_{\pi\in\mathcal{P}_k} T_{i_{\pi(1)},\dots,i_{\pi(k)}},$$

where $\pi$ is a permutation and $\mathcal{P}_k$ is the symmetric group on $1,\dots,k$.

**Lemma A.11.** *For any tensor $T$, $\|\mathrm{Sym}(T)\|_F \le \|T\|_F$.*

*Proof.* Please refer to Lemma 1 of Damian et al. (2023). $\square$

**Definition A.12.** A $k$-tensor $T$ is symmetric if for any permutation $\pi \in \mathcal{P}_k$,

$$T_{i_1,\dots,i_k} = T_{i_{\pi(1)},\dots,i_{\pi(k)}}.$$

**Lemma A.13.** *$v^{\otimes k}$ is a symmetric $k$-tensor. If $f$ is sufficiently smooth, then $\nabla^k f(x)$ is also a symmetric $k$-tensor.*

**Lemma A.14.** *If $C$ is a symmetric $k$-tensor and $B$ an $l$-tensor so that $l \le k$, then $C(B) = C(\mathrm{Sym}(B))$.*

*Proof.* By definition,

$$
\begin{aligned}
C(\mathrm{Sym}(B))_{i_1,\dots,i_{k-l}} &= \sum_{j_1,\dots,j_l} C_{i_1,\dots,i_{k-l},j_1,\dots,j_l}\mathrm{Sym}(B)_{j_1,\dots,j_l}\\
&= \sum_{j_1,\dots,j_l} C_{i_1,\dots,i_{k-l},j_1,\dots,j_l}\frac{1}{l!}\sum_{\pi\in\mathcal{P}_l} B_{j_{\pi(1)},\dots,j_{\pi(l)}}\\
&= \frac{1}{l!}\sum_{\pi\in\mathcal{P}_l}\sum_{j_1,\dots,j_l} C_{i_1,\dots,i_{k-l},j_1,\dots,j_l} B_{j_{\pi(1)},\dots,j_{\pi(l)}}.
\end{aligned}
$$

Since the permutation $\pi$ is a bijective operation, the change of variable $j'_1 = j_{\pi(1)}, \ldots, j'_l = j_{\pi(l)}$ leads to

$$
\begin{aligned}
C(\mathrm{Sym}(B))_{i_1,\ldots,i_{k-l}} &= \frac{1}{l!} \sum_{\pi \in \mathcal{P}_l} \sum_{j_1,\ldots,j_l} C_{i_1,\ldots,i_{k-l},j_1,\ldots,j_l} B_{j_{\pi(1)},\ldots,j_{\pi(l)}} \\
&= \frac{1}{l!} \sum_{\pi \in \mathcal{P}_l} \sum_{j'_1,\ldots,j'_l} C_{i_1,\ldots,i_{k-l},j'_{\pi^{-1}(1)},\ldots,j'_{\pi^{-1}(l)}} B_{j'_1,\ldots,j'_l} \\
&= \frac{1}{l!} \sum_{\pi \in \mathcal{P}_l} \sum_{j'_1,\ldots,j'_l} C_{i_1,\ldots,i_{k-l},j'_1,\ldots,j'_l} B_{j'_1,\ldots,j'_l} \\
&= \frac{1}{l!} \sum_{\pi \in \mathcal{P}_l} C(B)_{i_1,\ldots,i_{k-l}} \\
&= \frac{1}{l!} l! C(B)_{i_1,\ldots,i_{k-l}} \\
&= C(B)_{i_1,\ldots,i_{k-l}},
\end{aligned}
$$

where we used the symmetry of $C$ in the third equality. $\qquad\square$

### A.3. Hermite Polynomials

Based on the above notation, we introduce Hermite Polynomials and Hermite expansion. These will be briefly used to explain a few properties.

**Definition A.15.** The $k$-th Hermite polynomial $\mathrm{He}_k$ is a $k$-tensor in $(\mathbb{R}^d)^{\otimes k}$ defined as

$$
\mathrm{He}_k(x) := (-1)^k \frac{\nabla^k \mu(x)}{\mu(x)},
$$

where $\mu(x) = \frac{1}{(2\pi)^{d/2}} e^{-\|x\|^2/2}$.

**Definition A.16.** The Hermite expansion of a function $f : \mathbb{R}^d \to \mathbb{R}$ with $\mathbb{E}_{x \sim N(0,I_d)}[f(x)^2] < \infty$ is defined as

$$
f = \sum_{k \geq 0} \frac{1}{k!} \langle \mathrm{He}_k(x), T_k \rangle,
$$

where $T_k$ is a symmetric $k$-tensor such that $T_k = \mathbb{E}_{x \sim N(0,I_d)}[\nabla^k_x f(x)^2]$.

For example, the Hermite expansion of $\sigma(x) = \max\{0, x\}$ is

$$
\begin{aligned}
\sigma(x) &= \frac{1}{\sqrt{2\pi}} + \frac{1}{2}x + \frac{1}{\sqrt{2\pi}} \sum_{k \geq 1} \frac{(-1)^{k-1}}{k! 2^k (2k-1)} \mathrm{He}_{2k}(x) \\
&=: \sum_{k \geq 0} \frac{c_k}{k!} \mathrm{He}_k(x),
\end{aligned}
$$

where $0 = c_3 = c_5 = \ldots$. For its derivative, we obtain

$$
\begin{aligned}
\sigma'(x) &= \frac{1}{2} + \frac{1}{\sqrt{2\pi}} \sum_{k \geq 0} \frac{(-1)^k}{k! 2^k (2k+1)} \mathrm{He}_{2k+1}(x) \\
&= \sum_{k \geq 0} \frac{c_{k+1}}{k!} \mathrm{He}_k(x).
\end{aligned}
$$

## B. Proof of Main Theorems: Single Index Models

In this appendix, we prove our result for the single index setting where we define $f^*(x)$ as $\sigma^*(\langle \beta, x \rangle)$, notably showing that $M_1 = 1$ can be enough. Please refer to Appendix D for the general multi-index case. Some statements will be proved in

the multi-index setting so that they can be used later in Section D, while others correspond to the concrete case $r = 1$ of theorems, lemmas, or corollaries presented in that later section. Their proofs are nevertheless included here, as single-index models are extensively studied as an independent topic within the broader framework of machine learning (Dhifallah & Lu, 2020; Gerace et al., 2020; Ba et al., 2022; Oko et al., 2024b; Nishikawa et al., 2025).

The proof is constructed following each step of Algorithm 1 to keep as much clarity as possible. That is, we analyze Algorithm 1 phase by phase, using the result of each phase to study the next. Our goal is to 1) explicitly formulate the result of each distillation process (at $t = 1$ and $t = 2$) and 2) evaluate the required size of created synthetic datasets $\mathcal{D}_1^S$ and $\mathcal{D}_2^S$ so that retraining $f_\theta$ with the former in the first phase and with the latter in the second phase leads to a parameter $\tilde{\theta}^*$ whose generalization performance preserves that of the baseline trained on the large dataset $\mathcal{D}^{Tr}$ across both phases. As ReLU is invariant to scaling, we can consider without loss of generality that all the weights are normalized $\|w_i\| = 1$ and $w_i \sim U(S^{d-1})$. We use abbreviations like $\mathcal{L}^{Tr}(\cdot)$ and $\mathcal{L}^S(\cdot)$ when the data we use to evaluate the loss for a given parameter is clear.

### B.1. $t = 1$ Teacher Training

The corresponding equation in Algorithm 1 is

$$I_j^{Tr} = \{\theta_j^{(1)}, \, G_j^{(1)}\} = \mathcal{A}_1^{Tr}(\theta_j^{(0)}, \mathcal{D}^{Tr}, \xi_1^{Tr}, \eta_1^{Tr}, \lambda_1^{Tr}).$$

From Assumption 3.6, this is equivalent to

$$W_j^{(1)} = W_j^{(0)} - \eta_1^{Tr} \left\{ \nabla_W \mathcal{L}^{Tr}(\theta_j^{(1)}) + \lambda_1^{Tr} W_j^{(1)} \right\},$$

where $\eta_1^{Tr} = \tilde{\Theta}(\sqrt{d})$ and $\lambda_1^{Tr} = \left( \eta_1^{Tr} \right)^{-1}$. The teacher gradient for the $j$-th initialization of weight $w_i$ can be defined as

$$g_{i,j}^{Tr} := (G_j^{(1)})_i = \nabla_{w_i} \mathcal{L}^{Tr}(\theta_j^{(0)}).$$

$\nabla_{w_i} \mathcal{L}^{Tr}(\theta_j^{(0)})$ can be concretely computed with the following lemma. We denote $w_{i,j}^{(0)}$ as the $j$-th realization of the initialization of the $i$-th weight $w_i$.

**Definition B.1.** We define $\hat{f}^*(x)$ as $f^*(x) - \alpha - \langle \gamma, x \rangle$.

**Lemma B.2.** *Given a training data $\mathcal{D} = \{(x_n, y_n)\}_{n=1}^N$, the gradient of the training loss with respect to the weight $w_i$ is*

$$\nabla_{w_i} \mathcal{L}(\theta, \mathcal{D}) = \frac{1}{N} \sum_n \left( f_\theta(x_n) - y_n \right) \nabla_{w_i} f_\theta(x_n)$$

$$= \frac{1}{N} \sum_n \left( f_\theta(x_n) - y_n \right) a_i x_n \sigma'(\langle w_i, x_n \rangle).$$

*Proof.* This follows directly from the definition of the loss and neural network. $\square$

**Corollary B.3.** *For our training data, due to preprocessing and noise,*

$$\nabla_{w_i} \mathcal{L}^{Tr}(\theta) = \frac{1}{N} \sum_n \left( f_\theta(x_n) - \hat{f}^*(x_n) - \epsilon_n \right) \nabla_{w_i} f_\theta(x_n)$$

$$= \frac{1}{N} \sum_n \left( f_\theta(x_n) - \hat{f}^*(x_n) - \epsilon_n \right) a_i x_n \sigma'(\langle w_i, x_n \rangle).$$

### B.2. $t = 1$ Student Training

The corresponding equation in Algorithm 1 is

$$I_j^S = \{\tilde{\theta}_j^{(1)}, \, \tilde{G}_j^{(1)}\} = \mathcal{A}_1^S(\theta_j^{(0)}, \mathcal{L}^S(\theta_j^{(0)}), \xi_1^S, \eta_1^S, \lambda_1^S).$$

From Assumption 3.6, this is equivalent to

$$\tilde{W}_j^{(1)} = W_j^{(0)} - \eta_1^S \left\{ \nabla_W \mathcal{L}(\theta_j^{(0)}, \mathcal{D}_1^S) + \lambda_1^S W_j^{(0)} \right\},$$

where $\lambda_1^S = \left(\eta_1^S\right)^{-1}$. The teacher gradient for the $j$-th initialization of weight $w_i$ can be defined as

$$g_{i,j}^S := (\tilde{G}_j^{(1)})_i = \nabla_{w_i}\mathcal{L}^S(\theta_j^{(0)}).$$

## B.3. $t = 1$ Distillation

### B.3.1. PROBLEM FORMULATION AND RESULT

For the distillation, we consider one-step gradient matching. As mentioned in the main paper, it is difficult to consider performance matching since we do not have access to the final result of each training yet. Then,

$$\mathcal{D}_1^S \leftarrow \mathcal{M}_1(\mathcal{D}_0^S, \mathcal{D}^{Tr}, \{I_j^{Tr}, I_j^S\}, \xi_1^D, \eta_1^D, \lambda_1^D)$$

becomes

$$\mathcal{D}_1^S = \mathcal{D}_0^S - \eta_1^D \left( \frac{1}{J} \sum_j \nabla_{\mathcal{D}^s} \left( 1 - \frac{1}{L} \sum_{i=1}^L \langle g_{i,j}^S, g_{i,j}^{Tr}\rangle \right) + \lambda_1^D \mathcal{D}_0^S \right), \tag{1}$$

where we set $\lambda_1^D = (\eta_1^D)^{-1}$.

When $\mathcal{D}_0 = \{(\tilde{x}^{(0)}, \tilde{y}^{(0)})\}$, we can show that $\tilde{x}^{(1)}$ aligns to $\beta$, i.e., the principal subspace $S^*$ of the problem:

**Theorem B.4.** *Under Assumptions 3.1, 3.3, 3.4, 3.6, and 3.8, when $\mathcal{D}_0 = \{(\tilde{x}^{(0)}, \tilde{y}^{(0)})\}$, where $\tilde{x}^{(0)} \sim U(S^{d-1})$, the first step of distillation gives $\tilde{x}^{(1)}$ such that with high probability,*

$$\tilde{x}^{(1)} = -\eta_1^D \tilde{y}^{(0)} \left( c_d \langle \beta, \tilde{x}\rangle \beta + \tilde{O}\left( d^{\frac{1}{2}} N^{-\frac{1}{2}} + d^{-2} + d^{\frac{1}{2}} J^{*-\frac{1}{2}} \right) \right),$$

*where $c_d = \tilde{\Theta}\left( d^{-1} \right)$, $J^* = LJ/2$.*

We prove this theorem step by step from the next section (from Section B.3.2 to B.3.4) .

### B.3.2. FORMULATION OF DISTILLED DATA POINT

In our setting of gradient matching, the right hand side of Equation (1) can be simplified to

$$\tilde{x}^{(1)} = -\frac{\eta_1^D}{J} \sum_j \nabla_{\mathcal{D}^s} \left( 1 - \frac{1}{L} \sum_i \langle g_{i,j}^S, g_{i,j}^{Tr}\rangle \right), \tag{2}$$

which can be written further as follows:

**Lemma B.5.** *Under Assumptions 3.1, 3.3, 3.4, 3.6, 3.7 and 3.8,*

$$\tilde{x}^{(1)} = -\eta_1^D \tilde{y}^{(0)} (G + \epsilon),$$

*where*

$$G = \frac{1}{LJ} \sum_{i,j} g_{i,j} h'(\langle w_{i,j}^{(0)}, \tilde{x}^{(0)}\rangle) + \frac{1}{LJ} \sum_{i,j} w_{i,j}^{(0)} h''(\langle w_{i,j}^{(0)}, \tilde{x}^{(0)}\rangle) \left\langle g_{i,j}, \tilde{x}^{(0)}\right\rangle$$

$$\epsilon = \frac{1}{LJ} \sum_{i,j} \left\{ \frac{1}{N} \sum_n \epsilon_n x_n \sigma'(\langle w_{i,j}^{(0)}, x_n\rangle) \right\} h'(\langle w_{i,j}^{(0)}, \tilde{x}^{(0)}\rangle)$$

$$+ \frac{1}{LJ} \sum_{i,j} w_{i,j}^{(0)} h''(\langle w_{i,j}^{(0)}, \tilde{x}^{(0)}\rangle) \left\langle \frac{1}{N} \sum_n \epsilon_n x_n \sigma'(\langle w_{i,j}^{(0)}, x_n\rangle), \tilde{x}^{(0)}\right\rangle,$$

*with $g_{i,j} = \frac{1}{N} \sum_n \hat{f}^*(x_n) x_n \sigma'(\langle w_{i,j}^{(0)}, x_n\rangle)$.*

*Proof.* From Equation (2),

$$
\begin{aligned}
\tilde{x}^{(1)} =& -\frac{\eta_1^D}{J} \sum_j \nabla_{\tilde{x}^{(0)}} \left( 1 - \frac{1}{L} \sum_{i=1}^{L} \langle g_{i,j}^S, g_{i,j}^{Tr} \rangle \right) \\
=& \frac{\eta_1^D}{LJ} \sum_{i,j} \nabla_{\tilde{x}^{(0)}} \left\langle g_{i,j}^S, g_{i,j}^{Tr} \right\rangle \\
=& \frac{\eta_1^D}{LJ} \sum_{i,j} \nabla_{\tilde{x}^{(0)}}^{\top} \left\{ \left( f_{\theta_j^{(0)}}(\tilde{x}^{(0)}) - \tilde{y}^{(0)} \right) \nabla_{w_i} f_{\theta_j^{(0)}}(\tilde{x}^{(0)}) \right\} \frac{1}{N} \sum_n \left( f_{\theta_j^{(0)}}(x_n) - \hat{f}^*(x_n) - \epsilon_n \right) \nabla_{w_i} f_{\theta_j^{(0)}}(x_n) \\
=& \frac{\eta_1^D}{LJ} \sum_{i,j} \left\{ \nabla_{\tilde{x}^{(0)}}^{\top} f_{\theta_j^{(0)}}(\tilde{x}^{(0)}) \nabla_{w_i} f_{\theta_j^{(0)}}(\tilde{x}^{(0)}) + \tilde{y}^{(0)} \nabla_{\tilde{x}^{(0)}}^{\top} \nabla_{w_i} f_{\theta_j^{(0)}}(\tilde{x}^{(0)}) \right\} \frac{1}{N} \sum_n \left( \hat{f}^*(x_n) + \epsilon_n \right) \nabla_{w_i} f_{\theta_j^{(0)}}(x_n) \\
=& \frac{\eta_1^D}{LJ} \sum_{i,j} \tilde{y}^{(0)} \nabla_{\tilde{x}^{(0)}}^{\top} a_i^{(0)} \tilde{x}^{(0)} h'(\langle w_{i,j}^{(0)}, \tilde{x}^{(0)} \rangle) \frac{1}{N} \sum_n \left( \hat{f}^*(x_n) + \epsilon_n \right) a_i^{(0)} x_n \sigma'(\langle w_{i,j}^{(0)}, x_n \rangle) \\
& + \frac{\eta_1^D}{LJ} \sum_{i,j} (a_i^{(0)})^2 \nabla_{\tilde{x}^{(0)}}^{\top} h(\langle w_{i,j}^{(0)}, \tilde{x}^{(0)} \rangle) \nabla_{w_i} h(\langle w_{i,j}^{(0)}, \tilde{x}^{(0)} \rangle) \frac{1}{N} \sum_n \left( \hat{f}^*(x_n) + \epsilon_n \right) a_i^{(0)} x_n \sigma'(\langle w_{i,j}^{(0)}, x_n \rangle) \\
=& \frac{\eta_1^D \tilde{y}^{(0)}}{LJ} \sum_{i,j} (a_i^{(0)})^2 \left( h'(\langle w_{i,j}^{(0)}, \tilde{x}^{(0)} \rangle) I + \tilde{x}^{(0)} w_{i,j}^{(0)\top} h''(\langle w_{i,j}^{(0)}, \tilde{x}^{(0)} \rangle) \right) \left\{ \frac{1}{N} \sum_n \left( \hat{f}^*(x_n) + \epsilon_n \right) a_i^{(0)} x_n \sigma'(\langle w_{i,j}^{(0)}, x_n \rangle) \right\} \\
& + 0 \\
=& \frac{\eta_1^D \tilde{y}^{(0)}}{LJ} \sum_{i,j} g_{i,j} h'(\langle w_{i,j}^{(0)}, \tilde{x}^{(0)} \rangle) + \frac{\eta_1^D \tilde{y}^{(0)}}{LJ} \sum_{i,j} w_{i,j}^{(0)} h''(\langle w_{i,j}^{(0)}, \tilde{x}^{(0)} \rangle) \left\langle g_{i,j}, \tilde{x}^{(0)} \right\rangle \\
& + \frac{\eta_1^D \tilde{y}^{(0)}}{LJ} \sum_{i,j} \left\{ \frac{1}{N} \sum_n \epsilon_n x_n \sigma'(\langle w_{i,j}^{(0)}, x_n \rangle) \right\} h'(\langle w_{i,j}^{(0)}, \tilde{x}^{(0)} \rangle) \\
& + \frac{\eta_1^D \tilde{y}^{(0)}}{LJ} \sum_{i,j} w_{i,j}^{(0)} h''(\langle w_{i,j}^{(0)}, \tilde{x}^{(0)} \rangle) \left\langle \frac{1}{N} \sum_n \epsilon_n x_n \sigma'(\langle w_{i,j}^{(0)}, x_n \rangle), \tilde{x}^{(0)} \right\rangle,
\end{aligned}
$$

where we defined $g_{i,j} := \frac{1}{N} \sum_n \hat{f}^*(x_n) x_n \sigma'(\langle w_{i,j}^{(0)}, x_n \rangle)$ at the last equality, used Lemma B.2 and Corollary B.3 for the third equality, the symmetric initialization such that $f_{\theta_j^{(0)}}(x) = 0 \ \forall x$ in the fourth equality, and $a_i^{(0)^2} = 1$ for the last equality. Note that by symmetry of $a_i^{(0)}$ and $w_i^{(0)}$, the second term of the fifth equality is equivalent to 0.

*Remark* B.6. As mentioned in the main paper, the update of $\tilde{x}$ not well-defined for ReLU activation function in the case of gradient matching, and any other type of DD that include gradient information of $\tilde{x}$ in the distillation loss. We believe this is a fundamental problem that will be unavoidable in future theoretical work as well. Therefore, we have to substitute ReLU that comes from student gradients with an approximation $h$ whose second derivative is well-defined. Actually, in this paper, we show a somewhat stronger result. That is, for any $h$ whose second derivative is well defined, $h'$ and $h''$ are bounded, and $h'(t) > h'(-t)$ for all $t \in (0, 1)$, such as most continuous surrogates of ReLU such as softplus, DD (Algorithm 1) will output a set of synthetic points with mainly the same compression efficiency and generalization ability. Moreover, in Appendix C, we will treat a well-defined update for ReLU, which will also lead to similar result shown in this appendix.

$\square$

From this lemma, we can drastically simplify the notation as follows:

**Corollary B.7.** *$\tilde{x}^{(1)}$ can be regarded as taking $J^* = LJ/2$ draws $w_j \sim U(S^{d-1})$ $(j = 1, \ldots, J^*)$, and written as*

$$
\tilde{x}^{(1)} = - \eta_1^D \tilde{y}^{(0)} \left( G + \epsilon \right),
$$

*where*

$$G = \frac{1}{J^*}\sum_{j=1}^{J^*} g_j h'(\langle w_j, \tilde{x}^{(0)}\rangle) + \frac{1}{J^*}\sum_{j=1}^{J^*} h''(\langle w_j, \tilde{x}^{(0)}\rangle)\left\langle g_j, \tilde{x}^{(0)}\right\rangle$$

$$\epsilon = \frac{1}{J^*}\sum_{j=1}^{J^*}\left\{\frac{1}{N}\sum_n \epsilon_n x_n \sigma'(\langle w_j, x_n\rangle)\right\} h'(\langle w_j, \tilde{x}^{(0)}\rangle) + \frac{1}{J^*}\sum_{j=1}^{J^*} w_j h''(\langle w_j, \tilde{x}^{(0)}\rangle)\left\langle\frac{1}{N}\sum_n \epsilon_n x_n \sigma'(\langle w_j, x_n\rangle), \tilde{x}^{(0)}\right\rangle,$$

*with* $g_j = \frac{1}{N}\sum_n \hat{f}^*(x_n) x_n \sigma'(\langle w_j, x_n\rangle)$.

*Proof.* This follows from the symmetric assumption of $w_{i,j} = w_{L-i,j}$ (Assumption 3.4). □

*Remark* B.8. Note that we will only use this corollary to keep notation clean. We will never use this to blindly oversimplify our conclusion.

### B.3.3. FORMULATION OF POPULATION GRADIENT

In this section, we will focus on the population version of the empirical gradient $G$ derived in Corollary B.7 and analyze its properties. It can be written as follows:

**Definition B.9.** We define the population gradient of $G$ as

$$\hat{G} := \mathbb{E}_w\left[\mathbb{E}_x\left[\hat{f}^*(x)x\sigma'(\langle w, x\rangle)\right] h'(\langle w, \tilde{x}\rangle)\right] + \mathbb{E}_w\left[wh''(\langle w, \tilde{x}\rangle)\left\langle\mathbb{E}_x\left[\hat{f}^*(x)x\sigma'(\langle w, x\rangle)\right], \tilde{x}\right\rangle\right],$$

where $\tilde{x} := \tilde{x}^{(0)}$.

Let us first remind some properties from Damian et al. (2022).

**Definition B.10.** We define the Hermite expansion of $\hat{f}^*$ as

$$\hat{f}^*(x) = \sum_{k=0}^p \frac{\langle\hat{C}_k, \mathrm{He}_k(x)\rangle}{k!},$$

where $\hat{C}_k$ is a symmetric $k$-tensor defined as $\hat{C}_k := \mathbb{E}_{x\sim\mathcal{N}(0,I_d)}[\nabla_x^k \hat{f}^*(x)] \in (\mathbb{R}^d)^{\otimes k}$, and likewise,

$$f^*(x) = \sum_{k=0}^p \frac{\langle\bar{C}_k, \mathrm{He}_k(x)\rangle}{k!}, \text{ and } \sigma^*(z) = \sum_{k=0}^p \frac{C_k}{k!}\mathrm{He}_k(z),$$

where $\bar{C}_k := \mathbb{E}_{x\sim\mathcal{N}(0,I_d)}[\nabla_x^k f^*(x)] \in (\mathbb{R}^d)^{\otimes k}$ and $C_k := \mathbb{E}_{z\sim\mathcal{N}(0,I_r)}[\nabla_z^k \sigma^*(x)] \in (\mathbb{R}^r)^{\otimes k}$.

The following relation holds between $\hat{C}_k$, $\bar{C}_k$ and $C_k$.

**Lemma B.11.** *Under Assumption 3.1, $\hat{C}_k = \bar{C}_k = B^{\otimes k}C_k$ for $k \geq 2$, $\hat{C}_0 = \bar{C}_0 - \alpha$ and $\hat{C}_1 = \bar{C}_1 - \gamma$.*

*Proof.* Clearly $\hat{C}_k = \bar{C}_k$ for $k \geq 2$, $\hat{C}_0 = \bar{C}_0 - \alpha$ and $\hat{C}_1 = \bar{C}_1 - \gamma$ (see Definition B.1). Now, between $\bar{C}_k$ and $C_k$, $f^*(x) = \sigma^*(B^\top x)$ implies that

$$\nabla_x^k f(x) = \nabla_x^k \sigma^*(B^\top x) = B^{\otimes k}\nabla_z^k \sigma^*(B^\top x),$$

where we used Lemma A.14. Taking the expectation over $x \sim N(0, I_d)$ and since $B^\top B = I_r$,

$$\mathbb{E}_x[\nabla_x^k f(x)] = B^{\otimes k}\mathbb{E}_x[\nabla_z^k \sigma^*(B^\top x)] = B^{\otimes k}\mathbb{E}_z[\nabla_z^k \sigma^*(z)] = B^{\otimes k}C_k,$$

where we used that $z \sim \mathcal{N}(0, B^\top B) = \mathcal{N}(0, I_r)$. □

This leads to the following lemma.

**Lemma B.12.** *For the $\hat{C}_k$ defined as above and $k \geq 2$, under Assumption 3.1,*

$$\|\hat{C}_k\|_F^2 \leq k!, \text{ and } \|C_k\|_F^2 \leq k!$$

*Proof.* From Lemma 9 from Damian et al. (2022), $\|\bar{C}_k\|^2 \leq k!$. Therefore, $\|C_k\|_F^2 = \|B^{\otimes k} C_k\|_F^2 = \|\hat{C}_k\|_F^2 = \|\bar{C}_k\|_F^2 \leq k!$, where we used Lemma A.9 for the first equality and Lemma B.11 for the second and third equalities. □

Note that under Assumption 3.1, we also have $C_k\left(x^{\otimes k}\right) = C_k\left((\Pi^* x)^{\otimes k}\right)$.

Especially for $\hat{C}_0$ and $\hat{C}_1$, we know the following results:

**Lemma B.13** (Lemma 10 from Damian et al. (2022)). *Under Definition B.10 and Assumption 3.1, with high probability*

$$|\hat{C}_0| = \tilde{O}\left(\frac{1}{\sqrt{N}}\right), \quad and \quad \|\hat{C}_1\| = \tilde{O}\left(\sqrt{\frac{d}{N}}\right).$$

Now we can compute the population gradient of $\mathbb{E}_x\left[\hat{f}^*(x) x \sigma'(\langle w, x \rangle)\right]$ in closed form.

**Lemma B.14.** *Under Assumptions 3.1, with high probability,*

$$\mathbb{E}_x\left[\hat{f}^*(x) x \sigma'(\langle w, x \rangle)\right] = \sum_{k=1}^{p-1} \frac{c_{k+1} \hat{C}_{k+1}(w^{\otimes k})}{k!} + w \sum_{k=2}^{p} \frac{c_{k+2} \hat{C}_k(w^{\otimes k})}{k!} + \tilde{O}\left(\sqrt{\frac{d}{N}}\right).$$

*Proof.* The following equality immediately follows from the proof of Lemma 13 of Damian et al. (2022):

$$\mathbb{E}_x\left[\hat{f}^*(x) x \sigma'(\langle w, x \rangle)\right] = \left(\frac{C_1}{2} + \frac{w C_0}{\sqrt{2\pi}}\right) + \sum_{k=1}^{p-1} \frac{c_{k+1} \hat{C}_{k+1}(w^{\otimes k})}{k!} + w \sum_{k=2}^{p} \frac{c_{k+2} \hat{C}_k(w^{\otimes k})}{k!},$$

which leads to the statement by Lemma B.13,

□

**Corollary B.15.** *For single index models, with high probability,*

$$\hat{G} = \beta \sum_{k=1}^{p-1} \frac{c_{k+1} C_{k+1}}{k!} \mathbb{E}_w[\langle \beta, w \rangle^k h'(\langle w, \tilde{x} \rangle)] + \sum_{k=2}^{p} \frac{c_{k+2} C_k}{k!} \mathbb{E}_w[w \langle \beta, w \rangle^k h'(\langle w, \tilde{x} \rangle)]$$

$$+ \sum_{k=1}^{p-1} \frac{c_{k+1} C_{k+1}}{k!} \langle \beta, \tilde{x} \rangle \mathbb{E}_w[w \langle \beta, w \rangle^k h''(\langle w, \tilde{x} \rangle)] + \sum_{k=2}^{p} \frac{c_{k+2} C_k}{k!} \mathbb{E}_w[w \langle w, \tilde{x} \rangle \langle \beta, w \rangle^k h''(\langle w, \tilde{x} \rangle)]$$

$$+ \tilde{O}\left(\sqrt{\frac{d}{N}}\right).$$

*Proof.* By combining Lemma B.11 and Lemma B.14, we obtain

$$\mathbb{E}_x\left[\hat{f}^*(x) x \sigma'(\langle w, x \rangle)\right] = \beta \sum_{k=1}^{p-1} \frac{c_{k+1} C_{k+1} \langle \beta, w \rangle^k}{k!} + w \sum_{k=2}^{p} \frac{c_{k+2} C_k \langle \beta, w \rangle^k}{k!} + \tilde{O}\left(\sqrt{\frac{d}{N}}\right).$$

It suffices to substitute this to the definition of $\hat{G}$ (Definition B.9) to obtain the desired result. □

Let us now estimate each expectation present in the formulation of $\hat{G}$ as shown in the above corollary.

**Lemma B.16.** *When the activation function is a $C^2$ class function $h$ with $\sup_{|z| \leq 1} |h'(z)| \leq M_1$ and $\sup_{|z| \leq 1} |h''(z)| \leq M_2$, and $\|\beta\| = 1$, $\|\tilde{x}\| = 1$, then for $k \geq 0$,*

$$\mathbb{E}_w[w \langle \beta, w \rangle^k h'(\langle w, \tilde{x} \rangle)] = A_k(d) \tilde{x} + B_k(d) \beta_\perp,$$
$$\mathbb{E}_w[\langle \beta, w \rangle^k h'(\langle w, \tilde{x} \rangle)] = D_k(d),$$
$$\mathbb{E}_w[w \langle \beta, w \rangle^k h''(\langle w, \tilde{x} \rangle)] = E_k(d) \tilde{x} + F_k(d) \beta_\perp,$$
$$\mathbb{E}_w[w \langle w, \tilde{x} \rangle \langle \beta, w \rangle^k h''(\langle w, \tilde{x} \rangle)] = G_k(d) \tilde{x} + H_k(d) \beta_\perp,$$

where $\beta_\perp = \beta - \langle \beta, \tilde{x} \rangle \tilde{x}$, $A_k(d) = O(M_1 d^{-\frac{k+1}{2}})$, $B_k(d) = O(M_1 d^{-\frac{k+1}{2}})$, $D_k(d) = O(M_1 d^{-\frac{k}{2}})$, $E_k(d) = O(M_2 d^{-\frac{k+1}{2}})$, $F_k(d) = O(M_2 d^{-\frac{k+1}{2}})$, $G_k(d) = O(M_2 d^{-\frac{k+2}{2}})$, and $H_k(d) = O(M_2 d^{-\frac{k+2}{2}})$.

*Proof.* Let us first prove the first equality. Let $t = \langle w, \tilde{x} \rangle$, then we can write $w = t\tilde{x} + \sqrt{1-t^2}v$ where $v \sim U(\{v \mid v \in S^{d-1}, \langle v, \tilde{x} \rangle = 0\}) \cong U(S^{d-2})$. Since $\|\tilde{x}\|^2 = 1$, the distribution of $t$ is equivalent to that of the first coordinate $w_1$ which is

$$f_d(t) = \frac{\Gamma(\frac{d}{2})}{\sqrt{\pi}\Gamma(\frac{d-1}{2})}(1-t^2)^{\frac{d-3}{2}}.$$

Moreover,

$$\langle \beta, w \rangle = st + \sqrt{1-t^2}\langle \beta_\perp, v \rangle,$$

where $s = \langle \beta, \tilde{x} \rangle$, and $\beta_\perp = \beta - \langle \beta, \tilde{x} \rangle \tilde{x}$. Now,

$$\mathbb{E}_w[w\langle \beta, w \rangle^k h'(\langle w, \tilde{x} \rangle)]$$

$$= \mathbb{E}_{t,v}\left[\left(t\tilde{x} + \sqrt{1-t^2}v\right)\left(st + \|\beta_\perp\|\sqrt{1-t^2}\langle \beta_\perp/\|\beta_\perp\|, v \rangle\right)^k h'(t)\right]$$

$$= \frac{\Gamma(\frac{d}{2})}{\sqrt{\pi}\Gamma(\frac{d-1}{2})}\int_{-1}^{1} h'(t)(1-t^2)^{\frac{d-3}{2}} E_v\left[\left(t\tilde{x} + \sqrt{1-t^2}v\right)\left(st + \|\beta_\perp\|\sqrt{1-t^2}\langle \beta_\perp/\|\beta_\perp\|, v \rangle\right)^k\right]dt.$$

Since

$$\left(st + \|\beta_\perp\|\sqrt{1-t^2}\langle \beta_\perp/\|\beta_\perp\|, v \rangle\right)^k = \sum_{i=0}^{k}\binom{k}{i}(st)^{k-i}(\|\beta_\perp\|\sqrt{1-t^2})^i\langle \beta_\perp/\|\beta_\perp\|, v \rangle^i,$$

we can further develop the expectation as follows:

$$\mathbb{E}_w[w\langle \beta, w \rangle^k h'(\langle w, \tilde{x} \rangle)]$$

$$= \frac{\Gamma(\frac{d}{2})}{\sqrt{\pi}\Gamma(\frac{d-1}{2})}\int_{-1}^{1} h'(t)(1-t^2)^{\frac{d-3}{2}} \mathbb{E}_v\left[\left(t\tilde{x} + \sqrt{1-t^2}v\right)\left(st + \|\beta_\perp\|\sqrt{1-t^2}\langle \beta_\perp/\|\beta_\perp\|, v \rangle\right)^k\right]dt$$

$$= \frac{\Gamma(\frac{d}{2})}{\sqrt{\pi}\Gamma(\frac{d-1}{2})}\int_{-1}^{1} h'(t)(1-t^2)^{\frac{d-3}{2}}\left\{t\sum_{i=0}^{k}\binom{k}{i}(st)^{k-i}(\|\beta_\perp\|\sqrt{1-t^2})^i\mathbb{E}_v\left[\langle \beta_\perp/\|\beta_\perp\|, v \rangle^i\right]\tilde{x}\right.$$

$$\left. + \sqrt{1-t^2}\sum_{i=0}^{k}\binom{k}{i}(st)^{k-i}(\|\beta_\perp\|\sqrt{1-t^2})^i\mathbb{E}_v\left[\langle \beta_\perp/\|\beta_\perp\|, v \rangle^i v\right]\right\}dt.$$

By rotation symmetry, $\mathbb{E}_v\left[\langle \beta_\perp/\|\beta_\perp\|, v \rangle^i\right] = \mathbb{E}_{z \sim U(S^{d-2})}[z_1^i] = c_{\frac{i}{2}}(d)$ if $i$ is even and 0 if $i$ is odd. Likewise, $\mathbb{E}_v\left[\langle \beta_\perp/\|\beta_\perp\|, v \rangle^i v\right] = \mathbb{E}_{z \sim U(S^{d-2})}[z_1^{i+1}]\beta_\perp/\|\beta_\perp\| = c_{\frac{i+1}{2}}(d)\beta_\perp/\|\beta_\perp\|$ if $i$ is odd and 0 if $i$ is even. Here, we used the definition of $c_k(d)$ defined in Lemma B.24.

Therefore,

$$\mathbb{E}_w[w\langle\beta,w\rangle^k h'(\langle w,\tilde{x}\rangle)]$$

$$=\frac{\Gamma(\frac{d}{2})}{\sqrt{\pi}\Gamma(\frac{d-1}{2})}\int_{-1}^1 h'(t)(1-t^2)^{\frac{d-3}{2}}\left\{t\sum_{i=0}^{\lfloor\frac{k}{2}\rfloor}\left(\begin{array}{c}k\\2i\end{array}\right)(st)^{k-2i}(\|\beta_\perp\|\sqrt{1-t^2})^{2i}c_i(d)\tilde{x}\right.$$

$$\left.+\sqrt{1-t^2}\sum_{i=0}^{\lfloor\frac{k-1}{2}\rfloor}\left(\begin{array}{c}k\\2i+1\end{array}\right)(st)^{k-(2i+1)}(\|\beta_\perp\|\sqrt{1-t^2})^{2i+1}c_{i+1}(d)\beta_\perp/\|\beta_\perp\|\right\}\mathrm{d}t$$

$$=\frac{\Gamma(\frac{d}{2})}{\sqrt{\pi}\Gamma(\frac{d-1}{2})}\sum_{i=0}^{\lfloor\frac{k}{2}\rfloor}\left(\begin{array}{c}k\\2i\end{array}\right)s^{k-2i}\|\beta_\perp\|^{2i}c_i(d)\int_{-1}^1 h'(t)(1-t^2)^{\frac{d-3}{2}+i}t^{k-2i+1}\mathrm{d}t\cdot\tilde{x}$$

$$+\frac{\Gamma(\frac{d}{2})}{\sqrt{\pi}\Gamma(\frac{d-1}{2})}\sum_{i=0}^{\lfloor\frac{k-1}{2}\rfloor}\left(\begin{array}{c}k\\2i+1\end{array}\right)s^{k-(2i+1)}\|\beta_\perp\|^{2i}c_{i+1}(d)\int_{-1}^1 h'(t)(1-t^2)^{\frac{d-3}{2}+i+1}t^{k-(2i+1)}\mathrm{d}t\cdot\beta_\perp$$

$$=A_k(d)\tilde{x}+B_k(d)\beta_\perp,$$

where we defined

$$A_k(d):=\frac{\Gamma(\frac{d}{2})}{\sqrt{\pi}\Gamma(\frac{d-1}{2})}\sum_{i=0}^{\lfloor\frac{k}{2}\rfloor}\left(\begin{array}{c}k\\2i\end{array}\right)s^{k-2i}\|\beta_\perp\|^{2i}c_i(d)\int_{-1}^1 h'(t)(1-t^2)^{\frac{d-3}{2}+i}t^{k-2i+1}\mathrm{d}t,$$

and

$$B_k(d):=\frac{\Gamma(\frac{d}{2})}{\sqrt{\pi}\Gamma(\frac{d-1}{2})}\sum_{i=0}^{\lfloor\frac{k-1}{2}\rfloor}\left(\begin{array}{c}k\\2i+1\end{array}\right)s^{k-(2i+1)}\|\beta_\perp\|^{2i}c_{i+1}(d)\int_{-1}^1 h'(t)(1-t^2)^{\frac{d-3}{2}+i+1}t^{k-(2i+1)}\mathrm{d}t.$$

Using Lemma B.23, we obtain

$$|A_k(d)|\leq\left|\frac{\Gamma(\frac{d}{2})}{\sqrt{\pi}\Gamma(\frac{d-1}{2})}\sum_{i=0}^{\lfloor\frac{k}{2}\rfloor}\left(\begin{array}{c}k\\2i\end{array}\right)s^{k-2i}\|\beta_\perp\|^{2i}c_i(d)\int_{-1}^1 h'(t)(1-t^2)^{\frac{d-3}{2}+i}t^{k-2i+1}\mathrm{d}t\right|$$

$$\leq\frac{\Gamma(\frac{d}{2})}{\sqrt{\pi}\Gamma(\frac{d-1}{2})}\sum_{i=0}^{\lfloor\frac{k}{2}\rfloor}\left(\begin{array}{c}k\\2i\end{array}\right)|s|^{k-2i}\|\beta_\perp\|^{2i}c_i(d)\left|\int_{-1}^1 h'(t)(1-t^2)^{\frac{d-3}{2}+i}t^{k-2i+1}\mathrm{d}t\right|$$

$$\leq 2M_1\frac{\Gamma(\frac{d}{2})}{\sqrt{\pi}\Gamma(\frac{d-1}{2})}\sum_{i=0}^{\lfloor\frac{k}{2}\rfloor}\left(\begin{array}{c}k\\2i\end{array}\right)|s|^{k-2i}\|\beta_\perp\|^{2i}c_i(d)\int_0^1(1-t^2)^{\frac{d-3}{2}+i}t^{k-2i+1}\mathrm{d}t$$

$$=2M_1\frac{\Gamma(\frac{d}{2})}{\sqrt{\pi}\Gamma(\frac{d-1}{2})}\sum_{i=0}^{\lfloor\frac{k}{2}\rfloor}\left(\begin{array}{c}k\\2i\end{array}\right)|s|^{k-2i}\|\beta_\perp\|^{2i}c_i(d)\frac{1}{2}\mathrm{Beta}\left(\frac{k-2i+2}{2},\frac{d-3}{2}+i+1\right)$$

$$\lesssim M_1\frac{\Gamma(\frac{d}{2})}{\Gamma(\frac{d-1}{2})}\sum_{i=0}^{\lfloor\frac{k}{2}\rfloor}c_i(d)\mathrm{Beta}\left(\frac{k-2i+2}{2},\frac{d-3}{2}+i+1\right)$$

$$\lesssim M_1 d^{1/2}\sum_{i=0}^{\lfloor\frac{k}{2}\rfloor}d^{-i}\left(\frac{d-3}{2}+i+1\right)^{-\frac{k-2i+2}{2}}$$

$$\lesssim M_1 d^{1/2}\sum_{i=0}^{\lfloor\frac{k}{2}\rfloor}d^{-i}d^{-\frac{k+2}{2}+i}$$

$$\lesssim M_1 d^{-\frac{k+1}{2}},$$

where we used the definition of the Beta function

$$\text{Beta}(p_1, p_2) := \int_{-1}^{1} t^{p_1-1}(1-t)^{p_2-1}\mathrm{d}t = 2\int_0^1 t^{2p_1-1}(1-t^2)^{p_2-1}\mathrm{d}t, \; p_1, \; p_2 > 0,$$

in the equality, and Stirling's approximation when the second argument $p_2$ is large in the fifth inequality.

Likewise, for $B_k(d)$, we obtain

$$|B_k(d)| \leq 2M_1 \frac{\Gamma(\frac{d}{2})}{\sqrt{\pi}\Gamma(\frac{d-1}{2})} \sum_{i=0}^{\lfloor \frac{k-1}{2} \rfloor} \binom{k}{2i+1} |s|^{k-(2i+1)}\|\beta_\perp\|^{2i}c_{i+1}(d) \int_0^1 (1-t^2)^{\frac{d-3}{2}+i+1}t^{k-(2i+1)}\mathrm{d}t$$

$$=2M_1 \frac{\Gamma(\frac{d}{2})}{\sqrt{\pi}\Gamma(\frac{d-1}{2})} \sum_{i=0}^{\lfloor \frac{k-1}{2} \rfloor} \binom{k}{2i+1} |s|^{k-(2i+1)}\|\beta_\perp\|^{2i}c_{i+1}(d)\frac{1}{2}\text{Beta}\left(\frac{k-2i}{2}, \frac{d-3}{2}+i+2\right)$$

$$\lesssim M_1 \frac{\Gamma(\frac{d}{2})}{\Gamma(\frac{d-1}{2})} \sum_{i=0}^{\lfloor \frac{k-1}{2} \rfloor} c_{i+1}(d)\text{Beta}\left(\frac{k-2i}{2}, \frac{d-3}{2}+i+2\right)$$

$$\lesssim M_1 d^{\frac{1}{2}} \sum_{i=0}^{\lfloor \frac{k-1}{2} \rfloor} d^{-(i+1)}\left(\frac{d-3}{2}+i+2\right)^{-\frac{k-2i}{2}}$$

$$\lesssim M_1 d^{\frac{1}{2}} \sum_{i=0}^{\lfloor \frac{k-1}{2} \rfloor} d^{-(i+1)}d^{-\frac{k}{2}+i}$$

$$\lesssim M_1 d^{-\frac{k+1}{2}}.$$

We can proof analogously for $\mathbb{E}_w[w\langle\beta, w\rangle^k h''(\langle w, \tilde{x}\rangle)]$ and obtain

$$\mathbb{E}_w[w\langle\beta, w\rangle^k h''(\langle w, \tilde{x}\rangle)] = E_k(d)\tilde{x} + F_k(d)\beta_\perp$$

such that $E_k(d) = O(M_2 d^{-\frac{k+1}{2}})$, and $F_k(d) = O(M_2 d^{-\frac{k+1}{2}})$. Next, we similarly prove the second equality of the statement. By the same change of variable,

$$\mathbb{E}_w[\langle\beta, w\rangle^k h'(\langle w, \tilde{x}\rangle)] = \frac{\Gamma(\frac{d}{2})}{\sqrt{\pi}\Gamma(\frac{d-1}{2})} \int_{-1}^{1} h'(t)(1-t^2)^{\frac{d-3}{2}}\mathbb{E}_v\left[\left(st + \|\beta_\perp\|\sqrt{1-t^2}\langle\beta_\perp/\|\beta_\perp\|, v\rangle\right)^k\right]\mathrm{d}t$$

$$= \frac{\Gamma(\frac{d}{2})}{\sqrt{\pi}\Gamma(\frac{d-1}{2})} \int_{-1}^{1} h'(t)(1-t^2)^{\frac{d-3}{2}} \sum_{i=0}^{k} \binom{k}{i} (st)^{k-i}(\|\beta_\perp\|\sqrt{1-t^2})^i \mathbb{E}_v\left[\langle\beta_\perp/\|\beta_\perp\|, v\rangle^i\right]\mathrm{d}t$$

$$= \frac{\Gamma(\frac{d}{2})}{\sqrt{\pi}\Gamma(\frac{d-1}{2})} \int_{-1}^{1} h'(t)(1-t^2)^{\frac{d-3}{2}} \sum_{i=0}^{\lfloor \frac{k}{2} \rfloor} \binom{k}{2i} (st)^{k-2i}(\|\beta_\perp\|\sqrt{1-t^2})^{2i}c_i(d)\mathrm{d}t$$

$$= \frac{\Gamma(\frac{d}{2})}{\sqrt{\pi}\Gamma(\frac{d-1}{2})} \sum_{i=0}^{\lfloor \frac{k}{2} \rfloor} \binom{k}{2i} s^{k-2i}\|\beta_\perp\|^{2i}c_i(d) \int_{-1}^{1} h'(t)(1-t^2)^{\frac{d-3}{2}+i}t^{k-2i}\mathrm{d}t$$

$$\leq \left|\frac{\Gamma(\frac{d}{2})}{\sqrt{\pi}\Gamma(\frac{d-1}{2})} \sum_{i=0}^{\lfloor \frac{k}{2} \rfloor} \binom{k}{2i} s^{k-2i}\|\beta_\perp\|^{2i}c_i(d) \int_{-1}^{1} h'(t)(1-t^2)^{\frac{d-3}{2}+i}t^{k-2i}\mathrm{d}t\right|$$

$$\leq 2M_1 \frac{\Gamma(\frac{d}{2})}{\sqrt{\pi}\Gamma(\frac{d-1}{2})} \sum_{i=0}^{\lfloor \frac{k}{2} \rfloor} \binom{k}{2i} |s|^{k-2i}\|\beta_\perp\|^{2i}c_i(d) \int_0^1 (1-t^2)^{\frac{d-3}{2}+i}t^{k-2i}\mathrm{d}t$$

$$= 2M_1 \frac{\Gamma(\frac{d}{2})}{\sqrt{\pi}\Gamma(\frac{d-1}{2})} \sum_{i=0}^{\lfloor \frac{k}{2} \rfloor} \binom{k}{2i} |s|^{k-2i}\|\beta_\perp\|^{2i}c_i(d)\frac{1}{2}\text{Beta}\left(\frac{k-2i+1}{2}, \frac{d-3}{2}+i+1\right).$$

Therefore,

$$D_k(d) := \mathbb{E}_w[\langle\beta,w\rangle^k h'(\langle w,\tilde{x}\rangle)] \lesssim M_1 \frac{\Gamma(\frac{d}{2})}{\Gamma(\frac{d-1}{2})} \sum_{i=0}^{\lfloor\frac{k}{2}\rfloor} c_i(d)\text{Beta}\left(\frac{k-2i+1}{2}, \frac{d-3}{2}+i+1\right)$$

$$\lesssim M_1 d^{-\frac{k}{2}}.$$

Finally, let us focus on $\mathbb{E}_w[w\langle w,\tilde{x}\rangle\langle\beta,w\rangle^k h''(\langle w,\tilde{x}\rangle)]$. Following the same approach, we can reformulate this expectation as

$$
\begin{aligned}
&\mathbb{E}_w[w\langle\beta,w\rangle^k\langle w,\tilde{x}\rangle h''(\langle w,\tilde{x}\rangle)]\\
&=\mathbb{E}_{t,v}\left[\left(t\tilde{x}+\sqrt{1-t^2}v\right)\left(st+\|\beta_\perp\|\sqrt{1-t^2}\langle\beta_\perp/\|\beta_\perp\|,v\rangle\right)^k th''(t)\right]\\
&=\frac{\Gamma(\frac{d}{2})}{\sqrt{\pi}\Gamma(\frac{d-1}{2})}\int_{-1}^{1} th''(t)(1-t^2)^{\frac{d-3}{2}}E_v\left[\left(t\tilde{x}+\sqrt{1-t^2}v\right)\left(st+\|\beta_\perp\|\sqrt{1-t^2}\langle\beta_\perp/\|\beta_\perp\|,v\rangle\right)^k\right]dt\\
&=\frac{\Gamma(\frac{d}{2})}{\sqrt{\pi}\Gamma(\frac{d-1}{2})}\int_{-1}^{1} th''(t)(1-t^2)^{\frac{d-3}{2}}\left\{t\sum_{i=0}^{k}\binom{k}{i}(st)^{k-i}(\|\beta_\perp\|\sqrt{1-t^2})^i\mathbb{E}_v\left[\langle\beta_\perp/\|\beta_\perp\|,v\rangle^i\right]\tilde{x}\right.\\
&\quad\left.+\sqrt{1-t^2}\sum_{i=0}^{k}\binom{k}{i}(st)^{k-i}(\|\beta_\perp\|\sqrt{1-t^2})^i\mathbb{E}_v\left[\langle\beta_\perp/\|\beta_\perp\|,v\rangle^i v\right]\right\}dt\\
&=\frac{\Gamma(\frac{d}{2})}{\sqrt{\pi}\Gamma(\frac{d-1}{2})}\sum_{i=0}^{\lfloor\frac{k}{2}\rfloor}\binom{k}{2i}s^{k-2i}\|\beta_\perp\|^{2i}c_i(d)\int_{-1}^{1} h''(t)(1-t^2)^{\frac{d-3}{2}+i}t^{k-2i+2}dt\cdot\tilde{x}\\
&\quad+\frac{\Gamma(\frac{d}{2})}{\sqrt{\pi}\Gamma(\frac{d-1}{2})}\sum_{i=0}^{\lfloor\frac{k-1}{2}\rfloor}\binom{k}{2i+1}s^{k-(2i+1)}\|\beta_\perp\|^{2i}c_{i+1}(d)\int_{-1}^{1} h'(t)(1-t^2)^{\frac{d-3}{2}+i+1}t^{k-(2i+1)+1}dt\cdot\beta_\perp\\
&=G_k(d)\tilde{x}+H_k(d)\beta_\perp,
\end{aligned}
$$

where we defined

$$G_k(d) := \frac{\Gamma(\frac{d}{2})}{\sqrt{\pi}\Gamma(\frac{d-1}{2})}\sum_{i=0}^{\lfloor\frac{k}{2}\rfloor}\binom{k}{2i}s^{k-2i}\|\beta_\perp\|^{2i}c_i(d)\int_{-1}^{1} h'(t)(1-t^2)^{\frac{d-3}{2}+i}t^{k-2i+2}dt,$$

and

$$H_k(d) := \frac{\Gamma(\frac{d}{2})}{\sqrt{\pi}\Gamma(\frac{d-1}{2})}\sum_{i=0}^{\lfloor\frac{k-1}{2}\rfloor}\binom{k}{2i+1}s^{k-(2i+1)}\|\beta_\perp\|^{2i}c_{i+1}(d)\int_{-1}^{1} h'(t)(1-t^2)^{\frac{d-3}{2}+i+1}t^{k-(2i+1)+1}dt.$$

Therefore,

$$
\begin{aligned}
|G_k(d)| \leq & \frac{\Gamma(\frac{d}{2})}{\sqrt{\pi}\Gamma(\frac{d-1}{2})} \sum_{i=0}^{\lfloor \frac{k}{2} \rfloor} \binom{k}{2i} |s|^{k-2i} \|\beta_\perp\|^{2i} c_i(d) \left| \int_{-1}^{1} h'(t)(1-t^2)^{\frac{d-3}{2}+i} t^{k-2i+2} \mathrm{d}t \right| \\
\leq & 2M_2 \frac{\Gamma(\frac{d}{2})}{\sqrt{\pi}\Gamma(\frac{d-1}{2})} \sum_{i=0}^{\lfloor \frac{k}{2} \rfloor} \binom{k}{2i} |s|^{k-2i} \|\beta_\perp\|^{2i} c_i(d) \int_{0}^{1} (1-t^2)^{\frac{d-3}{2}+i} t^{k-2i+2} \mathrm{d}t \\
\leq & 2M_2 \frac{\Gamma(\frac{d}{2})}{\sqrt{\pi}\Gamma(\frac{d-1}{2})} \sum_{i=0}^{\lfloor \frac{k}{2} \rfloor} \binom{k}{2i} |s|^{k-2i} \|\beta_\perp\|^{2i} c_i(d) \frac{1}{2} \mathrm{Beta}\left( \frac{k-2i+3}{2}, \frac{d-3}{2}+i+1 \right) \\
\lesssim & M_2 \frac{\Gamma(\frac{d}{2})}{\Gamma(\frac{d-1}{2})} \sum_{i=0}^{\lfloor \frac{k}{2} \rfloor} c_i(d) \mathrm{Beta}\left( \frac{k-2i+3}{2}, \frac{d-3}{2}+i+1 \right) \\
\lesssim & M_2 d^{1/2} \sum_{i=0}^{\lfloor \frac{k}{2} \rfloor} d^{-i} \left( \frac{d-3}{2}+i+1 \right)^{-\frac{k-2i+3}{2}} \\
\lesssim & M_2 d^{1/2} \sum_{i=0}^{\lfloor \frac{k}{2} \rfloor} d^{-i} d^{-\frac{k+3}{2}+i} \\
\lesssim & M_2 d^{-\frac{k+2}{2}},
\end{aligned}
$$

and

$$
\begin{aligned}
|H_k(d)| \leq & 2M_2 \frac{\Gamma(\frac{d}{2})}{\sqrt{\pi}\Gamma(\frac{d-1}{2})} \sum_{i=0}^{\lfloor \frac{k-1}{2} \rfloor} \binom{k}{2i+1} |s|^{k-(2i+1)} \|\beta_\perp\|^{2i} c_{i+1}(d) \int_{0}^{1} (1-t^2)^{\frac{d-3}{2}+i+1} t^{k-(2i+1)+1} \mathrm{d}t \\
= & 2M_2 \frac{\Gamma(\frac{d}{2})}{\sqrt{\pi}\Gamma(\frac{d-1}{2})} \sum_{i=0}^{\lfloor \frac{k-1}{2} \rfloor} \binom{k}{2i+1} |s|^{k-(2i+1)} \|\beta_\perp\|^{2i} c_{i+1}(d) \frac{1}{2} \mathrm{Beta}\left( \frac{k-2i+1}{2}, \frac{d-3}{2}+i+2 \right) \\
\lesssim & M_2 \frac{\Gamma(\frac{d}{2})}{\Gamma(\frac{d-1}{2})} \sum_{i=0}^{\lfloor \frac{k-1}{2} \rfloor} c_{i+1}(d) \mathrm{Beta}\left( \frac{k-2i+1}{2}, \frac{d-3}{2}+i+2 \right) \\
\lesssim & M_2 d^{\frac{1}{2}} \sum_{i=0}^{\lfloor \frac{k-1}{2} \rfloor} d^{-(i+1)} \left( \frac{d-3}{2}+i+2 \right)^{-\frac{k-2i+1}{2}} \\
\lesssim & M_2 d^{\frac{1}{2}} \sum_{i=0}^{\lfloor \frac{k-1}{2} \rfloor} d^{-(i+1)} d^{-\frac{k+1}{2}+i} \\
\lesssim & M_2 d^{-\frac{k+2}{2}}.
\end{aligned}
$$

$\square$

We provide tighter bounds for a few lower order terms.

**Corollary B.17.** *If $\tilde{x} \sim U(S^{d-1})$ and $h'$ and $h''$ are non-zero continuous in $[-1,1]$, then with high probability,*

$$
D_1(d) + D_3(d) + A_2(d)\tilde{x} + B_2(d)\beta_\perp + \langle \beta, \tilde{x} \rangle (E_1(d)\beta + F_1(d)\beta_\perp) = c_d \langle \beta, \tilde{x} \rangle \beta + \tilde{O}(d^{-2}),
$$

*where $c_d = \Theta(d^{-1})$.*

*Proof.* Please refer to Corollary D.7, which proves the statement for the general multi-index case. $\square$

Combining this with Corollary B.15, we obtain the following result.

**Theorem B.18.** *Under Assumptions 3.1, 3.3, 3.4, 3.6, and 3.8, with high probability,*

$$\hat{G} = c_d\langle\beta, \tilde{x}\rangle\beta + \tilde{O}\left(d^{\frac{1}{2}}N^{-\frac{1}{2}} + d^{-2}\right),$$

*where $c_d = \tilde{\Theta}\left(d^{-1}\right)$.*

*Proof.*

$$\hat{G} = \beta \sum_{k=1}^{p-1} \frac{c_{k+1}C_{k+1}}{k!} \mathbb{E}_w[\langle\beta, w\rangle^k h'(\langle w, \tilde{x}\rangle)] + \sum_{k=2}^{p} \frac{c_{k+2}C_k}{k!} \mathbb{E}_w[w\langle\beta, w\rangle^k h'(\langle w, \tilde{x}\rangle)]$$

$$+ \sum_{k=1}^{p-1} \frac{c_{k+1}C_{k+1}}{k!}\langle\beta, \tilde{x}\rangle\mathbb{E}_w[w\langle\beta, w\rangle^k h''(\langle w, \tilde{x}\rangle)] + \sum_{k=2}^{p} \frac{c_{k+2}C_k}{k!}\mathbb{E}_w[w\langle w, \tilde{x}\rangle\langle\beta, w\rangle^k h''(\langle w, \tilde{x}\rangle)]$$

$$+ \tilde{O}\left(\sqrt{\frac{d}{N}}\right)$$

$$= \beta \sum_{k=1}^{p-1} \frac{c_{k+1}C_{k+1}}{k!} D_k(d) + \sum_{k=2}^{p} \frac{c_{k+2}C_k}{k!}(A_k(d)\tilde{x} + B_k(d)\beta_\perp)$$

$$+ \sum_{k=1}^{p-1} \frac{c_{k+1}C_{k+1}}{k!}\langle\beta, \tilde{x}\rangle(E_k(d)\tilde{x} + F_k(d)\beta_\perp) + \sum_{k=2}^{p} \frac{c_{k+2}C_k}{k!}(G_k(d)\tilde{x} + H_k(d)\beta_\perp)$$

$$+ \tilde{O}\left(\sqrt{\frac{d}{N}}\right)$$

$$= \beta\frac{c_2C_2}{1!}D_1(d) + \beta\frac{c_4C_4}{3!}D_3(d) + \beta\sum_{k=4}^{p-1} \frac{c_{k+1}C_{k+1}}{k!}D_k(d)$$

$$+ \frac{c_4C_2}{2!}(A_2(d)\tilde{x} + B_2(d)\beta_\perp) + \sum_{k=3}^{p} \frac{c_{k+2}C_k}{k!}(A_k(d)\tilde{x} + B_k(d)\beta_\perp)$$

$$+ \frac{c_2C_2}{1!}\langle\beta, \tilde{x}\rangle(E_1(d)\tilde{x} + F_1(d)\beta_\perp) + \sum_{k=3}^{p-1} \frac{c_{k+1}C_{k+1}}{k!}\langle\beta, \tilde{x}\rangle(E_k(d)\tilde{x} + F_k(d)\beta_\perp)$$

$$+ \sum_{k=2}^{p} \frac{c_{k+2}C_k}{k!}(G_k(d)\tilde{x} + H_k(d)\beta_\perp)$$

$$+ \tilde{O}\left(\sqrt{\frac{d}{N}}\right)$$

$$= c_d\langle\beta, \tilde{x}\rangle\beta + \tilde{O}\left(d^{-2}\right) + \tilde{O}\left(\sqrt{\frac{d}{N}}\right),$$

where we used Corollary B.17 for the last equality, where $c_d = \tilde{\Theta}(d^{-1})$.

$\square$

### B.3.4. CONCENTRATION OF EMPIRICAL GRADIENT

Now that we have computed the population gradient, we evaluate the concentration of the empirical gradient around the population gradient.

**Theorem B.19.** *Under Assumptions 3.1, 3.3, 3.4, 3.6, 3.7 and 3.8, with high probability,*

$$G = c_d\langle\beta, \tilde{x}\rangle\beta + \tilde{O}\left(d^{-\frac{1}{2}}N^{-\frac{1}{2}} + d^{-2} + d^{-\frac{1}{2}}J^{*-\frac{1}{2}}\right).$$

*Proof.* Since
$$G = \hat{G} + G - \hat{G} \leq \hat{G} + \sup_{\tilde{x} \in S^{d-1}} \|G - \hat{G}\|,$$

and the first term is already evaluated in Theorem B.18, we will evaluate the second. From the form of $G$ and $\hat{G}$, we can divide it into the following two terms:

$$\Delta_1 := \sup_{\tilde{x}} \left\| \frac{1}{J^*} \sum_j g_j h'(\langle w_j, \tilde{x} \rangle) - \mathbb{E}_w \left[ \mathbb{E}_x \left[ \hat{f}^*(x) x \sigma'(\langle w, x \rangle) \right] h'(\langle w, \tilde{x} \rangle) \right] \right\|,$$

$$\Delta_2 := \sup_{\tilde{x}} \left\| \frac{1}{J^*} \sum_j w_j h''(\langle w_j, \tilde{x} \rangle) \langle g_j, \tilde{x} \rangle - \mathbb{E}_w \left[ w h''(\langle w, \tilde{x} \rangle) \left\langle \mathbb{E}_x \left[ \hat{f}^*(x) x \sigma'(\langle w, x \rangle) \right], \tilde{x} \right\rangle \right] \right\|,$$

such that $\sup_{\tilde{x}} \|G - \hat{G}\| \leq \Delta_1 + \Delta_2$, and $g_j = \frac{1}{N} \sum_n \hat{f}^*(x_n) \sigma'(\langle w_j, x_n \rangle)$.

Let us first evaluate $\Delta_1$.

$$\Delta_1 = \sup_{\tilde{x} \in S^{d-1}} \left\| \frac{1}{J^* N} \sum_{j,n} \hat{f}^*(x_n) x_n \sigma'(\langle w_j, x_n \rangle) h'(\langle w_j, \tilde{x} \rangle) - \mathbb{E}_{w,x} \left[ \hat{f}^*(x) x \sigma'(\langle w, x \rangle) h'(\langle w, \tilde{x} \rangle) \right] \right\|$$

$$= \sup_{\tilde{x} \in S^{d-1}} \left\| \frac{1}{J^* N} \sum_{j,n} \hat{f}^*(x_n) x_n \sigma'(\langle w_j, x_n \rangle) h'(\langle w_j, \tilde{x} \rangle) - \frac{1}{J^*} \sum_j \mathbb{E}_x \left[ \hat{f}^*(x) x \sigma'(\langle w_j, x \rangle) h'(\langle w_j, \tilde{x} \rangle) \right] \right\|$$

$$+ \sup_{\tilde{x} \in S^{d-1}} \left\| \frac{1}{J^*} \sum_j \mathbb{E}_x \left[ \hat{f}^*(x) x \sigma'(\langle w_j, x \rangle) h'(\langle w_j, \tilde{x} \rangle) \right] - \mathbb{E}_{w,x} \left[ \hat{f}^*(x) x \sigma'(\langle w, x \rangle) h'(\langle w, \tilde{x} \rangle) \right] \right\|.$$

We define the first term of the right hand side as $\Delta_{1,1}$ and the second as $\Delta_{1,2}$. As for $\Delta_{1,1}$,

$$\Delta_{1,1} = \sup_{\tilde{x} \in S^{d-1}} \left\| \frac{1}{J^*} \sum_j \left\{ \frac{1}{N} \sum_n \hat{f}^*(x_n) x_n \sigma'(\langle w_j, x_n \rangle) h'(\langle w_j, \tilde{x} \rangle) - \mathbb{E}_x \left[ \hat{f}^*(x) x \sigma'(\langle w_j, x \rangle) h'(\langle w_j, \tilde{x} \rangle) \right] \right\} \right\|$$

$$\leq \sup_{\tilde{x} \in S^{d-1}} \left\| \sup_{w \in S^{d-1}} \left\| \frac{1}{N} \sum_n \hat{f}^*(x_n) x_n \sigma'(\langle w_j, x_n \rangle) h'(\langle w_j, \tilde{x} \rangle) - \mathbb{E}_x \left[ \hat{f}^*(x) x \sigma'(\langle w_j, x \rangle) h'(\langle w_j, \tilde{x} \rangle) \right] \right\| \right\|$$

$$= \sup_{\tilde{x} \in S^{d-1}} \left\| \sup_{w \in S^{d-1}} \left\| \left\{ \frac{1}{N} \sum_n \hat{f}^*(x_n) x_n \sigma'(\langle w_j, x_n \rangle) - \mathbb{E}_x \left[ \hat{f}^*(x) x \sigma'(\langle w_j, x \rangle) \right] \right\} h'(\langle w_j, \tilde{x} \rangle) \right\| \right\|$$

$$\leq \sup_{\tilde{x} \in S^{d-1}} \left\| \sup_{w \in S^{d-1}} \left\| \frac{1}{N} \sum_n \hat{f}^*(x_n) x_n \sigma'(\langle w_j, x_n \rangle) - \mathbb{E}_x \left[ \hat{f}^*(x) x \sigma'(\langle w_j, x \rangle) \right] \right\| \right\|,$$

where for the last inequality we used that the derivative of $h$ is bounded. Now, from Lemma 32 of Damian et al. (2022), we know that with high probability,

$$\sup_{w \in S^{d-1}} \left\| \frac{1}{N} \sum_n \hat{f}^*(x_n) x_n \sigma'(\langle w_j, x_n \rangle) - \mathbb{E}_x \left[ \hat{f}^*(x) x \sigma'(\langle w_j, x \rangle) \right] \right\| = \tilde{O} \left( \sqrt{\frac{d}{N}} \right).$$

As a result,

$$\Delta_{1,1} = \tilde{O} \left( \sqrt{\frac{d}{N}} \right). \tag{3}$$

Let us now focus on

$$\Delta_{1,2} = \sup_{\tilde{x} \in S^{d-1}} \left\| \frac{1}{J^*} \sum_j \mathbb{E}_x \left[ \hat{f}^*(x) x \sigma'(\langle w_j, x \rangle) h'(\langle w_j, \tilde{x} \rangle) \right] - \mathbb{E}_{w,x} \left[ \hat{f}^*(x) x \sigma'(\langle w, x \rangle) h'(\langle w, \tilde{x} \rangle) \right] \right\|.$$

We will proceed analogously to the proof of Lemma 32 of Damian et al. (2022). We define $Y(\tilde{x})$ as

$$Y(\tilde{x}) := \frac{1}{J^*} \sum_j \mathbb{E}_x \left[ \hat{f}^*(x) x \sigma'(\langle w_j, x \rangle) h'(\langle w_j, \tilde{x} \rangle) \right] = \frac{1}{J^*} \sum_j \mathbb{E}_x \left[ \hat{f}^*(x) x \sigma'(\langle w_j, x \rangle) \right] h'(\langle w_j, \tilde{x} \rangle).$$

Moreover, we define an $\epsilon$-net $\mathcal{N}_\epsilon$ such that for all $\tilde{x} \in S^{d-1}$, there exists $\hat{x} \in \mathcal{N}_\epsilon$ such that $\|\tilde{x} - \hat{x}\| \leq \epsilon$. By a standard argument, such net exist with size $|\mathcal{N}_\epsilon| \leq e^{Cd \log(1/\epsilon)}$, for some constant $C$.

Here, we will set $\epsilon = \sqrt{\frac{d}{J^*}}$, and also denote $\mathcal{N}_{\frac{1}{4}}$, the minimal $\frac{1}{4}$-net of $S^{d-1}$ with $|\mathcal{N}_{\frac{1}{4}}| \leq e^{Cd}$.

Then,

$$\begin{aligned}
\Delta_{1,2} &= \sup_{\tilde{x} \in S^{d-1}} \left\| \frac{1}{J^*} \sum_j \mathbb{E}_x \left[ \hat{f}^*(x) x \sigma'(\langle w_j, x \rangle) h'(\langle w_j, \tilde{x} \rangle) \right] - \mathbb{E}_{w,x} \left[ \hat{f}^*(x) x \sigma'(\langle w, x \rangle) h'(\langle w, \tilde{x} \rangle) \right] \right\| \\
&= \sup_{\tilde{x} \in S^{d-1}} \| Y(\tilde{x}) - \mathbb{E}_w[Y(\tilde{x})] \| \\
&= \sup_{\tilde{x} \in S^{d-1}} \sup_{u \in S^{d-1}} \langle u, Y(\tilde{x}) - \mathbb{E}_w[Y(\tilde{x})] \rangle \\
&\leq 2 \sup_{\tilde{x} \in S^{d-1}} \sup_{u \in \mathcal{N}_{1/4}} \langle u, Y(\tilde{x}) - \mathbb{E}_w[Y(\tilde{x})] \rangle.
\end{aligned}$$

Since $h'$ is $O(1)$-Lipschitz, for a fixed $u \in S^{d-1}$, $Y_w(\tilde{x}) := \mathbb{E}_x \left[ \hat{f}^*(x) \langle u, x \rangle \sigma'(\langle w, x \rangle) h'(\langle w, \tilde{x} \rangle) \right]$ is also $O(1)$-Lipschitz. Indeed,

$$\begin{aligned}
|Y_w(\tilde{x}_1) - Y_w(\tilde{x}_2)| &= \left| \mathbb{E}_x \left[ \hat{f}^*(x) \langle u, x \rangle \sigma'(\langle w, x \rangle) \right] \right| |h'(\langle w, \tilde{x}_1 \rangle) - h'(\langle w, \tilde{x}_2 \rangle)| \\
&\lesssim \left| \mathbb{E}_x \left[ \hat{f}^*(x) \langle u, x \rangle \sigma'(\langle w, x \rangle) \right] \right| \|\tilde{x}_1 - \tilde{x}_2\| \\
&\leq \mathbb{E}_x \left[ \hat{f}^*(x)^2 \right]^{1/2} \mathbb{E}_x \left[ \langle u, x \rangle^2 \mathbb{1}_{\langle w, x \rangle > 0} \right]^{1/2} \|\tilde{x}_1 - \tilde{x}_2\|,
\end{aligned}$$

where for the last inequality we used Cauchy-Shwarz inequality. As $\langle u, x \rangle \sim N(0,1)$, by symmetry, $\mathbb{E}_x \left[ \langle u, x \rangle^2 \mathbb{1}_{\langle w, x \rangle > 0} \right] = \frac{1}{2} \mathbb{E}_x[\langle u, x \rangle^2] = \frac{1}{2}$, which implies

$$\left| \mathbb{E}_x \left[ \hat{f}^*(x) \langle u, x \rangle \sigma'(\langle w, x \rangle) \right] \right| \leq \frac{1}{\sqrt{2}}, \tag{4}$$

which proves the $O(1)$-Lipschitzness of $Y_w(\tilde{x})$. Consequently, $\langle u, Y(\tilde{x}) \rangle$ and $\langle u, \mathbb{E}_w[Y(\tilde{x})] \rangle$ are also $O(1)$-Lipschitz since Lipschitzness is preserved under mean and expectation, and $\langle u, Y(\tilde{x}) - \mathbb{E}_w[Y(\tilde{x})] \rangle$ as well since the difference of two $L$-Lipschitz functions is $2L$-Lipschitz.

Now, for all $\tilde{x} \in S^{d-1}$ and any $u$, there exist a $\hat{x} \in \mathcal{N}_\epsilon$ such that $\|\tilde{x} - \hat{x}\| \leq \epsilon$. Combining with the Lipschitzness of $\langle u, Y(\tilde{x}) - \mathbb{E}_w[Y(\tilde{x})] \rangle$, we obtain

$$\begin{aligned}
|\langle u, Y(\tilde{x}) - \mathbb{E}_w[Y(\tilde{x})] \rangle| &= |\langle u, Y(\hat{x}) - \mathbb{E}_w[Y(\hat{x})] \rangle| + |\langle u, Y(\tilde{x}) - \mathbb{E}_w[Y(\tilde{x})] \rangle - \langle u, Y(\hat{x}) - \mathbb{E}_w[Y(\hat{x})] \rangle| \\
&\leq |\langle u, Y(\hat{x}) - \mathbb{E}_w[Y(\hat{x})] \rangle| + O(\epsilon) \\
&\leq \sup_{\hat{x} \in \mathcal{N}_\epsilon} \sup_{u \in \mathcal{N}_{1/4}} |\langle u, Y(\hat{x}) - \mathbb{E}_w[Y(\hat{x})] \rangle| + O(\epsilon).
\end{aligned}$$

Therefore,

$$\sup_{\tilde{x} \in S^{d-1}} \sup_{u \in \mathcal{N}_{1/4}} \langle u, Y(\tilde{x}) - \mathbb{E}_w[Y(\tilde{x})] \rangle \leq \sup_{\tilde{x} \in \mathcal{N}_\epsilon} \sup_{u \in \mathcal{N}_{1/4}} |\langle u, Y(\hat{x}) - \mathbb{E}_w[Y(\hat{x})] \rangle| + O(\epsilon).$$

Let us denote $Z_j(\hat{x}) := \mathbb{E}_x \left[ \hat{f}^*(x) \langle u, x \rangle \sigma'(\langle w_j, x \rangle) h'(\langle w_j, \hat{x} \rangle) \right]$. Since $\mathbb{E}_x \left[ \hat{f}^*(x) \langle u, x \rangle \sigma'(\langle w_j, x \rangle) \right] \leq \frac{1}{2}$ as proved earlier in equation (4) and $\sup_{|z| \leq 1} h'(z) < \infty$, $Z_j$ is bounded and $O(1)$-sub gaussian.

As a result, for each $u \in \mathcal{N}_{\frac{1}{4}}$,

$$\frac{1}{J^*} \sum_j Z_j(\hat{x}) - \mathbb{E}[Z_j(\hat{x})] = \langle u, Y(\hat{x}) - \mathbb{E}_w[Y(\hat{x})] \rangle \lesssim \sqrt{\frac{2z}{J^*}},$$

with probability $1 - 2\mathrm{e}^{-z}$. By taking the union bound over $\mathcal{N}_\epsilon$ and $\mathcal{N}_{1/4}$, we obtain that with probability $1 - 2\mathrm{e}^{Cd\log(J^*/\epsilon) - z}$,

$$2 \sup_{\tilde{x} \in \mathcal{N}_\epsilon, u \in \mathcal{N}_{\frac{1}{4}}} \langle u, Y(\tilde{x}) - \mathbb{E}_w[Y(\tilde{x})] \rangle \lesssim \sqrt{\frac{2z}{J^*}}.$$

We choose $z = \tilde{\Theta}(Cd\log(J^*/\epsilon))$ and $\epsilon = \gamma^{-1}\sqrt{d/J^*}$ to obtain with high probability

$$\Delta_{1,2} = \sup_{\tilde{x} \in S^{d-1}} \|Y(\tilde{x}) - \mathbb{E}_w[Y(\tilde{x})]\| = \tilde{O}\left(\sqrt{\frac{d}{J^*}}\right). \tag{5}$$

We combine the obtained two inequalities (3) and (5) to derive

$$\Delta_1 \le \Delta_{1,1} + \Delta_{1,2} = \tilde{O}\left(\sqrt{\frac{d}{N}} + \sqrt{\frac{d}{J^*}}\right). \tag{6}$$

We will now focus on the second term

$$\Delta_2 := \sup_{\tilde{x}} \left\| \frac{1}{J^*} \sum_j w_j h''(\langle w_j, \tilde{x} \rangle) \langle g_j, \tilde{x} \rangle - \mathbb{E}_w \left[ w h''(\langle w, \tilde{x} \rangle) \left\langle \mathbb{E}\left[ \hat{f}^*(x) x \sigma'(\langle w, x \rangle) \right], \tilde{x} \right\rangle \right] \right\|.$$

We can similarly decompose this into two different terms

$$\Delta_{2,1} := \sup_{\tilde{x}} \left\| \frac{1}{J^*} \sum_j w_j h''(\langle w_j, \tilde{x} \rangle) \langle g_j, \tilde{x} \rangle - \frac{1}{J^*} \sum_j w_j h''(\langle w_j, \tilde{x} \rangle) \left\langle \mathbb{E}\left[ \hat{f}^*(x) x \sigma'(\langle w_j, x \rangle) \right], \tilde{x} \right\rangle \right\|,$$

$$\Delta_{2,2} := \sup_{\tilde{x}} \left\| \frac{1}{J^*} \sum_j w_j h''(\langle w_j, \tilde{x} \rangle) \left\langle \mathbb{E}\left[ \hat{f}^*(x) x \sigma'(\langle w_j, x \rangle) \right], \tilde{x} \right\rangle - \mathbb{E}_w \left[ w h''(\langle w, \tilde{x} \rangle) \left\langle \mathbb{E}\left[ \hat{f}^*(x) x \sigma'(\langle w, x \rangle) \right], \tilde{x} \right\rangle \right] \right\|.$$

In order to keep the notation simple, we define

$$\hat{g}_j := \mathbb{E}_x \left[ \hat{f}^*(x) x \sigma'(\langle w_j, x \rangle) \right].$$

Concerning the first term,

$$\Delta_{2,1} = \sup_{\tilde{x}} \left\| \frac{1}{J^*} \sum_j w_j h''(\langle w_j, \tilde{x} \rangle) \langle g_j, \tilde{x} \rangle - \frac{1}{J^*} \sum_j w_j h''(\langle w_j, \tilde{x} \rangle) \langle \hat{g}_j, \tilde{x} \rangle \right\|$$

$$= \sup_{\tilde{x}} \left\| \frac{1}{J^*} \sum_j w_j h''(\langle w_j, \tilde{x} \rangle) \langle g_j - \hat{g}_j, \tilde{x} \rangle \right\|.$$

Under the same conditions as the first half of the proof, with high probability, we already have

$$\Delta_{2,1} \le \sup_{\tilde{x}} \frac{1}{J^*} \sum_j \|h''(\langle w_j, \tilde{x} \rangle) w_j \langle g_j - \hat{g}_j, \tilde{x} \rangle\|$$

$$\le \sup_{\tilde{x}} \frac{1}{J^*} \sum_j |h''(\langle w_j, \tilde{x} \rangle)| \|w_j\| \|g_j - \hat{g}_j\| \|\tilde{x}\|$$

$$\lesssim \sqrt{\frac{d}{N}} \sup_{\tilde{x}} \frac{1}{J^*} \sum_j h''(\langle w_j, \tilde{x} \rangle),$$

where we used Lemma 32 of Damian et al. (2022), $\|w_j\| = \|\tilde{x}\| = 1$ for the third inequality. Since $h''$ is bounded,

$$\Delta_{2,1} \lesssim \sqrt{\frac{d}{N}}. \tag{7}$$

As for $\Delta_{2,2}$,

$$
\begin{aligned}
\Delta_{2,2} &= \left\| \frac{1}{J^*} \sum_j w_j h''(\langle w_j, \tilde{x}\rangle) \langle \hat{g}_j, \tilde{x}\rangle - \mathbb{E}_w \left[ w h''(\langle w, \tilde{x}\rangle) \langle \hat{g}_j, \tilde{x}\rangle \right] \right\| \\
&= \sup_{u \in S^{d-1}} \left\langle u, \frac{1}{J^*} \sum_j w_j h''(\langle w_j, \tilde{x}\rangle) \langle \hat{g}_j, \tilde{x}\rangle - \mathbb{E}_w \left[ w h''(\langle w, \tilde{x}\rangle) \langle \hat{g}_j, \tilde{x}\rangle \right] \right\rangle \\
&= \sup_{u \in S^{d-1}} \frac{1}{J^*} \sum_j \langle u, w_j\rangle h''(\langle w_j, \tilde{x}\rangle) \langle \hat{g}_j, \tilde{x}\rangle - \mathbb{E}_w \left[ \langle u, w_j\rangle h''(\langle w, \tilde{x}\rangle) \langle \hat{g}_j, \tilde{x}\rangle \right] \\
&\leq 2 \sup_{u \in \mathcal{N}_{1/4}} \frac{1}{J^*} \sum_j \langle u, w_j\rangle h''(\langle w_j, \tilde{x}\rangle) \langle \hat{g}_j, \tilde{x}\rangle - \mathbb{E}_w \left[ \langle u, w_j\rangle h''(\langle w, \tilde{x}\rangle) \langle \hat{g}_j, \tilde{x}\rangle \right].
\end{aligned}
$$

We define

$$Z_j(u, \tilde{x}) := \langle u, w_j\rangle h''(\langle w_j, \tilde{x}\rangle) \langle \hat{g}_j, \tilde{x}\rangle - \mathbb{E}_w \left[ \langle u, w_j\rangle h''(\langle w, \tilde{x}\rangle) \langle \hat{g}_j, \tilde{x}\rangle \right].$$

Then,

$$
\begin{aligned}
|Z_j(u, \tilde{x})| &\leq |\langle u, w_j\rangle h''(\langle w_j, \tilde{x}\rangle) \langle \hat{g}_j, \tilde{x}\rangle| + |\mathbb{E}_w \left[ \langle u, w_j\rangle h''(\langle w, \tilde{x}\rangle) \langle \hat{g}_j, \tilde{x}\rangle \right]| \\
&\leq |\langle u, w_j\rangle| |h''(\langle w_j, \tilde{x}\rangle)| |\langle \hat{g}_j, \tilde{x}\rangle| + \mathbb{E}_w \left[ |\langle u, w_j\rangle| |h''(\langle w, \tilde{x}\rangle)| |\langle \hat{g}_j, \tilde{x}\rangle| \right] \\
&\leq 2 \cdot 1 \cdot \sup_{|z| \leq 1} |h''(z)| \cdot \frac{1}{\sqrt{2}} \\
&\lesssim 1,
\end{aligned}
$$

where we used $|\langle u, \tilde{x}\rangle| \leq 1$, and $\langle \hat{g}_j, \tilde{x}\rangle \leq 1/\sqrt{2}$ from Equation (4) for the fourth inequality, and that $h''$ is bounded on the compact set for the last inequality. Since $h''(\langle w_j, \tilde{x}\rangle)$ and $\langle \hat{g}_j, \tilde{x}\rangle$ are bounded, and Lipschitz ($C^2$ functions on a compact set is Lipschitz), $Z_j(u, \tilde{x})$ is $O(1)$-Lipschitz with respect to $\tilde{x}$. Therefore, similarly to the previous argument, we obtain

$$\sup_{\tilde{x} \in S^{d-1}} \sup_{u \in \mathcal{N}_{1/4}} \frac{1}{J^*} \sum_j Z_j(u, \tilde{x}) = \sup_{\hat{x} \in \mathcal{N}_\epsilon} \sup_{u \in \mathcal{N}_{1/4}} \frac{1}{J^*} \sum_j Z_j(u, \hat{x}) + O(\epsilon).$$

Since $Z_j(u, \tilde{x})$ is $O(1)$-sub Gaussian, for each $u \in \mathcal{N}_{1/4}$, we have with probability $1 - 2e^{-z}$

$$\frac{1}{J^*} \sum_j Z_j(u, \tilde{x}) \lesssim \sqrt{\frac{2z}{J^*}},$$

which by union bound over $\mathcal{N}_\epsilon$ and $\mathcal{N}_{1/4}$ and setting $z$ and $\epsilon$ accordingly leads to

$$\Delta_{2,2} = \tilde{O}\left( \sqrt{\frac{d}{J^*}} \right). \tag{8}$$

By combining the two inequalities (7) and (8), we obtain

$$\Delta_2 = \tilde{O}\left( \sqrt{\frac{d}{N}} + \sqrt{\frac{d}{J^*}} \right). \tag{9}$$

Finally, combining the three bounds (6) and (9) with Lemma B.18, we conclude the desired result.

$$\square$$

**Lemma B.20.** *Under Assumptions 3.1, 3.3, 3.4, 3.6, 3.7 and 3.8, with high probability,*

$$\epsilon = \tilde{O}\left(d^{\frac{1}{2}}N^{-\frac{1}{2}} + d^{\frac{1}{2}}J^{*-\frac{1}{2}}\right).$$

*where $\epsilon$ is defined in Corollary B.7.*

*Proof.* The statement follows by showing that with high probability,

$$\sup_{w,x}\left\|\frac{1}{J^*N}\sum_{j,n}\epsilon_n x_n \sigma'(\langle w_{i,j}, x_n\rangle)h'(\langle w_{i,j}, \tilde{x}\rangle)\right\| = \tilde{O}\left(d^{\frac{1}{2}}N^{-\frac{1}{2}}\right),$$

and

$$\sup_{w,x}\left\|\frac{1}{J^*}\sum_j w_{i,j}h''(\langle w_{i,j}, \tilde{x}\rangle)\left\langle\frac{1}{N}\sum_n\epsilon_n x_n\sigma'(\langle w_{i,j}, x_n\rangle), \tilde{x}\right\rangle\right\| = \tilde{O}\left(d^{\frac{1}{2}}N^{-\frac{1}{2}} + d^{\frac{1}{2}}J^{*-\frac{1}{2}}\right).$$

The first equation is a direct consequence of Lemma 33 of Damian et al. (2022) since $\|h'\|_\infty = 1$. The second inequality can be shown with the same approach as the derivation of $\Delta_2$ in the proof of Lemma B.19 by replacing $\hat{f}^*$ with $\epsilon_n$ which is bounded. $\qquad\square$

Finally, we can prove Theorem B.4.

*Proof of Theorem B.4.* We just need to substitute $G$ and $\epsilon$ in Corollary B.7 with the bounds we obtained in Theorem B.19 and Lemma B.20. $\qquad\square$

Therefore, the first phase of DD captures the latent structure of the original problem and translates it into the input space.

### B.3.5. CONCENTRATION OF EMPIRICAL GRADIENT (FOR SOFTPLUS)

In this part, we complement the previous proof by showing a version customized for the softplus activation function $h(z) = \frac{1}{\gamma_s}\log(1 + e^{\gamma_s z})$. We show that the dependence of $\gamma_s$ can be absorbed into $J^*$, the parameter that appears as a feature of distillation.

**Theorem B.21.** *Under Assumptions 3.1, 3.3, 3.4, 3.6 and 3.7, if we use the softplus activation function $h(z) = \frac{1}{\gamma_s}\log(1 + e^{\gamma_s z})$, then with high probability,*

$$G = c_d\langle\beta, \tilde{x}\rangle\beta + \tilde{O}\left(dN^{-\frac{1}{2}} + \gamma_s d^{-2} + \gamma_s^{\frac{1}{2}}d^{\frac{3}{4}}J^{*-\frac{1}{2}} + \gamma_s dJ^{*-1}\right).$$

*Proof.* Since

$$G = \hat{G} + G - \hat{G} \le \hat{G} + \sup_{\tilde{x}\in S^{d-1}}\|G - \hat{G}\|,$$

and the first term is already evaluated in Theorem B.18, we will evaluate the second. From the form of $G$ and $\hat{G}$, we can divide it into the following two terms:

$$\Delta_1 := \sup_{\tilde{x}}\left\|\frac{1}{J^*}\sum_j g_j h'(\langle w_j, \tilde{x}\rangle) - \mathbb{E}_w\left[\mathbb{E}_x\left[\hat{f}^*(x)x\sigma'(\langle w, x\rangle)\right]h'(\langle w, \tilde{x}\rangle)\right]\right\|,$$

$$\Delta_2 := \sup_{\tilde{x}}\left\|\frac{1}{J^*}\sum_j w_j h''(\langle w_j, \tilde{x}\rangle)\langle g_j, \tilde{x}\rangle - \mathbb{E}_w\left[wh''(\langle w, \tilde{x}\rangle)\left\langle\mathbb{E}\left[\hat{f}^*(x)x\sigma'(\langle w, x\rangle)\right], \tilde{x}\right\rangle\right]\right\|,$$

such that $\sup_{\tilde{x}}\|G - \hat{G}\| \le \Delta_1 + \Delta_2$, and $g_j = \frac{1}{N}\sum_n\hat{f}^*(x_n)\sigma'(\langle w_j, x_n\rangle)$.

Let us first evaluate $\Delta_1$.

$$\Delta_1 = \sup_{\tilde{x} \in S^{d-1}} \left\| \frac{1}{J^*N} \sum_{j,n} \hat{f}^*(x_n)x_n\sigma'(\langle w_j, x_n\rangle)h'(\langle w_j, \tilde{x}\rangle) - \mathbb{E}_{w,x}\left[\hat{f}^*(x)x\sigma'(\langle w, x\rangle)h'(\langle w, \tilde{x}\rangle)\right] \right\|$$

$$= \sup_{\tilde{x} \in S^{d-1}} \left\| \frac{1}{J^*N} \sum_{j,n} \hat{f}^*(x_n)x_n\sigma'(\langle w_j, x_n\rangle)h'(\langle w_j, \tilde{x}\rangle) - \frac{1}{J^*}\sum_j \mathbb{E}_x\left[\hat{f}^*(x)x\sigma'(\langle w_j, x\rangle)h'(\langle w_j, \tilde{x}\rangle)\right] \right\|$$

$$+ \sup_{\tilde{x} \in S^{d-1}} \left\| \frac{1}{J^*}\sum_j \mathbb{E}_x\left[\hat{f}^*(x)x\sigma'(\langle w_j, x\rangle)h'(\langle w_j, \tilde{x}\rangle)\right] - \mathbb{E}_{w,x}\left[\hat{f}^*(x)x\sigma'(\langle w, x\rangle)h'(\langle w, \tilde{x}\rangle)\right] \right\|.$$

We defined the first term of the right hand side as $\Delta_{1,1}$ and the second as $\Delta_{1,2}$. As for $\Delta_{1,1}$,

$$\Delta_{1,1} = \sup_{\tilde{x} \in S^{d-1}} \left\| \frac{1}{J^*}\sum_j \left\{ \frac{1}{N}\sum_n \hat{f}^*(x_n)x_n\sigma'(\langle w_j, x_n\rangle)h'(\langle w_j, \tilde{x}\rangle) - \mathbb{E}_x\left[\hat{f}^*(x)x\sigma'(\langle w_j, x\rangle)h'(\langle w_j, \tilde{x}\rangle)\right] \right\} \right\|$$

$$\leq \sup_{\tilde{x} \in S^{d-1}} \left\| \sup_{w \in S^{d-1}} \left\| \frac{1}{N}\sum_n \hat{f}^*(x_n)x_n\sigma'(\langle w_j, x_n\rangle)h'(\langle w_j, \tilde{x}\rangle) - \mathbb{E}_x\left[\hat{f}^*(x)x\sigma'(\langle w_j, x\rangle)h'(\langle w_j, \tilde{x}\rangle)\right] \right\| \right\|$$

$$= \sup_{\tilde{x} \in S^{d-1}} \left\| \sup_{w \in S^{d-1}} \left\| \left\{ \frac{1}{N}\sum_n \hat{f}^*(x_n)x_n\sigma'(\langle w_j, x_n\rangle) - \mathbb{E}_x\left[\hat{f}^*(x)x\sigma'(\langle w_j, x\rangle)\right] \right\} h'(\langle w_j, \tilde{x}\rangle) \right\| \right\|$$

$$\leq \sup_{\tilde{x} \in S^{d-1}} \left\| \sup_{w \in S^{d-1}} \left\| \frac{1}{N}\sum_n \hat{f}^*(x_n)x_n\sigma'(\langle w_j, x_n\rangle) - \mathbb{E}_x\left[\hat{f}^*(x)x\sigma'(\langle w_j, x\rangle)\right] \right\| \right\|,$$

where for the last inequality, we used that the derivative of $h$ is bounded by 1. Now, from Lemma 32 of Damian et al. (2022), we know that with high probability,

$$\sup_{w \in S^{d-1}} \left\| \frac{1}{N}\sum_n \hat{f}^*(x_n)x_n\sigma'(\langle w_j, x_n\rangle) - \mathbb{E}_x\left[\hat{f}^*(x)x\sigma'(\langle w_j, x\rangle)\right] \right\| = \tilde{O}\left( \sqrt{\frac{d}{N}} \right).$$

As a result,

$$\Delta_{1,1} = \tilde{O}\left( \sqrt{\frac{d}{N}} \right). \tag{10}$$

Let us now focus on

$$\Delta_{1,2} = \sup_{\tilde{x} \in S^{d-1}} \left\| \frac{1}{J^*}\sum_j \mathbb{E}_x\left[\hat{f}^*(x)x\sigma'(\langle w_j, x\rangle)h'(\langle w_j, \tilde{x}\rangle)\right] - \mathbb{E}_{w,x}\left[\hat{f}^*(x)x\sigma'(\langle w, x\rangle)h'(\langle w, \tilde{x}\rangle)\right] \right\|.$$

We will proceed analogously to the proof of Lemma 32 of Damian et al. (2022). We define $Y(\tilde{x})$ as

$$Y(\tilde{x}) := \frac{1}{J^*}\sum_j \mathbb{E}_x\left[\hat{f}^*(x)x\sigma'(\langle w_j, x\rangle)h'(\langle w_j, \tilde{x}\rangle)\right] = \frac{1}{J^*}\sum_j \mathbb{E}_x\left[\hat{f}^*(x)x\sigma'(\langle w_j, x\rangle)\right]h'(\langle w_j, \tilde{x}\rangle).$$

Moreover, we define an $\epsilon$-net $\mathcal{N}_\epsilon$ such that for all $\tilde{x} \in S^{d-1}$, there exists $\hat{x} \in \mathcal{N}_\epsilon$ such that $\|\tilde{x} - \hat{x}\| \leq \epsilon$. By a standard argument, such net exists with size $|\mathcal{N}_\epsilon| \leq e^{Cd\log(1/\epsilon)}$, for some constant $C$.

Here, we will set $\epsilon = \gamma_s^{-1}\sqrt{\frac{d}{J^*}}$, and also denote $\mathcal{N}_{\frac{1}{4}}$, the minimal $\frac{1}{4}$-net of $S^{d-1}$ with $|\mathcal{N}_{\frac{1}{4}}| \leq e^{Cd}$.

Then,

$$
\begin{aligned}
\Delta_{1,2} &= \sup_{\tilde{x} \in S^{d-1}} \left\| \frac{1}{J^*} \sum_j \mathbb{E}_x \left[ \hat{f}^*(x) x \sigma'(\langle w_j, x \rangle) h'(\langle w_j, \tilde{x} \rangle) \right] - \mathbb{E}_{w,x} \left[ \hat{f}^*(x) x \sigma'(\langle w, x \rangle) h'(\langle w, \tilde{x} \rangle) \right] \right\| \\
&= \sup_{\tilde{x} \in S^{d-1}} \| Y(\tilde{x}) - \mathbb{E}_w[Y(\tilde{x})] \| \\
&= \sup_{\tilde{x} \in S^{d-1}} \sup_{u \in S^{d-1}} \langle u, Y(\tilde{x}) - \mathbb{E}_w[Y(\tilde{x})] \rangle \\
&\leq 2 \sup_{\tilde{x} \in S^{d-1}} \sup_{u \in \mathcal{N}_{1/4}} \langle u, Y(\tilde{x}) - \mathbb{E}_w[Y(\tilde{x})] \rangle .
\end{aligned}
$$

Since $h'$ is $O(\gamma_s)$-Lipschitz, for a fixed $u \in S^{d-1}$, $Y_w(\tilde{x}) := \mathbb{E}_x \left[ \hat{f}^*(x) \langle u, x \rangle \sigma'(\langle w, x \rangle) h'(\langle w, \tilde{x} \rangle) \right]$ is also $O(\gamma_s)$-Lipschitz. Indeed,

$$
\begin{aligned}
|Y_w(\tilde{x}_1) - Y_w(\tilde{x}_2)| &= \left| \mathbb{E}_x \left[ \hat{f}^*(x) \langle u, x \rangle \sigma'(\langle w, x \rangle) \right] \right| |h'(\langle w, \tilde{x}_1 \rangle) - h'(\langle w, \tilde{x}_2 \rangle)| \\
&\lesssim \gamma_s \left| \mathbb{E}_x \left[ \hat{f}^*(x) \langle u, x \rangle \sigma'(\langle w, x \rangle) \right] \right| \|\tilde{x}_1 - \tilde{x}_2\| \\
&\leq \gamma_s \mathbb{E}_x \left[ \hat{f}^*(x)^2 \right]^{1/2} \mathbb{E}_x \left[ \langle u, x \rangle^2 \mathbb{1}_{\langle w, x \rangle > 0} \right]^{1/2} \|\tilde{x}_1 - \tilde{x}_2\|,
\end{aligned}
$$

where for the last inequality, we used Cauchy-Shwarz inequality. As $\langle u, x \rangle \sim N(0,1)$, by symmetry, $\mathbb{E}_x \left[ \langle u, x \rangle^2 \mathbb{1}_{\langle w, x \rangle > 0} \right] = \frac{1}{2} \mathbb{E}_x[\langle u, x \rangle^2] = \frac{1}{2}$, which implies

$$
\left| \mathbb{E}_x \left[ \hat{f}^*(x) \langle u, x \rangle \sigma'(\langle w, x \rangle) \right] \right| \leq \frac{1}{\sqrt{2}}, \tag{11}
$$

which proves the $O(\gamma_s)$-Lipschitzness of $Y_w(\tilde{x})$. Consequently, $\langle u, Y(\tilde{x}) \rangle$ and $\langle u, \mathbb{E}_w[Y(\tilde{x})] \rangle$ are also $O(\gamma_s)$-Lipschitz since Lipschitzness is preserved under mean and expectation, and $\langle u, Y(\tilde{x}) - \mathbb{E}_w[Y(\tilde{x})] \rangle$ as well since the difference of two $L$-Lipschitz functions is $2L$-Lipschitz.

Now, for all $\tilde{x} \in S^{d-1}$ and any $u$, there exists a $\hat{x} \in \mathcal{N}_\epsilon$ such that $\|\tilde{x} - \hat{x}\| \leq \epsilon$. Combining with the Lipschitzness of $\langle u, Y(\tilde{x}) - \mathbb{E}_w[Y(\tilde{x})] \rangle$, we obtain

$$
\begin{aligned}
|\langle u, Y(\tilde{x}) - \mathbb{E}_w[Y(\tilde{x})] \rangle| &= |\langle u, Y(\hat{x}) - \mathbb{E}_w[Y(\hat{x})] \rangle| + |\langle u, Y(\tilde{x}) - \mathbb{E}_w[Y(\tilde{x})] \rangle - \langle u, Y(\hat{x}) - \mathbb{E}_w[Y(\hat{x})] \rangle| \\
&\leq |\langle u, Y(\hat{x}) - \mathbb{E}_w[Y(\hat{x})] \rangle| + O(\gamma_s \epsilon) \\
&\leq \sup_{\hat{x} \in \mathcal{N}_\epsilon} \sup_{u \in \mathcal{N}_{1/4}} |\langle u, Y(\hat{x}) - \mathbb{E}_w[Y(\hat{x})] \rangle| + O(\gamma_s \epsilon).
\end{aligned}
$$

Therefore,

$$
\sup_{\tilde{x} \in S^{d-1}} \sup_{u \in \mathcal{N}_{1/4}} \langle u, Y(\tilde{x}) - \mathbb{E}_w[Y(\tilde{x})] \rangle \leq \sup_{\tilde{x} \in \mathcal{N}_\epsilon} \sup_{u \in \mathcal{N}_{1/4}} |\langle u, Y(\hat{x}) - \mathbb{E}_w[Y(\hat{x})] \rangle| + O(\gamma_s \epsilon).
$$

Let us denote $Z_j(\hat{x}) := \mathbb{E}_x \left[ \hat{f}^*(x) \langle u, x \rangle \sigma'(\langle w_j, x \rangle) h'(\langle w_j, \hat{x} \rangle) \right]$. Since $\mathbb{E}_x \left[ \hat{f}^*(x) \langle u, x \rangle \sigma'(\langle w_j, x \rangle) \right] \leq \frac{1}{2}$ as proved earlier and $\sup_{|z| \leq 1} h'(z) < \infty$, $Z_j$ is bounded and $O(1)$-sub gaussian.

As a result, for each $u \in \mathcal{N}_{\frac{1}{4}}$,

$$
\frac{1}{J^*} \sum_j Z_j(\hat{x}) - \mathbb{E}[Z_j(\hat{x})] = \langle u, Y(\hat{x}) - \mathbb{E}_w[Y(\hat{x})] \rangle \lesssim \sqrt{\frac{2z}{J^*}},
$$

with probability $1 - 2\mathrm{e}^{-z}$. By taking the union bound over $\mathcal{N}_\epsilon$ and $\mathcal{N}_{1/4}$, we obtain that with probability $1 - 2\mathrm{e}^{Cd \log(J^*/\epsilon) - z}$,

$$
2 \sup_{\tilde{x} \in \mathcal{N}_\epsilon, u \in \mathcal{N}_{\frac{1}{4}}} \langle u, Y(\tilde{x}) - \mathbb{E}_w[Y(\tilde{x})] \rangle \lesssim \sqrt{\frac{2z}{J^*}}.
$$

We choose $z = \tilde{\Theta}(Cd\log(J^*/\epsilon))$ and $\epsilon = \gamma_s^{-1}\sqrt{d/J^*}$ to obtain with high probability

$$\Delta_{1,2} = \sup_{\tilde{x} \in S^{d-1}} \|Y(\tilde{x}) - \mathbb{E}_w[Y(\tilde{x})]\| = \tilde{O}\left(\sqrt{\frac{d}{J^*}}\right). \tag{12}$$

We combine the obtained two inequalities (10) and (12) to derive

$$\Delta_1 \le \Delta_{1,1} + \Delta_{1,2} = \tilde{O}\left(\sqrt{\frac{d}{N}} + \sqrt{\frac{d}{J^*}}\right). \tag{13}$$

We will now focus on the second term

$$\Delta_2 := \sup_{\tilde{x}} \left\| \frac{1}{J^*}\sum_j w_j h''(\langle w_j, \tilde{x}\rangle)\langle g_j, \tilde{x}\rangle - \mathbb{E}_w\left[wh''(\langle w, \tilde{x}\rangle)\left\langle \mathbb{E}\left[\hat{f}^*(x)x\sigma'(\langle w, x\rangle)\right], \tilde{x}\right\rangle\right] \right\|.$$

We can similarly decompose this into two different terms

$$\Delta_{2,1} := \sup_{\tilde{x}} \left\| \frac{1}{J^*}\sum_j w_j h''(\langle w_j, \tilde{x}\rangle)\langle g_j, \tilde{x}\rangle - \frac{1}{J^*}\sum_j w_j h''(\langle w_j, \tilde{x}\rangle)\left\langle \mathbb{E}\left[\hat{f}^*(x)x\sigma'(\langle w_j, x\rangle)\right], \tilde{x}\right\rangle \right\|,$$

$$\Delta_{2,2} := \sup_{\tilde{x}} \left\| \frac{1}{J^*}\sum_j w_j h''(\langle w_j, \tilde{x}\rangle)\left\langle \mathbb{E}\left[\hat{f}^*(x)x\sigma'(\langle w_j, x\rangle)\right], \tilde{x}\right\rangle - \mathbb{E}_w\left[wh''(\langle w, \tilde{x}\rangle)\left\langle \mathbb{E}\left[\hat{f}^*(x)x\sigma'(\langle w, x\rangle)\right], \tilde{x}\right\rangle\right] \right\|.$$

In order to keep the notation simple, we define

$$\hat{g}_j := \mathbb{E}_x\left[\hat{f}^*(x)x\sigma'(\langle w_j, x\rangle)\right].$$

Concerning the first term,

$$\Delta_{2,1} = \sup_{\tilde{x}} \left\| \frac{1}{J^*}\sum_j w_j h''(\langle w_j, \tilde{x}\rangle)\langle g_j, \tilde{x}\rangle - \frac{1}{J^*}\sum_j w_j h''(\langle w_j, \tilde{x}\rangle)\langle \hat{g}_j, \tilde{x}\rangle \right\|$$

$$= \sup_{\tilde{x}} \left\| \frac{1}{J^*}\sum_j w_j h''(\langle w_j, \tilde{x}\rangle)\langle g_j - \hat{g}_j, \tilde{x}\rangle \right\|.$$

Under the same conditions as the first half of the proof, with high probability, we already have

$$\Delta_{2,1} \le \sup_{\tilde{x}} \frac{1}{J^*}\sum_j \|h''(\langle w_j, \tilde{x}\rangle)w_j\langle g_j - \hat{g}_j, \tilde{x}\rangle\|$$

$$\le \sup_{\tilde{x}} \frac{1}{J^*}\sum_j |h''(\langle w_j, \tilde{x}\rangle)|\|w_j\|\|g_j - \hat{g}_j\|\|\tilde{x}\|$$

$$\lesssim \sqrt{\frac{d}{N}}\sup_{\tilde{x}} \frac{1}{J^*}\sum_j h''(\langle w_j, \tilde{x}\rangle),$$

where we used Lemma 32 of Damian et al. (2022), $\|w_j\| = \|\tilde{x}\| = 1$, and $h'' \ge 0$ for the third inequality. Now by using Lemma B.25, we can show that

$$\Delta_{2,1} = \tilde{O}\left(\sqrt{\frac{d^2}{N}} + \sqrt{\frac{d}{N}}\left(\sqrt{\frac{\gamma_s d^{3/2}}{J^*}} + \frac{\gamma_s d}{J^*}\right)\right). \tag{14}$$

As for $\Delta_{2,2}$,

$$
\begin{aligned}
\Delta_{2,2} &= \left\| \frac{1}{J^*} \sum_j w_j h''(\langle w_j, \tilde{x} \rangle) \langle \hat{g}_j, \tilde{x} \rangle - \mathbb{E}_w \left[ w h''(\langle w, \tilde{x} \rangle) \langle \hat{g}_j, \tilde{x} \rangle \right] \right\| \\
&= \sup_{u \in S^{d-1}} \left\langle u, \frac{1}{J^*} \sum_j w_j h''(\langle w_j, \tilde{x} \rangle) \langle \hat{g}_j, \tilde{x} \rangle - \mathbb{E}_w \left[ w h''(\langle w, \tilde{x} \rangle) \langle \hat{g}_j, \tilde{x} \rangle \right] \right\rangle \\
&= \sup_{u \in S^{d-1}} \frac{1}{J^*} \sum_j \langle u, w_j \rangle h''(\langle w_j, \tilde{x} \rangle) \langle \hat{g}_j, \tilde{x} \rangle - \mathbb{E}_w \left[ \langle u, w_j \rangle h''(\langle w, \tilde{x} \rangle) \langle \hat{g}_j, \tilde{x} \rangle \right] \\
&\leq 2 \sup_{u \in \mathcal{N}_{1/4}} \frac{1}{J^*} \sum_j \langle u, w_j \rangle h''(\langle w_j, \tilde{x} \rangle) \langle \hat{g}_j, \tilde{x} \rangle - \mathbb{E}_w \left[ \langle u, w_j \rangle h''(\langle w, \tilde{x} \rangle) \langle \hat{g}_j, \tilde{x} \rangle \right].
\end{aligned}
$$

We define,

$$
Z_j(u, \tilde{x}) := \langle u, w_j \rangle h''(\langle w_j, \tilde{x} \rangle) \langle \hat{g}_j, \tilde{x} \rangle - \mathbb{E}_w \left[ \langle u, w_j \rangle h''(\langle w, \tilde{x} \rangle) \langle \hat{g}_j, \tilde{x} \rangle \right].
$$

Then,

$$
\begin{aligned}
|Z_j(u, \tilde{x})| &\leq |\langle u, w_j \rangle h''(\langle w_j, \tilde{x} \rangle) \langle \hat{g}_j, \tilde{x} \rangle| + |\mathbb{E}_w \left[ \langle u, w_j \rangle h''(\langle w, \tilde{x} \rangle) \langle \hat{g}_j, \tilde{x} \rangle \right]| \\
&\leq |\langle u, w_j \rangle| |h''(\langle w_j, \tilde{x} \rangle)| |\langle \hat{g}_j, \tilde{x} \rangle| + \mathbb{E}_w \left[ |\langle u, w_j \rangle| |h''(\langle w, \tilde{x} \rangle)| |\langle \hat{g}_j, \tilde{x} \rangle| \right] \\
&\leq 2 \cdot 1 \cdot \frac{\gamma_s}{4} \cdot \frac{1}{\sqrt{2}},
\end{aligned}
$$

where we used $|\langle u, \tilde{x} \rangle| \leq 1$, $\|h''\|_\infty = \gamma_s/4$, and $\langle \hat{g}_j, \tilde{x} \rangle \leq 1/\sqrt{2}$ from Equation (11) for the last inequality. Similarly,

$$
\mathbb{E} \left[ Z_j(u, \tilde{x})^2 \right] \lesssim \mathbb{E}_w [h''(\langle w, \tilde{x} \rangle)^2] = m_\gamma,
$$

where $m_\gamma$ is defined in Lemma B.25. Therefore, by Bernstein inequality,

$$
P \left( \left| \frac{1}{J^*} \sum_j Z_j(u, \tilde{x}) \right| \geq C \left( \sqrt{\frac{m_\gamma z}{J^*}} + \frac{\gamma_s z}{J^*} \right) \right) \leq 2 e^{-z}.
$$

Since, $h''(\langle w_j, \tilde{x} \rangle)$ and $\langle \hat{g}_j, \tilde{x} \rangle$ are respectively bounded by $O(\gamma_s)$ and $O(1)$, and Lipschitz with respect to $\tilde{x}$ with constants $O(\gamma_s^2)$ and $O(1)$, $Z_j(u, \tilde{x})$ is $O(\gamma_s^2)$-Lipschitz with respect to $\tilde{x}$. Therefore, similarly to the previous argument, we obtain

$$
\sup_{\tilde{x} \in S^{d-1}} \sup_{u \in \mathcal{N}_{1/4}} |Z_j(u, \tilde{x})| = \sup_{\hat{x} \in \mathcal{N}_\epsilon} \sup_{u \in \mathcal{N}_{1/4}} |Z_j(u, \hat{x})| + O(\gamma_s^2 \epsilon),
$$

which by union bound over $\mathcal{N}_\epsilon$ and $\mathcal{N}_{1/4}$ leads to

$$
\Delta_{2,2} = \tilde{O} \left( \sqrt{\frac{\gamma_s d^{3/2}}{J^*}} + \frac{\gamma_s d}{J^*} \right). \tag{15}
$$

By combining the two inequalities (14) and (15), we obtain

$$
\Delta_2 = \tilde{O} \left( \sqrt{\frac{d^2}{N}} + \sqrt{\frac{d}{N}} \left( \sqrt{\frac{\gamma_s d^{3/2}}{J^*}} + \frac{\gamma_s d}{J^*} \right) + \sqrt{\frac{\gamma_s d^{3/2}}{J^*}} + \frac{\gamma_s d}{J^*} \right). \tag{16}
$$

Finally, combining the three bounds (13) and (16) with Lemma B.18, we conclude the desired result.

$\square$

**Lemma B.22.** *Under Assumptions 3.1, 3.3, 3.4, 3.6, 3.7 and 3.8, With high probability, when the activation function is Softplus with parameter $\gamma_s$*

$$
\epsilon = \tilde{O} \left( d N^{-\frac{1}{2}} + \gamma_s^{\frac{1}{2}} d^{\frac{3}{2}} J^{*-\frac{1}{2}} + \gamma_s d J^{*-1} \right).
$$

*where $\epsilon$ is defined in Corollary B.7.*

*Proof.* The statement follows by showing that

$$\sup_{w,x} \left\| \frac{1}{J^*N} \sum_{j,n} \epsilon_n x_n \sigma'(\langle w_{i,j}, x_n \rangle) h'(\langle w_{i,j}, \tilde{x} \rangle) \right\| = \tilde{O}\left( d^{\frac{1}{2}} N^{-\frac{1}{2}} \right),$$

and

$$\sup_{w,x} \left\| \frac{1}{J^*} \sum_j w_{i,j} h''(\langle w_{i,j}, \tilde{x} \rangle) \left\langle \frac{1}{N} \sum_n \epsilon_n x_n \sigma'(\langle w_{i,j}, x_n \rangle), \tilde{x} \right\rangle \right\| = \tilde{O}\left( dN^{-\frac{1}{2}} + \gamma_s^{\frac{1}{2}} d^{\frac{3}{2}} J^{*-\frac{1}{2}} + \gamma_s d J^{*-1} \right).$$

The first equation is a direct consequence of Lemma 33 of Damian et al. (2022) since $\|h'\|_\infty = 1$. The second inequality can be shown with the same approach as the derivation of $\Delta_2$ in the proof of Lemma B.19 by replacing $\hat{f}^*$ to $\epsilon_n$ which is bounded. $\qquad\square$

Finally, we can prove Theorem B.4.

*Proof of Theorem B.4.* We just need to substitute $G$ and $\epsilon$ in Corollary B.7 with the bounds we obtained in Theorem B.19 and Lemma B.20. $\qquad\square$

We will use the general bound in the remainder of the proof.

### B.3.6. OTHER LEMMAS

**Lemma B.23.** *For non-negative constants $p_1$ and $p_2$, bounded function $m$ such that $\sup_{|t|\leq 1} |m(t)| = M < \infty$,*

$$\left| \int_{-1}^1 m(t)(1-t^2)^{p_1} t^{p_2} \mathrm{d}t \right| \leq 2M \int_0^1 (1-t^2)^{p_1} t^{p_2} \mathrm{d}t.$$

*Proof.* This can be proved as follows:

$$\left| \int_{-1}^1 m(t)(1-t^2)^{p_1} t^{p_2} \mathrm{d}t \right| \leq \int_{-1}^1 \left| m(t)(1-t^2)^{p_1} t^{p_2} \right| \mathrm{d}t$$

$$\leq \int_{-1}^1 |m(t)| (1-t^2)^{p_1} |t|^{p_2} \mathrm{d}t$$

$$\leq M \int_{-1}^1 (1-t^2)^{p_1} |t|^{p_2} \mathrm{d}t$$

$$= 2M \int_0^1 (1-t^2)^{p_1} t^{p_2} \mathrm{d}t.$$

$\qquad\square$

**Lemma B.24.** *let $z \sim S^{d-2}$. Then,*

$$\mathbb{E}_z[z_1^{2k}] = \frac{\mathbb{E}_{\mu \sim N(0,1)}[\mu^{2k}]}{\mathbb{E}_{\nu \sim \chi(d-1)}[\nu^{2k}]} =: c_k(d) = \Theta(d^{-k}).$$

*Proof.* If $\mu \sim N(0, I_{d-1})$, we can equivalently write it as $\mu = \nu z$ with independent $\nu \sim \chi(d-1)$ and $z \sim S^{d-2}$. Thus, $\mu_1 = z_1 \nu$ and $z_1^{2k} = \mu_1^{2k}/\nu^{2k}$. The numerator is independent of $d$, and the denominator is of order $\Theta(d^k)$ (Lemma 44 of Damian et al. (2022)). $\qquad\square$

**Lemma B.25.** *When $h(z) = \frac{1}{\gamma_s} \log(1 + e^{\gamma_s z})$, with high probability,*

$$\sup_{\tilde{x} \in S^{d-1}} \frac{1}{J} \sum_j h''(\langle w_j, \tilde{x} \rangle) = \tilde{O}\left( d^{1/2} + \left( \sqrt{\frac{\gamma_s d^{3/2}}{J}} + \frac{\gamma_s d}{J} \right) \right).$$

*Proof.* Let $t = \langle w, \tilde{x} \rangle$. When $w \sim U(S^{d-1})$, $t$ follows the probability distribution

$$p_d(t) = C_d(1 - t^2)^{\frac{d-3}{2}}, \quad C_d = \frac{\Gamma(d/2)}{\sqrt{\pi}\Gamma((d-1)/2)}.$$

Note that $\|h''(t)\|_\infty = \gamma_s/4$. Moreover,

$$\mu_\gamma := \mathbb{E}_w[h''(\langle w, \tilde{x} \rangle)] = \int_{-1}^1 h''(t)p_d(t)\mathrm{d}t \le C_d \int_{-1}^1 h''(t)\mathrm{d}t = C_d(h'(1) - h'(-1)) = C_d \tanh(\gamma_s/2) \le C_d,$$

and

$$m_\gamma := \mathbb{E}_w[h''(\langle w, \tilde{x} \rangle)^2] = \int_{-1}^1 h''(t)^2 p_d(t)\mathrm{d}t \le C_d \int_{\mathbb{R}} h''(t)^2 \mathrm{d}t = C_d \frac{\gamma_s}{6},$$

where we used Lemma B.26 for the last equality. By Bernstein inequality, for any $z > 0$

$$\mathrm{P}\left(\left|\frac{1}{J}\sum_j h''(\langle w_j, \tilde{x} \rangle) - \mu_\gamma\right| \ge C\left(\sqrt{\frac{m_\gamma z}{J}} + \frac{\|h''(t)\|_\infty z}{J}\right)\right) \le 2\mathrm{e}^{-z}.$$

Furthermore, $h''(\langle w_j, \tilde{x} \rangle)$ is $O(\gamma_s^2)$-Lipschitz with respect to $\tilde{x}$, which implies its mean is also $O(\gamma_s^2)$-Lipschitz. Consider now the $\epsilon$-net $\mathcal{N}_\epsilon$ over $\tilde{x}$, then for all $\tilde{x} \in S^{d-1}$, by definition, there exists a $\hat{x} \in \mathcal{N}$ such that

$$\frac{1}{J}\sum_j h''(\langle w_j, \tilde{x} \rangle) \le \frac{1}{J}\sum_j h''(\langle w_j, \hat{x} \rangle) + \left|\frac{1}{J}\sum_j h''(\langle w_j, \tilde{x} \rangle) - \frac{1}{J}\sum_j h''(\langle w_j, \hat{x} \rangle)\right|$$

$$\le \sup_{\tilde{x} \in \mathcal{N}_\epsilon} \frac{1}{J}\sum_j h''(\langle w_j, \hat{x} \rangle) + O(\gamma_s^2 \epsilon).$$

By union bound over $|\mathcal{N}_\epsilon| \le \mathrm{e}^{Cd\log(J/\epsilon)}$, with probability $1 - 2\mathrm{e}^{Cd\log(J/\epsilon)-z}$,

$$\sup_{\tilde{x} \in \mathcal{N}_\epsilon} \frac{1}{J}\sum_j h''(\langle w_j, \hat{x} \rangle) \le \mu_\gamma + C\left(\sqrt{\frac{m_\gamma z}{J}} + \frac{\|h''(t)\|_\infty z}{J}\right).$$

By setting $z = Cd\log(J/\epsilon) + \tilde{O}(1)$, we obtain with high probability

$$\sup_{\tilde{x} \in \mathcal{N}_\epsilon} \left|\frac{1}{J}\sum_j h''(\langle w_j, \hat{x} \rangle)\right| \lesssim d^{1/2} + C\left(\sqrt{\frac{\gamma_s d^{3/2}\log(J/\epsilon)}{J}} + \frac{\gamma_s d\log(J/\epsilon)}{J}\right),$$

where we also used $C_d \sim d^{1/2}$. Finally, $\epsilon = \gamma_s^{-2}\sqrt{\frac{d}{J}}$ gives with high probability

$$\sup_{\tilde{x} \in S^{d-1}} \frac{1}{J}\sum_j h''(\langle w_j, \tilde{x} \rangle) = \tilde{O}\left(d^{1/2} + \left(\sqrt{\frac{\gamma_s d^{3/2}}{J}} + \frac{\gamma_s d}{J}\right)\right).$$

$\square$

**Lemma B.26.** *When $h(z) = \frac{1}{\gamma_s}\log(1 + \mathrm{e}^{\gamma_s z})$,*

$$\int_{\mathbb{R}} h''(t)^2 \mathrm{d}t = \frac{\gamma_s}{6}.$$

*Proof.* First, since $h'(t) = \frac{\mathrm{e}^{\gamma_s t}}{1+\mathrm{e}^{\gamma_s t}}$

$$h''(t) = \gamma_s \frac{\mathrm{e}^{\gamma_s t}}{(1 + \mathrm{e}^{\gamma_s t})^2} = \frac{\gamma_s}{4}\left(2\frac{\mathrm{e}^{\gamma_s t/2}}{1 + \mathrm{e}^{\gamma_s t}}\right)^2 = \frac{\gamma_s}{4}\mathrm{sech}^2\left(\frac{\gamma_s t}{2}\right).$$

The integral becomes

$$\int_{\mathbb{R}} h''(t)^2 \mathrm{d}t = \frac{\gamma_s^2}{16} \int_{\mathbb{R}} \mathrm{sech}^4\left(\frac{\gamma_s t}{2}\right) \mathrm{d}t = \frac{\gamma_s}{8} \int_{\mathbb{R}} \mathrm{sech}^4(t) \, \mathrm{d}t.$$

By substitution $u = \tanh(t)$, $\mathrm{d}u = \mathrm{sech}^2(t)\mathrm{d}t$ and $\mathrm{sech}^2(t) = 1 - \tanh(t)^2 = 1 - u^2$, which leads to

$$\int_{\mathbb{R}} h''(t)^2 \mathrm{d}t = \frac{\gamma_s}{8} \int_{-1}^{1} 1 - u^2 \mathrm{d}u = \frac{\gamma_s}{6}.$$

$\square$

## B.4. $t = 1$ Retraining

Now, by retraining according to the teacher training with the distilled dataset $\mathcal{D}_1^S = \{(\tilde{x}^{(1)}, \tilde{y}^{(0)})\}$ from Theorem B.4, $\theta^{(1)} \leftarrow \mathcal{A}_1^R(\theta^{(0)}, \mathcal{L}(\theta^{(0)}, \mathcal{D}_1^S), \xi_1^R, \eta_1^R, \lambda_1^R)$ becomes as follows. We set $\eta_1^D = \tilde{\Theta}(\sqrt{d})$, $\lambda_1^D = (\eta_1^D)^{-1}$, and $\lambda_1^R = (\eta_1^R)^{-1}$

**Lemma B.27.** *Under the Assumptions of Theorem B.4, with the synthetic data $\mathcal{D}_1^S$ from the distillation phase at $t = 1$, we obtain the following parameter at the end of this first step (i.e., after the retraining phase of $t = 1$):*

$$W^{(1)} = W^{(0)} - \eta_1^R\left\{\nabla_W \mathcal{L}^S(\theta^{(0)}) + \lambda_1^R W^{(0)}\right\} = -\eta_1^R \tilde{y}^{(0)} \tilde{x}^{(1)} \left(a \odot \sigma'(W^{(0)\top} \tilde{x}^{(1)})\right)^\top.$$

*Index-wise, this means*

$$w_i^{(1)} = a_i \sigma'(\langle w_i^{(0)}, \tilde{x}^{(1)}\rangle) g(x),$$

*where $g(x) = -\eta_1^R \tilde{y}^{(0)} \tilde{x}^{(1)} = -\eta_1^R \eta_1^D (\tilde{y}^{(0)})^2 \left(c_d \langle \beta, \tilde{x}^{(0)} \rangle \beta + \tilde{O}\left(d^{\frac{1}{2}} N^{-\frac{1}{2}} + d^{-2} + d^{\frac{1}{2}} J^{*-\frac{1}{2}}\right)\right)$. Note that $a$ (the second layer of the neural network) remains unchanged.*

*Remark B.28.* Since $\sigma'(z) = \mathbb{1}_{z>0}$, we may have in expectation $L/2$ weights $w_j^{(1)}$ that are just 0. In the following analysis, we will treat neural networks with an effective width $L^*$, denoting the number of non-zero weights after $t = 1$. When $L = \tilde{\Theta}(L^*)$, this is satisfied with high probability.[5]

## B.5. $t = 2$ Teacher Training

### B.5.1. PROBLEM FORMULATION AND RESULT

We will now focus on the second step of our algorithm. Note that by the progressive nature of Algorithm 1, $W_1$ is now fixed (see Lemma B.27) and the training phase will concentrate on the second layer, which means the distillation algorithm will also distill information of the second layer. After resetting the biases (see Assumption 3.7), teacher training runs ridge regression on the last layer (see Assumption 3.6). As a result,

$$I_j^{Tr} = \{\theta_j^{(2)}, G_j^{(2)}\} = \mathcal{A}_2^{Tr}(\theta_j^{(1)}, \mathcal{L}^{Tr}(\theta_j^{(1)}), \eta_2^{Tr}, \lambda_2^{Tr}).$$

becomes

$$a^{(\tau)} = a^{(\tau-1)} - \eta_2^{Tr}\left\{\frac{1}{N} K(K^\top a^{(\tau-1)} - y) + \lambda_2^{Tr} a^{(\tau-1)}\right\}, \quad a^{(0)} = a_j^{(0)},$$

and we output the $\xi_2^{Tr}$-th step of the ridge regression with kernel $K$ where $(K)_{in} = \sigma(\langle w_i^{(1)}, x_n\rangle + b_i^{(0)})$. Gradient information is

$$G_j^{(2)} = \{g_{\tau,j}^{Tr}\}_{\tau=0}^{\xi_2^{Tr}-1} = \left\{\frac{1}{N} K(K^\top a^{(\tau)} - y) + \lambda_2^{Tr} a^{(\tau)}\right\}_{\tau=0}^{\xi_2^{Tr}-1}.$$

Interestingly, one distilled data point is enough so that this $a^{(\xi_2^{Tr})}$ is a good solution for the problem in question. We fix $j = 0$.

**Theorem B.29.** *Under the assumptions of Theorem B.4 and $\langle \beta, \tilde{x}^{(0)} \rangle$ is not too small with order $\tilde{\Theta}(d^{-1/2})$, $N \geq \tilde{\Omega}(d^4)$ $J^* \geq \tilde{\Omega}(d^4)$, $\eta_1^D = \tilde{\Theta}(\sqrt{d})$, and $\eta_1^R = \tilde{\Theta}(d)$ there exists $\lambda_2^{Tr}$ such that if $\eta_2^{Tr}$ is sufficiently small and $\xi_2^{Tr} =$*

---

[5]In our final bound, we thus replace $L^*$ with $L$.

$\tilde{\Theta}(\{\eta_2^{Tr}\lambda_2^{Tr}\}^{-1})$, *then the final iterate of the teacher training at $t = 2$ output a parameter $a^* = a^{(\xi_2^{Tr})}$ that satisfies with probability at least 0.99,*

$$\mathbb{E}_{x,y}[|f_{(a^{(\xi_2^{Tr})}, W^{(1)}, b^{(0)})}(x) - y|] - \zeta \leq \tilde{O}\left(\sqrt{\frac{d}{N}} + \frac{1}{\sqrt{L^*}} + \frac{1}{N^{1/4}}\right).$$

### B.5.2. Proof of Theorem B.29

**Lemma B.30** (Lemma 23 from Nishikawa et al. (2025) restated). *Suppose there exists $g(x)$ such that $g(x) = P\langle\beta, x\rangle + c(x)$, where $P = \tilde{\Theta}(1)$ and $c(x) = o_d(P\log^{-2p+2} d)$. Then, there exists $\pi(a, b)$ such that*

$$\left|\mathbb{E}_{a\sim\mathrm{Unif}\{\pm 1\}, b\sim N(0, I)}[\pi(a, b)\sigma(v \cdot g(x) + b)] - f^*(x)\right| = o_d(1),$$

*and $\sup_{a,b}|\pi(a, b)| = \tilde{O}(1)$.*

*Proof.* The proof follows from Lemma 23 of Nishikawa et al. (2025), and the fact that we can redefine $\pi(a, b)$ defined for $b \sim [-1, 1]$ to $\mathbb{1}_{|b|\leq 1}\frac{\sqrt{2\pi}}{2e^{-x^2/2}}\pi(a, b)$ for $b \sim N(0, I)$. $\qquad\square$

**Lemma B.31** (Lemma 24 from Nishikawa et al. (2025) restated). *Under the assumptions of Lemma B.30, there exists $a^{\ddagger} \in \mathbb{R}^m$ such that*

$$\left|\sum_{i=1}^{L^*} a_i^{\ddagger}\sigma(v_i \cdot g(x) + b_j) - f^*(x)\right| = O\left(L^{*-\frac{1}{2}}\right) + o_d(1),$$

*with high probability over $x \sim N(0, I_d)$, where $\|a^{\ddagger}\|^2 = \tilde{O}(1/L^*)$ with high probability.*

*Proof.* This also follows from Nishikawa et al. (2025) Lemma 24. $\qquad\square$

*Proof of Theorem B.29.* We will first evaluate conditions on $\eta_1^R$, $\eta_1^D$, $N$ and $J^* = LJ/2$ to satisfy conditions $P = \tilde{\Theta}(1)$ and $c(x) = o_d(P\log^{-2p+2} d)$ of Lemma B.30. From Lemma B.27, we know that $P = -\eta_1^R\eta_1^D(\tilde{y}^{(0)})^2 c_d\langle\beta, \tilde{x}^{(0)}\rangle$ and $c(x) = -\eta_1^R\eta_1^D(\tilde{y}^{(0)})^2\langle\epsilon, x\rangle$ where $\epsilon = \tilde{O}\left(d^{\frac{1}{2}}N^{-\frac{1}{2}} + d^{-2} + d^{\frac{1}{2}}J^{*-\frac{1}{2}}\right)$. If we set $\eta_1^D = \tilde{\Theta}(\sqrt{d})$ and $\eta_1^R = \tilde{\Theta}(d)$, the first condition is fulfilled since $\tilde{y}^{(0)}$ is constant and $c_d\langle\beta, \tilde{x}^{(0)}\rangle = \tilde{\Theta}(d^{-3/2})$ by assumption. Now, from Corollary B.15 and Lemma B.16, $\langle\epsilon, x\rangle$ can be decomposed into the $\tilde{O}\left(d^{\frac{1}{2}}N^{-\frac{1}{2}} + d^{-2} + d^{\frac{1}{2}}J^{*-\frac{1}{2}}\right)$ coefficients and the inner products $\langle\beta, x\rangle$, $\langle\beta_{\perp}, x\rangle$, and $\langle\tilde{x}, x\rangle$ which are of $O(\log d)$ with high probability. As a result,

$$c(x) = \tilde{O}\left(d^2 N^{-\frac{1}{2}} + d^{-\frac{1}{2}} + d^2 J^{*-\frac{1}{2}}\right),$$

it suffices that $N \geq \tilde{\Omega}(d^4)$ and $J^* \geq \tilde{\Omega}(d^4)$.

From Lemma B.31 and Lemma 26 from Damian et al. (2022), we know that with high probability there exists $a^{\ddagger}$ such that if $\theta = (a^{\ddagger}, W^{(1)}, b^{(0)})$,

$$\mathcal{L}^{Tr}(\theta) - \zeta^2 = O\left(1/L^* + 1/\sqrt{N}\right).$$

As result, by the equivalence between norm constrained linear regression and ridge regression, there exists $\lambda > 0$ such that if

$$a^{(\infty)} = \min_a \mathcal{L}\left((a, W^{(1)}, b^{(1)})\right) + \frac{\lambda}{2}\|a\|^2,$$

then

$$\mathcal{L}\left((a^{(\infty)}, W^{(1)}, b^{(1)})\right) \leq \mathcal{L}\left((a^{\ddagger}, W^{(1)}, b^{(1)})\right),$$

and

$$\left\|a^{(\infty)}\right\| \leq \|a^{\ddagger}\|.$$

By the same procedure as Lemma 14 by Damian et al. (2022), we can conclude that

$$\mathbb{E}_{x,y}[|f_{a^{(T)}, W^{(1)}}(x) - y|] - \zeta \leq \tilde{O}\left(\sqrt{\frac{d}{N}} + \frac{1}{\sqrt{L^*}} + \frac{1}{N^{1/4}}\right)$$

$\qquad\square$

**B.6.** $t = 2$ **Student Training**

Here, the student training for the distillation dataset $\{(\hat{x}_m^{(0)}, \hat{y}_m^{(0)})\}_{m=1}^{M_2}$ will be a one-step regression.

$$I_j^S = \mathcal{A}_2^D(\theta_j^{(1)}, \mathcal{L}^S(\theta_j^{(1)}), \xi_2^S, \eta_2^S, \lambda_2^S)$$

becomes

$$\tilde{a}_j^{(1)} = a_j^{(0)} - \frac{\eta_2^S}{M_2}\tilde{K}(\tilde{K}^\top a_j^{(0)} - \hat{y}^{(0)}) = \left(I - \frac{\eta_2^S}{M_2}\tilde{K}\tilde{K}^\top\right)a_j^{(0)} + \frac{\eta_2^S}{M_2}\tilde{K}\hat{y}^{(0)},$$

where $(\tilde{K})_{im} = \sigma(\langle w_i^{(1)}, \hat{x}_m^{(0)}\rangle + b_i^{(0)})$ and $\hat{y} = (\hat{y}_1^{(0)} \cdots \hat{y}_{M_2}^{(0)})^\top$. Gradient information for the $j$-th initialization is

$$\{g_{0,j}^S\} = \left\{\frac{1}{M_2}\tilde{K}(\tilde{K}^\top a_j^{(0)} - \hat{y}^{(0)})\right\}.$$

**B.7.** $t = 2$ **Distillation and Retraining**

Now, based on the teacher and student trainings above, we analyze the behavior of the distillation and retraining phases. These two phases are considered at the same time here. The initial state of the second dataset $\mathcal{D}_2^S$ is denoted as $\{\hat{x}_m^{(0)}, \hat{y}_m^{(0)}\}_{m=1}^{M_2}$.

B.7.1. ONE-STEP GRADIENT MATCHING

For the one-step gradient matching, $\hat{y}$ is updated as follows:

$$\hat{y}^{(1)} = \hat{y}^{(0)} - \eta_2^D \frac{1}{J}\sum_{j=1}^{J}\nabla_{\hat{y}}\left(1 - \left\langle\sum_{\tau=0}^{\xi_2^{Tr}-1} g_{\tau,j}^{Tr}, g_{0,j}^S\right\rangle\right), \tag{17}$$

where we set $\xi_2^D = 1$, and $\lambda_2^D = 0$. We assume $\hat{y}_i^{(0)} = \frac{1}{J}\sum_j f_{\theta_j^{(1)}}(\hat{x}_i^{(0)})$.

At the retraining, we consider the regression setting similarly to the teacher training with $a^{(0)} = 0$ and use the resulting $\mathcal{D}_2^S = \{\hat{x}_m^{(0)}, \hat{y}_m^{(1)}\}$.

$$\theta_j^{(2)} \leftarrow \mathcal{A}_2^R(\theta_j^{(1)}, \mathcal{L}^S(\theta_j^{(1)}), \xi_2^R, \eta_2^R, \lambda_2^R)$$

becomes,

$$\tilde{a}^{(\tau+1)} = \tilde{a}^{(\tau)} - \eta_2^R \nabla_a\left(\frac{1}{M_2}\|\tilde{K}^\top\tilde{a}^{(\tau)} - \hat{y}^{(1)}\|^2\right), \quad \tilde{a}^{(0)} = a^{(0)},$$

where $\lambda_2^R = 0$ and we output $\tilde{a}^{(\xi_2^R)}$. Indeed, this $\tilde{a}^{(\xi_2^R)}$ can reconstruct the generalization ability of the teacher.

**Theorem B.32.** *Based on the Assumptions of Theorem B.29, if $\{\hat{x}_m^{(0)}\}$ satisfies the regularity condition B.36, $\eta_2^D = M_2\eta_2^{Tr}$, then for a sufficiently small $\eta_2^R$ and $\xi_2^R = \tilde{\Theta}((\eta_2^R\sigma_{\min})^{-1})$, where $\sigma_{\min} > 0$ is the smallest eigenvalue of $\tilde{K}^\top\tilde{K}$, the one-step gradient matching (17) finds $M_2$ labels $\tilde{y}_m^{(1)}$ so that*

$$\mathbb{E}_{x,y}[|f_{(\tilde{a}^{(\xi_1^R)}, W^{(1)}, b^{(0)})}(x) - y|] - \zeta \leq \tilde{O}\left(\sqrt{\frac{d}{N}} + \frac{1}{\sqrt{L^*}} + \frac{1}{N^{1/4}}\right),$$

*The overall memory cost is $O(d + L)$.*

*Proof.* From the definition of $g_{t,j}^{Tr}$, with $T = \xi_2^{Tr}$,

$$a_j^{(T)} = a_j^{(T-1)} - \eta_2^{Tr}g_{T-1,j}^{Tr} = a_j^{(0)} - \eta_2^{Tr}\sum_{\tau=0}^{T-1}g_{\tau,j}^{Tr},$$

which implies

$$\sum_{\tau=0}^{T-1}g_{\tau,j}^{Tr} = -(a^{(T)} - a^{(0)})/\eta_2^{Tr}.$$

Consequently,

$$
\begin{aligned}
\hat{y}^{(1)} &= \hat{y}^{(0)} - \eta_2^D \nabla_{\hat{y}} \left\{ 1 - \frac{1}{J} \sum_{j=1}^{J} \langle -(a_j^{(T)} - a_j^{(0)})/\eta_2^{Tr}, g_t^S \rangle \right\} \\
&= \frac{1}{J} \sum_j \tilde{K}^\top a_j^{(0)} + \frac{\eta_2^D}{\eta_2^{Tr} M_2} \tilde{K}^\top \frac{1}{J} \sum_{j=1}^{J} (a_j^{(T)} - a_j^{(0)}) \\
&= \frac{1}{J} \sum_{j=1}^{J} \tilde{K}^\top a_j^{(T)}.
\end{aligned}
$$

Now, given our distilled dataset $\{\tilde{x}_i, \hat{y}_i^{(1)}\}$ on retraining, $a$ is updated following the teacher training which follows

$$
\tilde{a}^{(t+1)} = \tilde{a}^{(t)} - \eta_2^R \nabla_a \left\{ \|\tilde{K}^\top \tilde{a}^{(t)} - \hat{y}^{(1)}\|^2 \right\}, \quad \tilde{a}^{(0)} = 0.
$$

By the implicit bias of gradient descent on linear regression,

$$
\tilde{a}^{(\infty)} = (\tilde{K}^\top)^\dagger \hat{y}^{(1)},
$$

where $(K^\top)^\dagger$ is the Moore-Penrose pseudoinverse. Since by Lemma B.34, $a_j^{(T)}$ is in $\mathrm{col}(K)$, Lemma B.37 implies that $a_j^{(T)}$ is in $\mathrm{col}(\tilde{K})$ as well. Therefore, $\tilde{a}^{(\infty)} = (\tilde{K}^\top)^\dagger y^{(1)} = (\tilde{K}^\top)^\dagger \tilde{K}^\top \frac{1}{J} \sum_{j=1}^{J} a_j^{(T)} = \frac{1}{J} \sum_{j=1}^{J} a_j^{(T)}$. Note that we can make $a_j^{(T)}$ arbitrarily close to $a_j^{(\infty)} = a^{(\infty)}$ within $\xi_2^{Tr} = \tilde{\theta}((\eta_2^{Tr} \lambda_2^{Tr})^{-1})$ steps since each $a_j^{(T)}$ converges to the same limit and $a_j^{(0)}$ is bounded.

For a finite iteration, we can approximate $\tilde{a}^{(\infty)}$ by $\tilde{a}^{(\xi_2^R)}$ within an arbitrary accuracy with $\xi_2^R = \tilde{\Theta}((\eta_2^R \sigma_{\min})^{-1})$, where $\sigma_{\min} > 0$ is the smallest eigenvalue of $\tilde{K}\tilde{K}^\top$. Therefore, we can reconstruct an $\tilde{a}^{(\xi_2^R)}$ that is dominated by the same error as Theorem B.29. $\qquad\square$

### B.7.2. ONE-STEP PERFORMANCE MATCHING

For the one-step gradient matching, $\tilde{y}$ is updated as follows:

$$
\hat{y}^{(\tau+1)} = \hat{y}^{(\tau)} - \eta_2^D \nabla_{\tilde{y}} \left( \frac{1}{N} \|K\tilde{a}^{(1)} - y\|^2 + \frac{\lambda_2^D}{2} \|\hat{y}^{(\tau)}\|^2 \right), \quad (\tau = 0, \dots, \xi_2^D - 1).
$$

At the retraining, we consider the one-step regression setting following the performance matching incentive.

$$
\theta_j^{(2)} \leftarrow \mathcal{A}_2^R(\theta_j^{(1)}, \mathcal{L}^S(\theta_j^{(1)}), \xi_2^R, \eta_2^R, \lambda_2^R)
$$

becomes,

$$
\tilde{a}^{(1)} = a^{(0)} - \frac{\eta_2^R}{M_2} \tilde{K}(\tilde{K}^\top \tilde{a}^{(0)} - \hat{y}^{(\xi_2^D)}) = \left( I - \frac{\eta_2^R}{M_2} \tilde{K}\tilde{K}^\top \right) a^{(0)} + \frac{\eta_2^R}{M_2} \tilde{K}\hat{y}^{(\xi_2^D)},
$$

where $\lambda_2^R = 0$ and we output $\tilde{a}^{(1)}$ ($\xi_2^R = 1$). Indeed, this $\tilde{a}^{(1)}$ can reconstruct the generalization ability of the teacher. We fix $j = 0$.

**Theorem B.33.** *Based on the Assumptions of Theorem B.29, if $\{\hat{x}_m^{(0)}\}$ satisfies the regularity condition B.36, $\eta_2^S = \eta_2^R$, $\eta_2^D$ is sufficiently small and $\xi_2^D = \tilde{\Theta}((\eta_2^D \lambda_2^D)^{-1})$ then there exists a $\lambda_2^D$ such that one-step performance matching can find with high probability a distilled dataset at the second step of distillation with*

$$
\mathbb{E}_{x,y}[|f_{(\tilde{a}^{(1)}, W^{(1)}, b^{(0)})}(x) - y|] - \zeta \leq \tilde{O}\left( \sqrt{\frac{d}{N}} + \frac{1}{\sqrt{L^*}} + \frac{1}{N^{1/4}} \right),
$$

*where $\tilde{a}^{(1)}$ is the output of the retraining algorithm at $t = 2$ with $\tilde{a}^{(0)} = 0$. The overall memory usage is only $O(d + L)$.*

*Proof.* In the one-step performance matching, we run one gradient update and then minimize the training loss. That is,

$$a^{(1)} = a^{(0)} - \frac{\eta_2^S}{M_2}\tilde{K}(\tilde{K}^\top a^{(0)} - \hat{y}) = \left(I - \frac{\eta_2^S}{M_2}\tilde{K}\tilde{K}^\top\right)a^{(0)} + \frac{\eta_2^S}{M_2}\tilde{K}\hat{y} = \frac{\eta_2^S}{M_2}\tilde{K}\hat{y},$$

where $(\tilde{K})_{ij} = \sigma(\langle w_i^{(1)}, \tilde{x}_j \rangle + b_i)$. Since for the reference $a^{(0)}$ is set to 0, The training loss becomes

$$\mathcal{L}((a,^{(1)}, W^{(1)}, b^{(0)}), \mathcal{D}^{Tr}) = \frac{1}{N}\|f_{(a^{(1)}, W^{(1)}, b^{(0)})}(X) - y\|^2 = \frac{1}{N}\left\|K^\top\frac{\eta_2^S}{M_2}\tilde{K}\hat{y} - y\right\|^2,$$

where $(K)_{ij} = \sigma(\langle w_i^{(1)}, x_j \rangle + b_i)$.

From the training step of $t = 2$, we know there exists $a^{(\infty)}$ such that

$$\frac{1}{N}\|f_{(a^{(\infty)}, W^{(1)}, b^{(0)})}(X) - y\|^2 = \frac{1}{N}\|K^\top a^{(\infty)} - y\|^2 = \tilde{O}\left(1/L^* + 1/\sqrt{N}\right).$$

Similarly to Theorem B.29, if we can assure there exists a $\hat{y}^*$ such that with high probability

$$\frac{\eta_2^S}{M_2}\tilde{K}\hat{y}^* = a^{(\infty)}, \tag{18}$$

then there exists a $\lambda_2^D$ such that

$$\hat{y}^{(\infty)} = \operatorname{argmin}_{\hat{y}}\mathcal{L}^{Tr}((a^{(1)}, W^{(1)}, b^{(0)}), \mathcal{D}^{Tr}) + \frac{\lambda_2^D}{2}\|\hat{y}\|^2 = \operatorname{argmin}_{\hat{y}}\mathcal{L}\left(\left(\frac{\eta_2^S}{M_2}\tilde{K}\hat{y}, W^{(1)}, b^{(0)}\right), \mathcal{D}^{Tr}\right) + \frac{\lambda_2^D}{2}\|\hat{y}\|^2,$$

which satisfies with $\mathcal{L}^{Tr}(\cdot) := \mathcal{L}(\cdot, \mathcal{D}^{Tr})$

$$\mathcal{L}^{Tr}\left(\left(\frac{\eta_2^S}{M_2}\tilde{K}\hat{y}^{(\infty)}, W^{(1)}, b^{(0)}\right)\right) \le \mathcal{L}^{Tr}\left(\left(\frac{\eta_2^S}{M_2}\tilde{K}\hat{y}^*, W^{(1)}b^{(0)}\right)\right) = \mathcal{L}^{Tr}((a^{(\infty)}, W^{(1)}, b^{(0)})) = \tilde{O}\left(1/L^* + 1/\sqrt{N}\right).$$

It now suffices to consider the condition for Equation (18) to be satisfied. However, since we want

$$a \in \operatorname{span}\{\sigma(W^{(1)}\hat{x}_1 + b^{(0)}), \dots, \sigma(W^{(1)}\hat{x}_m + b^{(0)})\},$$

this is satisfied by Lemma B.37 and our assumption since $a^{(\infty)} \in \operatorname{col}(K)$ by Lemma B.34.

As a result, the one-step PM can find a $\hat{y}^{(\infty)}$ by solving its distillation loss

$$\hat{y}^{(\infty)} = \operatorname{argmin}_{\hat{y}}\mathcal{L}^{Tr}((a^{(1)}, W^{(1)}, b^{(0)})) + \frac{\lambda_2^D}{2}\|\hat{y}\|^2.$$

that satisfies an error of $\tilde{O}\left(1/L^* + 1/\sqrt{N}\right)$. Since we can approximate $\hat{y}^{(\infty)}$ by $\hat{y}^{(\xi_2^D)}$ within an arbitrary accuracy, by similarly proceeding as Theorem B.29, by retraining $a$ on $\{\hat{x}_m^{(0)}, \hat{y}_m^{(\xi_2^D)}\}$ for one step, we obtain a $\tilde{a}^{(1)}$ such that

$$\mathbb{E}_{x,y}[|f_{(\tilde{a}^{(1)}, W^{(1)}, b^{(0)})}(x) - y|] - \zeta \le \tilde{O}\left(\sqrt{\frac{d}{N}} + \frac{1}{\sqrt{L^*}} + \frac{1}{N^{1/4}}\right).$$

$\square$

### B.7.3. OTHER LEMMAS

**Lemma B.34.** *Consider the following ridge regression of* $a \in \mathbb{R}^m$ *for a given* $K \in \mathbb{R}^{m \times n}$, $b \in \mathbb{R}^n$ *and* $\lambda > 0$:

$$a^{(\infty)} = \operatorname{argmin}_a L(a) := \operatorname{argmin}_a\|K^\top a - y\|^2 + \frac{\lambda}{2}\|a\|^2.$$

*and the finite time iterate at time* $t = 0, 1, \dots$

$$a^{(t+1)} = a^{(t)} - \eta\nabla L(a^{(t)}), \quad a^{(0)} = 0.$$

*Then, both* $a^{(\infty)}$ *and* $a^{(t)}$ *are in the column space of* $K$.

*Proof.* Let us define $k_j$ $(j = 1, \ldots, n)$ as the columns of $K$ and $V := \text{span}\{k_1, \ldots, k_n\}$. If we define the orthogonal complement of $V$ as $V^\perp$, then $\forall a \in \mathbb{R}^m$,

$$a = v + u, \quad v \in V, u \in V^\perp.$$

Therefore,

$$\|K^\top a - y\|^2 + \frac{\lambda}{2}\|a\|^2 = \|K^\top(v+u) - y\|^2 + \frac{\lambda}{2}\|v+u\|^2 = \|K^\top v - y\|^2 + \frac{\lambda}{2}\|v\|^2 + \frac{\lambda}{2}\|u\|^2,$$

since $\langle u, v \rangle = 0 \,\forall v \in V, u \in V^\perp$. As a result, the minimization of the above objective function requires $u = 0$, which implies $a^{(\infty)} \in \text{col}(K)$.

Furthermore,

$$a^{(t+1)} = (1 - \eta\lambda)a^{(t)} - \eta K(K^\top a^{(t)} - y)$$

directly shows that if $a^{(t)} \in \text{col}(K)$, then $a^{(t+1)} \in \text{col}(K)$ since the second term is already in the column space of $K$. Now, as we set $a^{(0)} = 0$, we obtain the desired result by mathematical induction. $\qquad\square$

**Corollary B.35.** *If $K$ is a kernel with $(K)_{ij} = \sigma(\langle w_i^{(1)}, x_j \rangle + b_i)$, then for some $c \in \mathbb{R}^n$*

$$a_i^{(\infty)} = \sum_{j=1}^n c_j \sigma(\langle w_i^{(1)}, x_j \rangle + b_i).$$

## B.8. Construction of Initializations of $\mathcal{D}_2^S$

The sufficient condition for one-step GM and one-step PM to perfectly distill the information of the second layer as described in the above proofs is the following.

**Assumption B.36** (Regularity Condition)**.** The second distilled dataset $\mathcal{D}_2^S$ is initialized as $\{(\hat{x}_m^{(0)}, \hat{y}_m^{(0)})\}_{m=1}^{M_2}$ so that the kernel of $f_{\theta^{(1)}}$ after $t = 1$, $(\tilde{K})_{im} = \sigma(\langle w_i^{(1)}, \hat{x}_m^{(0)} \rangle + b_i^{(0)})$ has the maximum attainable rank, and its memory cost does not exceed $\tilde{\Theta}(r^2 d + L)$. When $D := \{i \mid w_i^{(1)} \neq 0\}$, the maximum attainable rank is $|D| + 1$ if there exists an $i_0 \in [L] \setminus D$ such that $b_{i_0} > 0$, and $|D|$ otherwise.[6]

**Lemma B.37.** *Consider the two kernels $K \in \mathbb{R}^{L \times N}$ and $\tilde{K} \in \mathbb{R}^{L \times M_2}$ $(L \geq 2)$ where $(K)_{in} = \sigma(\langle w_i^{(1)}, x_n \rangle + b_i)$, $(\tilde{K})_{im} = \sigma(\langle w_i^{(1)}, \hat{x}_m \rangle + b_i)$, and $w_i^{(1)} = ca_i \frac{1}{M_1} \sum_m \sigma'(\langle w_i^{(0)}, \tilde{x}_m^{(1)} \rangle)\tilde{x}_m^{(1)}$ ($c$ is a constant) with $a_i \sim \{-1, 1\}$ and $w_i^{(0)} \sim U(S^{d-1})$. If the regularity condition B.36 is satisfied and $a^* \in \text{col}(K)$, then $a^* \in \text{col}(\tilde{K})$.*

*Proof.* $a^* \in \text{col}(\tilde{K})$ is equivalent to stating that there is a $\tilde{y} \in \mathbb{R}^M$ such that

$$a_i^* = \sum_{m=1}^{M_2} \hat{y}_m \sigma(\langle w_i^{(1)}, \hat{x}_m \rangle + b_i), \quad \forall i = 1, \ldots, L. \tag{19}$$

We define the set $D = \{i \mid w_i^{(1)} \neq 0\}$. Then for $i \in D$, Equation (19) becomes

$$a_i^* = \sum_{m=1}^{M_2} \hat{y}_m \sigma(b_i) = \sigma(b_i) \sum_{m=1}^{M_2} \hat{y}_m.$$

Therefore, for Equation (19) to be satisfied, we need at least that $\alpha^* \in \text{span}\{\sigma(b)\}$ for the indices not in $D$. However, this follows from the assumption of $a^* \in \text{col}(K)$ as for $i \notin D$

$$a_i^* = \sum_{j=1}^n c_j \sigma(\langle w_i^{(1)}, x_j \rangle + b_i) = \sum_{j=1}^n c_j \sigma(b_i) = \sigma(b_i) \sum_{j=1}^n c_j.$$

---

[6]We may accept a memory cost up to $\tilde{\Theta}(r^2 d + \text{poly}(r)L)$. Since $d \gg r$ in our theory, this will still lead to smaller storage cost than the whole network.

Since we can rearrange $\tilde{K}$ such that the rows in $D$ are grouped to the beginning, Equation (19) can be simplified to the first $L^* := |D|$ rows $\tilde{K}_{[L^*]}$ (plus one constant non-zero row if such a row exists).

$$a_i^* = \sum_{m=1}^{M} \tilde{y}_m \sigma(\langle w_i^{(1)}, \tilde{x}_m \rangle + b_i) = \sum_{m=1}^{m} \tilde{y}_m \sigma(ca_i \sigma'(\langle w_i^{(0)}, \tilde{x}^{(1)} \rangle) \langle \tilde{x}^{(1)}, \tilde{x}_m \rangle + b_i), \quad \forall i = 1, \ldots, L^*. \tag{20}$$

It suffices to consider the full row rank condition of $\tilde{K}_{[L^*]}$. This is satisfied by the regularity condition. $\qquad \square$

While randomly sampling $\hat{x}_m^{(0)}$ from a continuous distribution may be a relatively safe strategy to achieve Assumption B.36, this would require too much memory storage. Here, we propose a construction with reasonable properties that works well in practice and, at the same time, provides low memory cost. We start from the observation that we can find one $v \in \mathbb{R}^d$ such that $\langle w_i^{(1)}, v \rangle \neq 0$ for all $i \in D = \{i \mid w_i^{(1)} \neq 0\}$. Then, $\sigma(\langle w_i^{(1)}, s_m v \rangle + b_i)$ can be written as $\sigma(c\alpha_i s_m + b_i)$ for $c \in \mathbb{R} \setminus \{0\}$, $\alpha \in \mathbb{R}^L$ and $b \in \mathbb{R}^L$. We provide a construction of $\{s_m\}$ so that $\{\hat{x}_m^{(0)}\} = \{s_m v\}$ provides some guaranteed behavior and achieves the best possible rank for this configuration. This can be stated as follows.

**Lemma B.38.** *For each $s \in \mathbb{R}$, we define $v(s) \in \mathbb{R}^L$ as*

$$v_i(s) = \sigma(c\,\alpha_i s + b_i), \quad i = 1, \ldots, L,$$

*where $c \in \mathbb{R} \setminus \{0\}$, $\alpha_i \in \mathbb{R}$ and $b \in \mathbb{R}^L$. For any finite set $S = \{s_1, \ldots, s_{M_2}\} \subset \mathbb{R}$ define $K(S) \in \mathbb{R}^{L \times M_2}$ by*

$$K(S) := [v(s_1) \cdots v(s_m)].$$

*Moreover, $D := \{i : \alpha_i \neq 0\}$, $L^* := |D|$, and assume there exists an $i_0$ with $\alpha_{i_0} = 0$ and $b_{i_0} > 0$. For each $i \in D$, we define the hinge $\tau_i := -\frac{b_i}{c\alpha_i}$, which are all different almost surely. Let us reorder them as $\{\tau_1 < \cdots < \tau_{L^*}\}$ and define the open intervals*

$$I_0 := (-\infty, \tau_1), \qquad I_k := (\tau_k, \tau_{k+1}) \ (k = 1, \ldots, L^* - 1), \qquad I_{L^*} := (\tau_{L^*}, \infty).$$

*Now, for each $k = 1, \ldots, L^* - 1$ we prepare two points*

$$s_k^{(1)} := \frac{3\tau_k + \tau_{k+1}}{4}, \quad s_k^{(2)} := \frac{\tau_k + 3\tau_{k+1}}{4},$$

*and two points in each unbounded interval*

$$C_{\text{left}} := \{\tau_1 - \bar{\tau}, \tau_1 - 2\bar{\tau}\}, \quad C_{\text{right}} := \{\tau_{L^*} + \bar{\tau}, \tau_{L^*} + 2\bar{\tau}\},$$

*where $\bar{\tau} := 1 + \max_{1 \leq j \leq L^*} |\tau_j|$. Let $C'$ be the union of all these points $\{s_1^{(1)}, s_1^{(2)}, \ldots, s_{L^*-1}^{(1)}, s_{L^*-1}^{(2)}\} \cup C_{\text{left}} \cup C_{\text{right}}$ and consider $\{s_m\} = C = C' \cup (-C')$. Then, for every finite $S \subset \mathbb{R}$, $\text{rank}(K(S)) \leq L^* + 1$. Moreover, the matrix $K(C)$ attains the maximal achievable rank*

$$\text{rank}(K(C)) = \max_{S:\text{finite}} \text{rank}(K(S)).$$

*Consequently, either $\text{rank}(K(C)) = L^* + 1$ (and rank $L^* + 1$ is attainable), or else $\text{rank}(K(C)) < L^* + 1$ (and rank $L^* + 1$ is impossible for this specific $(a, b, c)$). The memory cost of $C$ is $\Theta(L^*) = O(L)$ (or sufficiently $\Theta(L)$).*

*If $i_0$ does not exist, the whole statement is valid by replacing $L^* + 1$ by $L^*$.*

*Proof.* If $\alpha_i = 0$, then $v_i(s) = \sigma(b_i)$ is independent of $s$. As a result, all rows with $\alpha_i = 0$ span a subspace of dimension of 1, since there exist an index $i_0$ so that $\alpha_{i_0} = 0$ and $b_{i_0} > 0$., and $\text{rank}(K(S)) \leq L^* + 1$ for any $S$.

Fix an interval $I_k$. For any active index $i \in D$, the affine function $c\alpha_i s + b_i$ can change sign only at $s = t_i$, and by construction no hinge lies inside $I_k$. Therefore, on $I_k$, each coordinate $v_i(s) = \sigma(c\alpha_i s + b_i)$ is either identically 0 or equals the affine function $c\alpha_i s + b_i$. Hence, there exist vectors $u_k, w_k \in \mathbb{R}^L$ such that for all $s \in I_k$,

$$v(s) = u_k + s\, w_k.$$

*Table 2.* Rank rate and MSE reconstruction rate for single index model ($r = 1$) case with $d = 10$. Note that our two methods are the same when $r = 1$.

| | $L = 10$ | $L = 100$ | $L = 500$ | $L = 1000$ |
|---|---|---|---|---|
| Rank Rate (%) | $100.0 \pm 0.0$ | $100.0 \pm 0.0$ | $100.0 \pm 0.0$ | $99.6 \pm 0.4$ |
| MSE Reconstruction Rate (%) | $100.0 \pm 0.0$ | $100.0 \pm 0.0$ | $100.0 \pm 0.0$ | $100.0 \pm 0.0$ |

In other words, two points suffice to span the whole interval. Indeed, consider $s, s' \in I_k$ with $s \neq s'$. From the affine form above,

$$w_k = \frac{v(s') - v(s)}{s' - s}, \quad u_k = v(s) - s\, w_k.$$

As a result, for any $t \in I_k$,

$$v(t) = u_k + t w_k \in \text{span}\{v(s), v(s')\}.$$

This also implies that for $\mathcal{V} := \text{span}\{v(s) : s \in \mathbb{R}\}$ and $\mathcal{V}_C := \text{span}\{v(s) : s \in C\}$, $\mathcal{V} = \mathcal{V}_C$.

Finally, for any finite $S$, the columns of $K(S)$ are vectors $v(s)$ with $s \in S$, so $\text{col}(K(S)) \subseteq \mathcal{V}$, which means $\text{rank}(K(S)) \leq \dim(\mathcal{V})$. On the other hand, $\text{col}(K(C)) = \mathcal{V}_C = \mathcal{V}$, so $\text{rank}(K(C)) = \dim(\mathcal{V})$. Therefore $\text{rank}(K(C)) = \max_S \text{rank}(K(S))$. Since $\max_S \text{rank}(K(S)) \leq L^* + 1$ and $K(C)$ achieves this maximum, either $\text{rank}(K(C)) = L^* + 1$ or else rank $L^* + 1$ is impossible for the given $(a, b, c)$. $\qquad\square$

**Corollary B.39.** *The overall memory complexity of $C$ is $\Theta(L)$.*

Based on the above lemma, we propose two heuristic strategies for the choice of $v$.

- We prepare $M_1$ candidates of $v$ as $\{v_i\}_{i \in [M_1]} = \{\tilde{x}_m^{(1)}\}_{m \in [M_1]}$. In other words, we reuse distilled data of $\mathcal{D}_1^S$. This does not increase the memory cost. For each $v_i$, we apply Lemma B.38, to obtain a set of scalars customized for $v_i$ as $C_i = \{s_{k,i}\}$ and define $\{\hat{x}_m^{(0)}\} := \{s_{k,i} v_i\}$. The overall memory cost is $\Theta(M_1 L) = \Theta(\text{poly(r)L})$.

- A more compact choice is to compute one common $C = \{s_k\}$ for all $\{v_i\}_{m \in [M_1]} = \{\tilde{x}_m^{(1)}\}_{m \in [M_1]}$. Therefore, we apply Lemma B.38 to the representative choice of $\sum_{i \in [M_1]} v_i / M_1$, and define $\{\hat{x}_m^{(0)}\} := \{s_k v_i\}$. This requires only a memory complexity of $\Theta(L)$. We call this the *compact* construction.

We report experiments in Figure 3 and Tables 2, 3 and 4 to illustrate that the above constructions satisfy our regularity condition B.36, suggesting that compact distillation is also possible for the second phase of Algorithm 1. We show this by comparing 1) the rank of $\tilde{K}$, where $(\tilde{K})_{im} = \sigma(\langle w_i^{(1)}, \hat{x}_m^{(0)} \rangle + b_i)$ and $\{\hat{x}_m^{(0)}\}$ is defined following our proposed constructions, with the maximum attainable rank defined in the regularity condition B.36, and 2) test loss of $f_{\tilde{\theta}^*}$ (output of retraining at $t = 2$) with that of $f_{\theta^*}$ (output of teacher training at $t = 2$).[7] GM was used in the distillation phase of $t = 2$. Table 2 presents the result for the single index models with increasing width of the two-layer neural network. Furthermore, Figure 3 and Tables 3 and 4 share the behavior for multi-index models with $r = 3$ or $r = 10$ with increasing size of $\mathcal{D}_1^S$. The remaining experimental setup is exactly the same as Experiment 5.1.

We can observe that in both settings the suggested formations of $\{\hat{x}_m^{(0)}\}$ consistently achieve an almost 100% reproduction accuracy. Notably, the proposed compact construction behaves more stably, with a lower variability and a closer percentage around 100.

Finally, we provide below a construction that provably satisfies regularity condition B.36. For simplicity, we only consider indices in $D := \{i \mid w_i^{(1)} \neq 0\}$.

**Lemma B.40.** *If there exists a vector $v$ such that $\forall i \in [L]$*

$$\alpha_i := \langle w_i^{(1)}, v \rangle \neq 0,$$

---

[7]We compare these two outputs as our initialization of $\hat{x}^{(0)}$ should reconstruct the performance of the model with parameters $\theta^*$ as stated in Theorems B.32 and B.33.

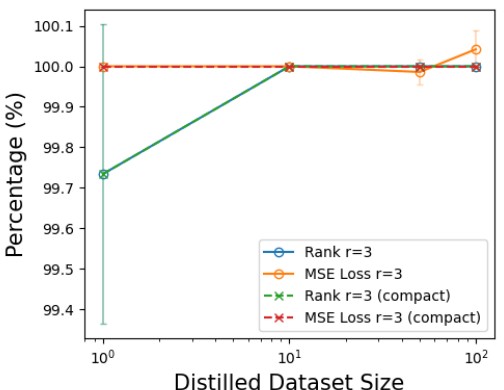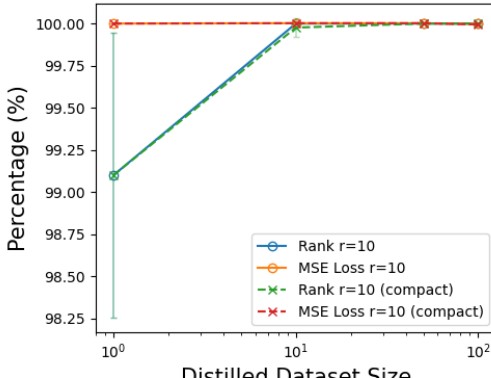

*Figure 3.* Reconstruction percentage when using $\mathcal{D}_2^S$ with different size $(1, 10, 50, 100)$ constructed following Lemma B.38 and its compact variant with respect to the MSE of teacher training $t = 2$ (MSE loss) and the maximal attainable rank $L^* + 1$ (Rank), for $r = 3$ (left) and $r = 10$ (right). d was set to 100.

*Table 3.* Numerical details of Figure 3 for the first construction of $\{\hat{x}_m^{(0)}\}$.

|  | $M_1 = 1$ | $L = 10$ | $L = 50$ | $L = 100$ |
|---|---|---|---|---|
| Rank Rate (%) | $99.7 \pm 0.4$ | $100.0 \pm 0.0$ | $100.0 \pm 0.0$ | $100.0 \pm 0.0$ |
| MSE Reconstruction Rate (%) | $100.0 \pm 0.0$ | $100.0 \pm 0.0$ | $100.0 \pm 0.0$ | $100.0 \pm 0.0$ |
| Rank Rate $r = 10$ (%) | $99.1 \pm 0.8$ | $100.0 \pm 0.0$ | $100.0 \pm 0.0$ | $100.0 \pm 0.0$ |
| MSE Reconstruction Rate $r = 10$ (%) | $100.0 \pm 0.0$ | $100.0 \pm 0.0$ | $100.0 \pm 0.0$ | $100.0 \pm 0.0$ |

*then, by setting*

$$\hat{x}_m^{(0)} = s_m v, \qquad m = 1, \ldots, 2(L + 2),$$

*for suitable scalars $s_m \in \mathbb{R}$, condition B.36 is satisfied almost surely.*

*Proof.* Since we restrict to points of the form $\hat{x}_m^{(0)} = s_m v$, we have

$$(\tilde{K})_{im} = \sigma(\alpha_i s_m + b_i), \qquad \alpha_i := \langle w_i^{(1)}, v \rangle \neq 0.$$

For $t \geq 0$, we define the vector

$$c(t) := \begin{pmatrix} \sigma(\alpha_1 t + b_1) \\ \vdots \\ \sigma(\alpha_L t + b_L) \end{pmatrix} \in \mathbb{R}^L,$$

and the paired-difference function

$$F(t) := c(t) - c(-t) = \begin{pmatrix} f_1(t) \\ \vdots \\ f_L(t) \end{pmatrix},$$

where

$$f_i(t) := \sigma(\alpha_i t + b_i) - \sigma(-\alpha_i t + b_i), \qquad t \geq 0.$$

Our goal is to choose values $t_0, \ldots, t_{L+1} \geq 0$ and then set

$$s_{2j+1} = t_j, \qquad s_{2j+2} = -t_j, \qquad j = 0, \ldots, L + 1.$$

If we do this, then the $j$-th paired column difference of $\tilde{K}$ is precisely

$$c(t_j) - c(-t_j) = F(t_j).$$

*Table 4.* Numerical details of Figure 3 for the second compact construction of $\{\hat{x}_m^{(0)}\}$.

| | $L = 10$ | $L = 100$ | $L = 500$ | $L = 1000$ |
|---|---|---|---|---|
| Rank Rate $r = 3$ (%) | $99.7 \pm 0.4$ | $100.0 \pm 0.0$ | $100.0 \pm 0.0$ | $100.0 \pm 0.0$ |
| MSE Reconstruction Rate $r = 3$ (%) | $100.0 \pm 0.0$ | $100.0 \pm 0.0$ | $100.0 \pm 0.0$ | $100.0 \pm 0.0$ |
| Rank Rate $r = 10$ (%) | $99.1 \pm 0.8$ | $100.0 \pm 0.0$ | $100.0 \pm 0.0$ | $100.0 \pm 0.0$ |
| MSE Reconstruction Rate $r = 10$ (%) | $100.0 \pm 0.0$ | $100.0 \pm 0.0$ | $100.0 \pm 0.0$ | $100.0 \pm 0.0$ |

Hence, if we can show that the vectors $F(t_0), \ldots, F(t_{L+1})$ span $\mathbb{R}^L$, then $\tilde{K}$ has row rank $L$. Now, we fix one $i$ and define $\rho_i := \left|\frac{b_i}{\alpha_i}\right|$. Since $b_i$ ($\forall i \in [L]$) are independent and have a continuous distribution, with probability one:

$$b_i \neq 0 \quad \text{for all } i,$$

and

$$\left|\frac{b_i}{\alpha_i}\right| \neq \left|\frac{b_j}{\alpha_j}\right| \qquad \text{for } i \neq j.$$

Hence, with probability one, the numbers $\rho_1, \ldots, \rho_L$ are all distinct. Reordering the rows if necessary (which does not change row rank), we may assume

$$0 < \rho_1 < \rho_2 < \cdots < \rho_L.$$

Now, for $t \geq 0$, $f$ can be reformulated as

$$f_i(t) = \beta_i t + \delta_i \sigma(t - \rho_i),$$

where

$$(\beta_i, \delta_i) = \begin{cases} (2\alpha_i, -\alpha_i), & b_i > 0, \\ (0, \alpha_i), & b_i < 0. \end{cases}$$

In particular,

$$\delta_i \neq 0 \qquad \text{and} \qquad \beta_i + \delta_i = \alpha_i.$$

Choose

$$t_0 \in (0, \rho_1), \qquad t_i = \rho_i \ (i = 1, \ldots, L), \qquad t_{L+1} > \rho_L.$$

Now, we define

$$s_{2j+1} = t_j, \qquad s_{2j+2} = -t_j, \qquad j = 0, \ldots, L+1.$$

This gives $M = 2(L+2)$ columns. Let

$$E_j := F(t_j) = c(t_j) - c(-t_j) \in \mathbb{R}^L, \qquad j = 0, \ldots, L+1,$$

and let

$$E := \begin{bmatrix} E_0 & E_1 & \cdots & E_{L+1} \end{bmatrix} \in \mathbb{R}^{L \times (L+2)}.$$

Since each $E_j$ is a linear combination of two columns of $\tilde{K}$, we have

$$\text{rank}(E) \leq \text{rank}(\tilde{K}).$$

Therefore it is enough to prove that $\text{rank}(E) = L$.

For $j = 1, \ldots, L+1$, we again define

$$S_j := \frac{E_j - E_{j-1}}{t_j - t_{j-1}} \in \mathbb{R}^L.$$

Since

$$f_i(t) = \beta_i t + \delta_i (t - \rho_i)_+,$$

$$(S_j)_i = \frac{f_i(t_j) - f_i(t_{j-1})}{t_j - t_{j-1}} = \beta_i + \delta_i \frac{(t_j - \rho_i)_+ - (t_{j-1} - \rho_i)_+}{t_j - t_{j-1}}.$$

Due to the ordering of the points $t_j$, if $j \leq i$, then $t_j \leq \rho_i$ and both

$$(t_j - \rho_i)_+ = 0, \qquad (t_{j-1} - \rho_i)_+ = 0.$$

As a result,

$$(S_j)_i = \beta_i.$$

If $j \geq i + 1$, then $t_{j-1} \geq \rho_i$, and both arguments are nonnegative, leading to

$$(t_j - \rho_i)_+ - (t_{j-1} - \rho_i)_+ = (t_j - \rho_i) - (t_{j-1} - \rho_i) = t_j - t_{j-1}.$$

Hence

$$(S_j)_i = \beta_i + \delta_i = \alpha_i.$$

Thus we have shown that

$$(S_j)_i = \begin{cases} \beta_i, & j \leq i, \\ \alpha_i, & j \geq i + 1. \end{cases}$$

Equivalently, for each fixed row $i$, the sequence

$$(S_1)_i, \ (S_2)_i, \ \ldots, \ (S_{L+1})_i$$

is constant equal to $\beta_i$ up to index $i$, and then constant equal to $\alpha_i$ from index $i + 1$ onward. So it has exactly one jump, occurring between $j = i$ and $j = i + 1$.

For $j = 1, \ldots, L$, by defining

$$B_j := S_{j+1} - S_j \in \mathbb{R}^L.$$

and

$$B := \begin{bmatrix} B_1 & B_2 & \cdots & B_L \end{bmatrix} \in \mathbb{R}^{L \times L}.$$

we again obtain a matrix obtained from the linear combinations of the columns of $E$. In other words,

$$\mathrm{rank}(B) \leq \mathrm{rank}(E).$$

For a fixed $i$, if $j < i$, then both $j$ and $j + 1$ satisfy $j + 1 \leq i$, hence

$$(S_j)_i = \beta_i, \qquad (S_{j+1})_i = \beta_i,$$

and

$$(B_j)_i = 0.$$

If $j = i$, then

$$(S_i)_i = \beta_i, \qquad (S_{i+1})_i = \alpha_i,$$

and

$$(B_i)_i = \alpha_i - \beta_i = \delta_i.$$

Finally, if $j > i$, then both $j$ and $j + 1$ are at least $i + 1$, hence

$$(S_j)_i = \alpha_i, \qquad (S_{j+1})_i = \alpha_i,$$

and

$$(B_j)_i = 0.$$

Combining the three cases,

$$(B_j)_i = \begin{cases} \delta_i, & j = i, \\ 0, & j \neq i, \end{cases}$$

which means that
$$B = \text{diag}(\delta_1, \dots, \delta_L).$$

Since each $\delta_i \neq 0$, the matrix $B$ is invertible, leading to

$$\text{rank}(B) = L.$$

This implies that

$$L = \text{rank}(B) \leq \text{rank}(E) \leq \text{rank}(\tilde{K}),$$

which proves our statement. $\qquad\square$

## C. Proof of Main Theorems: Single Index Models (for ReLU Activation Function)

### C.1. Well-defined Gradient Matching for ReLU

In this appendix, we consider a gradient matching algorithm that is well-defined for student gradient information formulated with ReLU activation function. This option is particularly motivated by the fact that practical frameworks such as PyTorch adopt a specific convention for the ReLU second derivative, which effectively makes DD well-defined in practice. Under this ReLU-defined update, we show that the same result as in the main paper continues to hold. In this sense, this also provides a possible explanation for why DD continues to work well empirically even when ReLU is used.

As ReLU is invariant to scaling, we can consider without loss of generality that all weights are normalized $\|w_i\| = 1$, which means that $w_i \sim U(S^{d-1})$. The idea is to look back at Lemma B.5 and define a novel update rule that is well-defined in the case $h = \sigma$ (where $\sigma =$ReLU). We only show the result for the single index model as the proof for multi-index model follows similarly.

**Definition C.1** (Well-defined Gradient Matching for ReLU). We defined the gradient update of $\tilde{x}^{(1)}$ as follows:

$$\tilde{x}^{(1)} = - \eta_1^D \tilde{y}^{(0)} \left( G + \epsilon \right),$$

where

$$G = \frac{1}{LJ} \sum_{i,j} g_{i,j} \sigma'(\langle w_{i,j}^{(0)}, \tilde{x} \rangle)$$

$$\epsilon = \frac{1}{LJ} \sum_{i,j} \left\{ \frac{1}{N} \sum_n \epsilon_n x_n \sigma'(\langle w_{i,j}^{(0)}, x_n \rangle) \right\} \sigma'(\langle w_{i,j}^{(0)}, \tilde{x} \rangle),$$

with $g_{i,j} = \frac{1}{N} \sum_n \hat{f}^*(x_n) x_n \sigma'(\langle w_{i,j}^{(0)}, x_n \rangle)$.

### C.2. Analysis

Since in Appendix B, we always treated $h'$ and $h''$ separately, we can prove all lemmas and theorems by analogy, substituting the case $h' = \sigma'$ is bounded by 1 and $h'' = 0$. The corresponding population gradient becomes as follows.

**Definition C.2.** We define the population gradient of $G$ as

$$\hat{G} := \mathbb{E}_w \left[ \mathbb{E}_x \left[ \hat{f}^*(x) x \sigma'(\langle w, x \rangle) \right] \sigma'(\langle w, \tilde{x} \rangle) \right],$$

where $\tilde{x} := \tilde{x}^{(0)}$.

This leads to the following bound.

**Theorem C.3.** *Under Assumptions 3.1, 3.3, 3.4 and 3.6, with high probability,*

$$\hat{G} = c_d \langle \beta, \tilde{x} \rangle \beta + \tilde{O} \left( d^{\frac{1}{2}} N^{-\frac{1}{2}} + d^{-2} \right),$$

*where $c_d = \Theta \left( d^{-1} \right)$ is a constant coefficient.*

*Proof.* This can be proved by reusing Lemma B.16. □

The concentration of the empirical gradient around the population gradient can be also computed similarly.

**Theorem C.4.** *Under Assumptions 3.1, 3.3, 3.4, 3.6, and 3.8 and Definition C.2, and Definition C.2,*

$$G = c_d \langle \beta, \tilde{x} \rangle \beta + \tilde{O}\left(d^{\frac{1}{2}} N^{-\frac{1}{2}} + d^{-2} + d^{\frac{1}{2}} J^{*-\frac{1}{2}}\right).$$

**Lemma C.5.** *Under Assumptions 3.1, 3.3, 3.4, 3.6, and 3.8 and Definition C.2, and Definition C.2, with high probability,*

$$\epsilon = \tilde{O}\left(d^{\frac{1}{2}} N^{-\frac{1}{2}}\right),$$

*where $\epsilon$ is defined in Corollary B.7.*

Therefore, this well-defined distillation also captures the latent structure of the original problem and translates it into the input space.

**Theorem C.6.** *Under Assumptions 3.1, 3.3, 3.4, 3.6, and 3.8 and Definition C.2, when $\mathcal{D}_0 = \{(\tilde{x}^{(0)}, \tilde{y}^{(0)})\}$, where $\tilde{x}^{(0)} \sim U(S^{d-1})$, the first step of distillation gives $\tilde{x}^{(1)}$ such that*

$$\tilde{x}^{(1)} = -\eta_1^D \tilde{y}^{(0)} \left( c_d \langle \beta, \tilde{x}^{(0)} \rangle \beta + \tilde{O}\left(d^{\frac{1}{2}} N^{-\frac{1}{2}} + d^{-2} + d^{\frac{1}{2}} J^{*-\frac{1}{2}}\right)\right),$$

*where $c_d = \tilde{O}\left(d^{-1}\right)$, $J^* = LJ/2$.*

The remainder of the analysis also follows Appendix B. Especially, for the teacher training at $t = 2$, the following theorem holds.

**Theorem C.7.** *Under the assumptions of Theorem B.4, $\langle \beta, \tilde{x}^{(0)} \rangle$ is not too small with order $\tilde{\Theta}(d^{-1/2})$, $N \geq \tilde{\Omega}(d^4)$ and $J^* \geq \tilde{\Omega}(d^4)$, there exists $\lambda_2^{Tr}$ such that if $\eta_2^{Tr}$ is sufficiently small and $\xi_2^{Tr} = \tilde{\Theta}(\{\eta_2^{Tr} \lambda_2^{Tr}\}^{-1})$ so that the final iterate of the teacher training at $t = 2$ output a parameter $a^* = a^{(\xi_2^{Tr})}$ that satisfies with probability at least 0.99,*

$$\mathbb{E}_{x,y}[|f_{(a^{(\xi_2^{Tr})}, W^{(1)}, b^{(1)})}(x) - y|] - \zeta \leq \tilde{O}\left(\sqrt{\frac{d}{N}} + \frac{1}{\sqrt{L^*}} + \frac{1}{N^{1/4}}\right).$$

We defer the readers to Appendix B.7 for the remainder of the proof, which does not change for this appendix.

## D. Proof of Main Theorems: Multi-index Models (General Case)

In this appendix, we prove our main result for multi-index models. As ReLU is invariant to scaling, we can consider without loss of generality that all weights are normalized $\|w_i\| = 1$, which means that $w_i \sim U(S^{d-1})$. Throughout this section, we will repeatedly refer to statements and proofs of Appendix B to keep the presentation clear and avoid redundancy.

### D.1. Proof Flow

The proof flow mainly follows that of Appendix B and is constituted of three parts: the distillation at $t = 1$, the teacher learning at $t = 2$ and the distillation at $t = 2$. The remainder of the analysis is the same as Appendix B.7.

### D.2. Concrete Formulation of Theorem 4.2

We first present our main Theorem for this appendix which is the analogue of Theorem B.29 in Appendix B.

**Theorem D.1.** *Under Assumptions 3.1, 3.3, 3.4, 3.6, 3.7 and 3.8, with parameters $\eta_1^D = \tilde{\Theta}(\sqrt{d})$, $\eta_1^R = \tilde{\Theta}(r^{-1}\sqrt{d})$, $\lambda_1^R = 1/\eta_1^R$, $\lambda_1^D = 1/\eta_1^D$, $M \geq \tilde{\Omega}(\hat{\kappa}_p^2 \lambda_{\min}^2 \lambda_{\max}^2 r^2)$, $d \geq \tilde{\Omega}(\hat{r}_p^2 r^2 \vee r^3 \lambda_{\min}^2 \hat{\kappa}_p^2)$, $N \geq \tilde{\Omega}(\hat{r}_p^2 d^4 \vee r \lambda_{\min}^2 \hat{\kappa}_p^2 d^4 \vee d^4)$, $J^* \geq \tilde{\Omega}(\hat{r}_p^2 d^4 \vee r \lambda_{\min}^2 \hat{\kappa}_p^2 d^4 \vee d^4)$, where $\hat{r}_p = r^{4p+1/2} \hat{\kappa}_p^{4p+1} \lambda_{\min}^{4p+1} \lambda_{\max}^{8p+1}$ and $\hat{\kappa}_p = \max_{1 \leq i \leq 4k, 1 \leq k \leq p} \kappa^{k/i}$, there exists $\lambda_2^{Tr}$ such that if $\eta_2^{Tr}$ is sufficiently small and $T = \tilde{\Theta}(\{\eta_2^{Tr} \lambda_2^{Tr}\}^{-1})$ so that the final iterate of the teacher training at $t = 2$ output a parameter $a^{(\xi_2^{Tr})}$ that satisfies with probability at least 0.99,*

$$\mathbb{E}_{x,y}[|f_{(a^{(\xi_2^{Tr})}, W^{(1)}, b^{(1)})}(x) - y|] - \zeta \leq \tilde{O}\left(\sqrt{\frac{d r^{3p} \kappa^{2p}}{N}} + \sqrt{\frac{r^{3p} \kappa^{2p}}{L^*}} + \frac{1}{N^{1/4}}\right).$$

**D.3.** $t = 1$ **Distillation**

Based on our results for single index models, we can directly start from the formulation of population gradient which we remind below. The main difference lies in the number of $M_1$ required, but this only comes into play in Section D.4. In this section, we thus only consider the behavior of one $\tilde{x}^{(0)}$.

**Definition D.2.** We define the population gradient of $G$ as

$$\hat{G} := \mathbb{E}_w \left[ \mathbb{E}_x \left[ \hat{f}^*(x) x \sigma'(\langle w, x \rangle) \right] h'(\langle w, \tilde{x} \rangle) \right] + \mathbb{E}_w \left[ w h''(\langle w, \tilde{x} \rangle) \left\langle \mathbb{E}_x \left[ \hat{f}^*(x) x \sigma'(\langle w, x \rangle) \right], \tilde{x} \right\rangle \right],$$

where $\tilde{x} := \tilde{x}_m^{(0)}$.

By Lemma B.14, this can be reformulated as follows:

**Corollary D.3.** *Under Assumption 3.1, with high probability,*

$$\hat{G} = \sum_{k=1}^{p-1} \frac{c_{k+1}}{k!} \mathbb{E}_w \left[ \hat{C}_{k+1}(w^{\otimes k}) h'(\langle w, \tilde{x} \rangle) \right] + \sum_{k=2}^{p} \frac{c_{k+2}}{k!} \mathbb{E}_w \left[ w \hat{C}_k(w^{\otimes k}) h'(\langle w, \tilde{x} \rangle) \right]$$

$$+ \sum_{k=1}^{p-1} \frac{c_{k+1}}{k!} \mathbb{E}_w \left[ w h''(\langle w, \tilde{x} \rangle) \left\langle \hat{C}_{k+1}(w^{\otimes k}), \tilde{x} \right\rangle \right] + \sum_{k=2}^{p} \frac{c_{k+2}}{k!} \mathbb{E}_w \left[ w h''(\langle w, \tilde{x} \rangle) \hat{C}_k(w^{\otimes k}) \langle w, \tilde{x} \rangle \right]$$

$$+ \tilde{O} \left( \sqrt{\frac{d}{N}} \right).$$

Let us now compute each expectation. We define each term as follows:

$$T_1^{(k)} := \mathbb{E}_w \left[ \hat{C}_{k+1}(w^{\otimes k}) h'(\langle w, \tilde{x} \rangle) \right],$$

$$T_2^{(k)} := \mathbb{E}_w \left[ w \hat{C}_k(w^{\otimes k}) h'(\langle w, \tilde{x} \rangle) \right],$$

$$T_3^{(k)} := \mathbb{E}_w \left[ w h''(\langle w, \tilde{x} \rangle) \langle \hat{C}_{k+1}(w^{\otimes k}), \tilde{x} \rangle \right],$$

$$T_4^{(k)} := \mathbb{E}_w \left[ w h''(\langle w, \tilde{x} \rangle) \hat{C}_k(w^{\otimes k}) \langle w, \tilde{x} \rangle \right].$$

**Lemma D.4.** *Under Assumptions 3.1,*

$$T_1^{(k)} = B \sum_{l=0}^{\lfloor k/2 \rfloor} \binom{k}{2l} c_l(d) C_{k+1}(S_{k,l}) \mathcal{I}_{k-2l,l}[h'],$$

$$T_2^{(k)} = \tilde{x} \sum_{l=0}^{\lfloor k/2 \rfloor} \binom{k}{2l} c_l(d) C_k(S_{k,l}) \mathcal{I}_{k-2l+1,l}[h'] + B_\perp \sum_{l=0}^{\lfloor (k-1)/2 \rfloor} \binom{k}{2l+1} C_l c_{l+1}(d) C_k(\bar{S}_{k,l}) \mathcal{I}_{k-2l-1,l+1}[h'],$$

$$T_3^{(k)} = \tilde{x} \sum_{l=0}^{\lfloor k/2 \rfloor} \binom{k}{2l} c_l(d) \tilde{C}_{k+1}(S_{k,l}) \mathcal{I}_{k-2l+1,l}[h''] + B_\perp \sum_{l=0}^{\lfloor (k-1)/2 \rfloor} \binom{k}{2l+1} C_l c_{l+1}(d) \tilde{C}_{k+1}(\bar{S}_{k,l}) \mathcal{I}_{k-2l-1,l+1}[h''],$$

$$T_4^{(k)} = \tilde{x} \sum_{l=0}^{\lfloor k/2 \rfloor} \binom{k}{2l} c_l(d) C_k(S_{k,l}) \mathcal{I}_{k-2l+2,l}[h''] + B_\perp \sum_{l=0}^{\lfloor (k-1)/2 \rfloor} \binom{k}{2l+1} C_l c_{l+1}(d) C_k(\bar{S}_{k,l}) \mathcal{I}_{k-2l,l+1}[h''],$$

*where* $B_\perp = PB$, $P = I_d - \tilde{x}\tilde{x}^\top$, $\mathcal{I}_{p_1,p_2}[i] = \int_{-1}^{1} t^{p_1}(1-t^2)^{p_2} i(t) f_d(t) \mathrm{d}t$, $f_d(t) = \frac{\Gamma(\frac{d}{2})}{\sqrt{\pi}\Gamma(\frac{d-1}{2})}(1-t^2)^{\frac{d-3}{2}}$, $S_{k,l} :=$ Sym$((B^\top \tilde{x})^{\otimes k-2l} \otimes \Sigma^{\otimes l})$ *with* $\hat{v} \sim N(0, \Sigma)$, $\bar{S}_{k,l} :=$ Sym$((B^\top \tilde{x})^{\otimes k-2l-1} \otimes \Sigma^{\otimes l})$, $\tilde{C}_{k+1} = C_{k+1}(b)$ *and* $\Sigma = B^\top(I - \tilde{x}\tilde{x}^\top)B$.

*Proof.* In this proof, we simplify the notation by treating $T_i^{(k)}$ as $T_i$ ($i = 1, 2, 3, 4$).

Similarly to the single index case, we use a change of variable. Let $t = \langle w, \tilde{x} \rangle$, then we can write $w = t\tilde{x} + \sqrt{1-t^2}v$ where $v \sim U(\{v \mid v \in S^{d-1}, \langle v, \tilde{x} \rangle = 0\}) \cong U(S^{d-2})$. Since $\|\tilde{x}\|^2 = 1$, the distribution of $t$ is can be shown to be

$$f_d(t) = \frac{\Gamma(\frac{d}{2})}{\sqrt{\pi}\Gamma(\frac{d-1}{2})}(1-t^2)^{\frac{d-3}{2}}.$$

Moreover,

$$\langle \beta, w \rangle = st + \sqrt{1-t^2}\langle \beta_\perp, v \rangle,$$

where $s = \langle \beta, \tilde{x} \rangle$, and $\beta_\perp = \beta - \langle \beta, \tilde{x} \rangle \tilde{x}$. We also define $a := B^\top w = bt + \sqrt{1-t^2}\eta$, where $b := B^\top \tilde{x}$ and $\eta := B^\top v$, $P := I_d - \tilde{x}\tilde{x}^\top$, $B_\perp := PB$, $\Sigma := B^\top PB = I_r - bb^\top$. By Lemma B.11, since $\hat{C}_k = B^{\otimes k}C_k$, $\hat{C}_k(w^{\otimes k}) = C_k(a^{\otimes k})$ and $\hat{C}_{k+1}(w^{\otimes k}) = BC_{k+1}(a^{\otimes k})$. Now, let us first consider $T_1 = \mathbb{E}_{t,v}\left[BC_{k+1}(a^{\otimes k})h'(t)\right]$. Since $C_{k+1}$ is a symmetric tensor,

$$C_{k+1}(a^{\otimes k}) = C_{k+1}(\text{Sym}(a^{\otimes k}))$$

$$= C_{k+1}\left(\sum_{l=0}^{k}\binom{k}{l}t^{k-l}(1-t^2)^{l/2}\text{Sym}(b^{\otimes k-l}\otimes\eta^{\otimes l})\right).$$

Therefore, by taking the expectation over $v$ and by symmetry,

$$\mathbb{E}_v[C_{k+1}(a^{\otimes k}) \mid t] = C_{k+1}\left(\sum_{l=0}^{\lfloor k/2\rfloor}\binom{k}{2l}t^{k-2l}(1-t^2)^l\text{Sym}(b^{\otimes k-2l}\otimes\mathbb{E}_v[\eta^{\otimes 2l}])\right).$$

$\mathbb{E}_v[\eta^{\otimes 2l}]$ can be computed by using Lemma 45 from Damian et al. (2022) as

$$\mathbb{E}_v[\eta^{\otimes 2l}] = \mathbb{E}_v[(B^\top v)^{\otimes 2l}] = (B^\top)^{\otimes 2l}\mathbb{E}_v[v^{\otimes 2l}] = c_l(d)(B^\top)^{\otimes 2l}\mathbb{E}_g[(Pg)^{\otimes 2l}] = c_l(d)\mathbb{E}_{\hat{v}}[\hat{v}^{\otimes 2l}] =: c_l(d)S_l,$$

where we used that fact that $v = Pg/\nu$ with $\nu \sim \chi(d)$ and $g \sim N(0, I_d)$ and defined $\hat{v} \sim N(0, \Sigma)$. To summarize, we obtain,

$$T_1 = B\sum_{l=0}^{\lfloor k/2\rfloor}\binom{k}{2l}c_l(d)C_{k+1}(S_{k,l})\int_{-1}^{1}t^{k-2l}(1-t^2)^lh'(t)f_d(t)\mathrm{d}t,$$

where $S_{k,l} := \text{Sym}(b^{\otimes k-2l}\otimes S_l)$.

Next, by proceeding similarly, $T_2 = \mathbb{E}_{t,v}\left[(t\tilde{x} + \sqrt{1-t^2}v)C_k(a^{\otimes k})h'(t)\right]$. We can divide this into two terms where

$$T_{2,1} := \mathbb{E}_{t,v}\left[tC_k(a^{\otimes k})h'(t)\right]\tilde{x},$$
$$T_{2,2} := \mathbb{E}_{t,v}\left[\sqrt{1-t^2}vC_k(a^{\otimes k})h'(t)\right].$$

By analogy from $T_1$, we immediately obtain

$$T_{2,1} = \tilde{x}\sum_{l=0}^{\lfloor k/2\rfloor}\binom{k}{2l}c_l(d)C_k(S_{k,l})\int_{-1}^{1}t^{k-2l+1}(1-t^2)^lh'(t)f_d(t)\mathrm{d}t.$$

We now focus on $T_{2,2}$.

$$\mathbb{E}_v\left[vC_k(a^{\otimes k})\right] = \sum_{l=0}^{\lfloor (k-1)/2\rfloor}\binom{k}{2l+1}t^{k-2l-1}(1-t^2)^{(2l+1)/2}\mathbb{E}_v\left[vC_k\left(\text{Sym}(b^{\otimes k-2l-1}\otimes\eta^{\otimes 2l+1})\right)\right].$$

We define the tensor $V_l := C_k\left(\text{Sym}(b^{\otimes k-2l-1})\right)$. Since $\mathbb{E}_v[vV_l(\eta^{\otimes 2l+1})] = \mathbb{E}_v[v\otimes\eta^{\otimes 2l+1}](V_l)$, we obtain

$$\mathbb{E}_v\left[vC_k(a^{\otimes k})\right] = \sum_{l=0}^{\lfloor (k-1)/2\rfloor}\binom{k}{2l+1}t^{k-2l-1}(1-t^2)^{(2l+1)/2}\mathbb{E}_v[v\otimes\eta^{\otimes 2l+1}](V_l).$$

Let $a \in [d]$ and $i_1, \ldots, i_{2l+1} \in [r]$. Put $j_0 = a$ and $j_t = b_t$ for $t \geq 1$. Using $\eta_i = \sum_{b=1}^d B_{bi} v_b$,

$$\mathbb{E}[v_a \eta_{i_1} \cdots \eta_{i_{2l+1}}] = \sum_{b_1, \ldots, b_{2l+1}} \left( \prod_{t=1}^{2l+1} B_{b_t i_t} \right) \mathbb{E}[v_a v_{b_1} \cdots v_{b_{2l+1}}] = c_{l+1}(d) \sum_{\pi \in \mathcal{P}_{2l+2}} \sum_{b_1, \ldots, b_{2l+1}} \left( \prod_{t=1}^{2l+1} B_{b_t i_t} \right) \prod_{(p,q) \in \pi} P_{j_p j_q},$$

where $\mathcal{P}_{2l+1}$ is the set of all permutations for $2l + 1$ elements and we used Lemma 36 from [Damian et al. (2022)](#). For one fixed $\pi$, index 0 is paired with exactly one $s \in \{1, \ldots, 2l + 1\}$. The factor involving $b_s$ is $B_{b_s i_s} P_{a b_s}$, whose sum over $b_s$ equals $(PB)_{a i_s}$. Each remaining pair $(p, q)$ contributes

$$\sum_{b_p, b_q} (B)_{b_p i_p} P_{b_p b_q} (B)_{b_q i_q} = (B^\top P B)_{i_p i_q} = \Sigma_{i_p i_q}.$$

Since the sum over $\pi$ is equivalent to choosing the partner $s$ of index 0 (there are $2l + 1$ choices) and a pairing $\pi'$ of the remaining $2l$ indices, we can conclude

$$\mathbb{E}_v[v \otimes \eta^{\otimes 2l+1}] = C_l c_{l+1}(d) \mathrm{Sym}(PB \otimes \Sigma^{\otimes l}),$$

where $C_l$ is a constant that only depends on $l$. Putting back to our equation, we have

$$\begin{aligned}
\mathbb{E}_v \left[ v C_k(a^{\otimes k}) \right] &= \sum_{l=0}^{\lfloor (k-1)/2 \rfloor} \binom{k}{2l+1} t^{k-2l-1} (1-t^2)^{(2l+1)/2} \mathbb{E}_v[v \otimes \eta^{\otimes 2l+1}] (V_l) \\
&= \sum_{l=0}^{\lfloor (k-1)/2 \rfloor} \binom{k}{2l+1} C_l c_{l+1}(d) t^{k-2l-1} (1-t^2)^{(2l+1)/2} \mathrm{Sym}(PB \otimes \Sigma^{\otimes l}) (V_l) \\
&= \sum_{l=0}^{\lfloor (k-1)/2 \rfloor} \binom{k}{2l+1} C_l c_{l+1}(d) t^{k-2l-1} (1-t^2)^{(2l+1)/2} PB(C_k(\mathrm{Sym}(b^{\otimes k-2l-1} \otimes \Sigma^{\otimes l}))) \\
&= \sum_{l=0}^{\lfloor (k-1)/2 \rfloor} \binom{k}{2l+1} C_l c_{l+1}(d) t^{k-2l-1} (1-t^2)^{(2l+1)/2} PB(C_k(\bar{S}_{k,l})),
\end{aligned}$$

where we defined $\bar{S}_{k,l} := \mathrm{Sym}(b^{\otimes k-2l-1} \otimes \Sigma^{\otimes l})$. Combining with the definition of $T_{2.1}$, we obtain

$$T_{2,2} = PU \sum_{l=0}^{\lfloor (k-1)/2 \rfloor} \binom{k}{2l+1} C_l c_{l+1}(d) C_k(\bar{S}_{k,l}) \int_{-1}^1 t^{k-2l-1} (1-t^2)^{l+1} h'(t) f_d(t) \mathrm{d}t.$$

Next, we consider $T_3$. Actually, since $\langle \hat{C}_{k+1}(w^{\otimes k}), \tilde{x} \rangle = \langle C_{k+1}(a^{\otimes k}), b \rangle = \tilde{C}_{k+1}(a^{\otimes k})$, where $\tilde{C}_{k+1} = C_{k+1}(b)$, we can directly reuse the result of $T_2$, leading to

$$\begin{aligned}
T_3 =& \tilde{x} \sum_{l=0}^{\lfloor k/2 \rfloor} \binom{k}{2l} c_l(d) \tilde{C}_{k+1}(S_{k,l}) \int_{-1}^1 t^{k-2l+1} (1-t^2)^l h''(t) f_d(t) \mathrm{d}t \\
&+ PB \sum_{l=0}^{\lfloor (k-1)/2 \rfloor} \binom{k}{2l+1} C_l c_{l+1}(d) \tilde{C}_{k+1}(\bar{S}_{k,l}) \int_{-1}^1 t^{k-2l-1} (1-t^2)^{l+1} h''(t) f_d(t) \mathrm{d}t \\
=& \tilde{x} \sum_{l=0}^{\lfloor k/2 \rfloor} \binom{k}{2l} c_l(d) \langle C_{k+1}(S_{k,l}), B^\top \tilde{x} \rangle \int_{-1}^1 t^{k-2l+1} (1-t^2)^l h''(t) f_d(t) \mathrm{d}t \\
&+ PB \sum_{l=0}^{\lfloor (k-1)/2 \rfloor} \binom{k}{2l+1} C_l c_{l+1}(d) \langle C_{k+1}(\bar{S}_{k,l}), B^\top \tilde{x} \rangle \int_{-1}^1 t^{k-2l-1} (1-t^2)^{l+1} h''(t) f_d(t) \mathrm{d}t.
\end{aligned}$$

Finally, $T_4 = \mathbb{E}_{t,v}\left[ wh''(t)tC_k(a^{\otimes k})t \right]$, this is again $T_2$ with an additional factor of $t$. As result, by analogy,

$$
\begin{aligned}
T_4 =& \tilde{x} \sum_{l=0}^{\lfloor k/2 \rfloor} \binom{k}{2l} c_l(d) C_k(S_{k,l}) \int_{-1}^{1} t^{k-2l+2}(1-t^2)^l h''(t) f_d(t)\mathrm{d}t \\
&+ PB \sum_{l=0}^{\lfloor (k-1)/2 \rfloor} \binom{k}{2l+1} C_l c_{l+1}(d) C_k(\bar{S}_{k,l}) \int_{-1}^{1} t^{k-2l}(1-t^2)^{l+1} h''(t) f_d(t)\mathrm{d}t.
\end{aligned}
$$

$\square$

Let us now estimate each term.

**Lemma D.5.** *When* $\sup_{|z|\leq 1}|h'(z)| \leq M_1$, $\sup_{|z|\leq 1}|h''(z)| \leq M_2$, $B^\top B = I_r$ *and* $\|\tilde{x}\| = 1$, *then for* $k \geq 0$, $T_1^{(k)} = \tilde{O}(d^{-k/2}r^{\lfloor k/2 \rfloor/2})$, $T_2^{(k)} = \tilde{O}(d^{-(k+1)/2}r^{\lfloor k/2 \rfloor/2})$, $T_3^{(k)} = \tilde{O}(d^{-(k+1)/2}r^{\lfloor k/2 \rfloor/2})$ *and* $T_4^{(k)} = \tilde{O}(d^{-(k+2)/2}r^{\lfloor k/2 \rfloor/2})$.

*Proof.* As a reminder, from the proof of Lemma B.16, we know that for a smooth bounded function $i$ defined on $[-1, 1]$, we have

$$
|\mathcal{I}_{p_1,p_2}[i]| \sim d^{-p_1/2}.
$$

Therefore,

$$
\begin{aligned}
\|T_1\| &\leq \left\| B \sum_{l=0}^{\lfloor k/2 \rfloor} \binom{k}{2l} c_l(d) C_{k+1}(S_{k,l}) \mathcal{I}_{k-2l,l}[h'] \right\| \\
&\leq \sum_{l=0}^{\lfloor k/2 \rfloor} \binom{k}{2l} c_l(d) \|C_{k+1}(S_{k,l})\| |\mathcal{I}_{k-2l,l}[h']| \\
&\leq \sum_{l=0}^{\lfloor k/2 \rfloor} \binom{k}{2l} c_l(d) \|C_{k+1}\|_F \|\mathrm{Sym}((B^\top \tilde{x})^{\otimes k-2l} \otimes \Sigma^{\otimes l})\| |\mathcal{I}_{k-2l,l}[h']| \\
&\leq \sum_{l=0}^{\lfloor k/2 \rfloor} \binom{k}{2l} c_l(d) \|C_{k+1}\|_F \|\Sigma\|_F^l |\mathcal{I}_{k-2l,l}[h']| \\
&\lesssim \sum_{l=0}^{\lfloor k/2 \rfloor} d^{-l} r^{l/2} d^{-(k-2l)/2} \\
&\lesssim d^{-k/2} r^{\lfloor k/2 \rfloor/2},
\end{aligned}
$$

where we used $\|C_{k+1}\|_F = O(1)$ from Lemma B.12, $\|\tilde{x}\| = 1$ and $\|\Sigma\|_F \leq \sqrt{r}$. Likewise, we can follow the same procedure to conclude, $T_2 = \tilde{O}(d^{-(k+1)/2}r^{\lfloor k/2 \rfloor/2})$, $T_3 = \tilde{O}(d^{-(k+1)/2}r^{\lfloor k/2 \rfloor/2})$ and $T_4 = \tilde{O}(d^{-(k+2)/2}r^{\lfloor k/2 \rfloor/2})$. $\square$

In the next lemma, we provide tighter bounds for a few lower order terms.

**Lemma D.6.** *If* $h'$ *and* $h''$ *are continuous and* $h''(t) > 0$ *in the interval* $[-1, 1]$, *then with high probability,* $T_1^{(1)} = \frac{1}{d}\mathbb{E}_z[h''(z)]H\tilde{x}$, $T_3^{(1)} = \frac{1}{d-1}\mathbb{E}_t[(1-t^2)h''(t)]H\tilde{x} + \tilde{O}(rd^{-2})$, $T_1^{(3)} = \tilde{O}(rd^{-2})$ *and* $T_2^{(2)} = \tilde{O}(rd^{-2})$, *where the probability density function of the random variable* $t$ *is* $f_d(t) = \frac{\Gamma(\frac{d}{2})}{\sqrt{\pi}\Gamma(\frac{d-1}{2})}(1-t^2)^{\frac{d-3}{2}}$, *and that of* $z$ *is* $f_{d+2}(t)$.

*Proof.* The assumption on $h'$ and $h''$ implies that $h'(t) \leq M_1$ and $m_2 \leq h'(t) \leq M_2$ for all $t \in [-1, 1]$.

We first prove an important equality that we will use throughout the proof. Since

$$
\frac{\mathrm{d}}{\mathrm{d}t}(1-t^2)^{\frac{d-1}{2}} = -(d-1)t(1-t^2)^{\frac{d-3}{2}},
$$

the integration by part leads to

$$\mathbb{E}[th'(t)] = \int_{-1}^{1} th'(t)f_d(t)\mathrm{d}t$$

$$= \left[-\frac{1}{d-1}h'(t)(1-t^2)^{\frac{d-1}{2}}\right]_{-1}^{1} + \int_{-1}^{1}\frac{1}{d-1}\frac{\Gamma(\frac{d}{2})}{\sqrt{\pi}\Gamma(\frac{d-1}{2})}h''(t)(1-t^2)^{\frac{d-1}{2}}\mathrm{d}t$$

$$= \frac{1}{d-1}\frac{\Gamma(\frac{d}{2})}{\sqrt{\pi}\Gamma(\frac{d-1}{2})}\frac{\sqrt{\pi}\Gamma(\frac{d+1}{2})}{\Gamma(\frac{d+2}{2})}\int_{-1}^{1}\frac{\Gamma(\frac{d+2}{2})}{\sqrt{\pi}\Gamma(\frac{d+1}{2})}h''(z)(1-z^2)^{\frac{d-1}{2}}\mathrm{d}z$$

$$= \frac{1}{d-1}\frac{\Gamma(\frac{d}{2})}{\sqrt{\pi}\Gamma(\frac{d-1}{2})}\frac{\sqrt{\pi}\frac{d-1}{2}\Gamma(\frac{d-1}{2})}{\frac{d}{2}\Gamma(\frac{d}{2})}\mathbb{E}[h''(z)]$$

$$= \frac{1}{d}\mathbb{E}[h''(z)]. \tag{21}$$

From this equation, we can derive the following relations.

$$\mathbb{E}[t^2] = \frac{1}{d}, \tag{22}$$

$$|\mathbb{E}[t^3 h'(t)]| \leq \frac{2M_1 + M_2}{d}, \tag{23}$$

$$|\mathbb{E}[t(1-t^2)h'(t)]| \leq \frac{2M_1 + M_2}{d}, \tag{24}$$

where we used equation (21) once for each relation.

Moreover,

$$\mathbb{E}_t[(1-t^2)] = \int_{-1}^{1} c_d(1-t^2)^{\frac{d-1}{2}}\mathrm{d}t = \frac{c_d}{c_{d+2}} = \Theta(1), \tag{25}$$

and following Corollary 46 (Damian et al., 2022), with probability $1 - 2\mathrm{e}^{-\iota}$

$$\|B^\top \tilde{x}\| \lesssim \sqrt{\frac{r\iota}{d}}. \tag{26}$$

Based on the parity of $k$, $T_3^{(1)}$ introduces an additional $\sqrt{\frac{r\iota}{d}}$ factors, leading to the bound of the statement.

Let us now move on to proving our main statements. The evaluation of $T_1^{(1)}$ is straightforward as

$$T_1^{(1)} = \mathbb{E}[\hat{C}_2(w^{\otimes 1})h'(\langle w, \tilde{x}\rangle)] = H\mathbb{E}[wh'(\langle w, \tilde{x}\rangle)] = H\tilde{x}\mathbb{E}[th'(t)] = \frac{1}{d}\mathbb{E}[h''(t)]H\tilde{x},$$

where we used the same change of variable and reasoning as Lemma D.4 for the third inequality, and equation 21 for the last equality.

Again with the same change of variable, for $T_3^{(1)}$, we obtain

$$T_3^{(1)} = \mathbb{E}_w[wh''(\langle w, \tilde{x}\rangle)\langle \hat{C}_2(w), \tilde{x}\rangle]$$

$$= \mathbb{E}_w[wh''(\langle w, \tilde{x}\rangle)\langle Hw, \tilde{x}\rangle]$$

$$= \mathbb{E}_w[ww^\top h''(\langle w, \tilde{x}\rangle)]H\tilde{x}$$

$$= \mathbb{E}_{v,t}[(t\tilde{x} + \sqrt{1-t^2}v)(t\tilde{x} + \sqrt{1-t^2}v)^\top h''(t)]H\tilde{x}$$

$$= \mathbb{E}_{v,t}[t^2\tilde{x}\tilde{x}^\top h''(t) + (1-t^2)vv^\top h''(t)]H\tilde{x}$$

$$= \mathbb{E}_t[t^2 h''(t)](\tilde{x}^\top H\tilde{x})\tilde{x} + \mathbb{E}_t[(1-t^2)h''(t)]\mathbb{E}_v[vv^\top]H\tilde{x}$$

$$= \mathbb{E}_t[t^2 h''(t)](\tilde{x}^\top H\tilde{x})\tilde{x} + \mathbb{E}_t[(1-t^2)h''(t)]\frac{P}{d-1}H\tilde{x}$$

$$= \mathbb{E}_t[t^2 h''(t)](\tilde{x}^\top H\tilde{x})\tilde{x} + \frac{1}{d-1}\mathbb{E}_t[(1-t^2)h''(t)]H\tilde{x} - \frac{1}{d-1}\mathbb{E}_t[(1-t^2)h''(t)](\tilde{x}^\top H\tilde{x})\tilde{x},$$

where we used that $(d-1)\mathbb{E}[vv^\top] = P := I - \tilde{x}\tilde{x}^\top$ for the seventh equality. Since $|\tilde{x}^\top H\tilde{x}| = \tilde{O}(rd^{-1})$ by Lemma B.11 and equation (26), the third term is $\tilde{O}(rd^{-2})$. The first term is also $O(rd^{-2})$ as $|\mathbb{E}_t[t^2 h''(t)]| \leq M_2\mathbb{E}[t^2] = M_2/d$ where we used equation (22). The second term is the dominant term as $H\tilde{x}$ is $\tilde{O}(d^{-1/2})$ and $\mathbb{E}_t[(1-t^2)h''(t)] = \Theta(1)$ since $m_2\mathbb{E}[(1-t^2)] \leq \mathbb{E}_t[(1-t^2)h''(t)] \leq M_2$ and equation 25 holds. Consequently,

$$T_3^{(1)} = \frac{1}{d-1}\mathbb{E}_t[(1-t^2)h''(t)]H\tilde{x} + \tilde{O}(rd^{-2}).$$

The bound for $T_1^{(3)}$ follows immediately by the result of Lemma D.5, as $T_1^{(3)}$ includes a coefficient $b$ that we omitted in the computation of the upper bound in the proof of Lemma D.4, which adds an additional $\sqrt{r/d}$ coefficient to the bounds, leading to $\tilde{O}(rd^{-2})$.

Finally, as for $T_2^{(2)}$, we carefully develop its expression as follows:

$$
\begin{aligned}
T_2^{(2)} &= \mathbb{E}_w[w\hat{C}_2(w^{\otimes 2})h'(\langle w, \tilde{x}\rangle)] \\
&= \mathbb{E}_w[ww^\top Hwh'(\langle w, \tilde{x}\rangle)] \\
&= \mathbb{E}_{t,v}[(t\tilde{x} + \sqrt{1-t^2}v)(t\tilde{x} + \sqrt{1-t^2}v)^\top H(t\tilde{x} + \sqrt{1-t^2}v)h'(t)] \\
&= \mathbb{E}_{t,v}\left[\left\{t^3(\tilde{x}^\top H\tilde{x})\tilde{x} + \frac{\text{Tr}(PH)}{d-1}t(1-t^2)\tilde{x} + 2t(1-t^2)\frac{PH}{d-1}\tilde{x}\right\}h'(t)\right] \\
&= \mathbb{E}_{t,v}[t^3 h'(t)](\tilde{x}^\top H\tilde{x})\tilde{x} + \frac{\text{Tr}(PH)}{d-1}\mathbb{E}[t(1-t^2)h'(t)]\tilde{x} + 2\mathbb{E}[t(1-t^2)h'(t)]\frac{PH}{d-1}\tilde{x}.
\end{aligned}
$$

Here, the first and third terms are $\tilde{O}(rd^{-2})$ by equations 23, 24 and 26. Concerning the second term, we will just need to prove that $\text{Tr}(PH)$ is independent of $d$ as the remainder is $O(d^{-2})$ following equation 24. However,

$$
\begin{aligned}
|\text{Tr}(PH)| &\leq |\text{Tr}(H)| + |\text{Tr}(\tilde{x}\tilde{x}^\top H)| \\
&\leq |\text{Tr}(H)| + |\text{Tr}(\tilde{x}^\top H\tilde{x})| \\
&\lesssim |\text{Tr}(H)| \\
&= |\text{Tr}(BC_2 B^\top)| \\
&= |\text{Tr}(C_2 B^\top B)| \\
&= |\text{Tr}(C_2)| \\
&\leq \sqrt{r}\|C_2\|_F \lesssim \sqrt{r},
\end{aligned}
$$

where we used Lemma B.11 in the first equality and Lemma B.12 for the last inequality.

Therefore, we have proved the desired statements. □

**Corollary D.7.** *Notably, we obtain*

$$T_1^{(1)} + T_3^{(1)} + T_1^{(3)} + T_2^{(2)} = c_d H\tilde{x} + \tilde{O}(rd^{-2}),$$

*where $c_d = \Theta(d^{-1})$.*

*Proof.* This follows as $E_z[h''(z)] = \Theta(1)$ and $\mathbb{E}_t[(1-t^2)h''(t)] = \Theta(1)$, which implies

$$T_1^{(1)} + T_3^{(1)} = \left(\frac{1}{d}E_z[h''(z)] + \frac{1}{d-1}\mathbb{E}_t[(1-t^2)h''(t)]\right)H\tilde{x} + O(rd^{-2}).$$

Therefore, by defining $c_d := \frac{1}{d}E_z[h''(z)] + \frac{1}{d-1}\mathbb{E}_t[(1-t^2)h''(t)] = \Theta(\frac{1}{d}) > 0$, we obtain the desired result. □

This leads to the following result.

**Theorem D.8.** *3.3, 3.4, 3.6, 3.7 and 3.8, when $\tilde{x}^{(0)} \sim U(S^{d-1})$, with high probability,*

$$\hat{G} = c_d H \tilde{x} + \tilde{O}\left(d^{\frac{1}{2}} N^{-\frac{1}{2}} + r d^{-2}\right),$$

*where $c_d = \Theta\left(d^{-1}\right)$ is a constant coefficient.*

*Proof.* From Lemma D.3 and Corollary D.7,

$$
\begin{aligned}
\hat{G} =& \sum_{k=1}^{p-1} \frac{c_{k+1}}{k!} \mathbb{E}_w \left[\hat{C}_{k+1}(w^{\otimes k}) h'(\langle w, \tilde{x} \rangle)\right] + \sum_{k=2}^{p} \frac{c_{k+2}}{k!} \mathbb{E}_w \left[w \hat{C}_k(w^{\otimes k}) h'(\langle w, \tilde{x} \rangle)\right] \\
&+ \sum_{k=1}^{p-1} \frac{c_{k+1}}{k!} \mathbb{E}_w \left[w h''(\langle w, \tilde{x} \rangle) \left\langle \hat{C}_{k+1}(w^{\otimes k}), \tilde{x} \right\rangle\right] + \sum_{k=2}^{p} \frac{c_{k+2}}{k!} \mathbb{E}_w \left[w h''(\langle w, \tilde{x} \rangle) \hat{C}_k(w^{\otimes k}) \langle w, \tilde{x} \rangle\right] \\
&+ \tilde{O}\left(\sqrt{\frac{d}{N}}\right) \\
=& \sum_{k=1}^{p-1} \frac{c_{k+1}}{k!} T_1^{(k)} + \sum_{k=2}^{p} \frac{c_{k+2}}{k!} T_2^{(k)} + \sum_{k=1}^{p-1} \frac{c_{k+1}}{k!} T_3^{(k)} + \sum_{k=2}^{p} \frac{c_{k+2}}{k!} T_4^{(k)} + \tilde{O}\left(\sqrt{\frac{d}{N}}\right) \\
=& \frac{c_2}{1!} T_1^{(1)} + \frac{c_4}{3!} T_1^{(3)} + \frac{c_4}{2!} T_2^{(2)} + \frac{c_2}{1!} T_3^{(1)} \\
&+ \sum_{k=4}^{p-1} \frac{c_{k+1}}{k!} T_1^{(k)} + \sum_{k=3}^{p} \frac{c_{k+2}}{k!} T_2^{(k)} + \sum_{k=2}^{p-1} \frac{c_{k+1}}{k!} T_3^{(k)} + \sum_{k=2}^{p} \frac{c_{k+2}}{k!} T_4^{(k)} + \tilde{O}\left(\sqrt{\frac{d}{N}}\right) \\
=& \frac{c_2}{1!} T_1^{(1)} + \frac{c_2}{1!} T_3^{(1)} + \frac{c_4}{3!} T_1^{(3)} + \frac{c_4}{2!} T_2^{(2)} + \tilde{O}(r d^{-2} + d^{1/2} N^{-1/2}) \\
=& c_d H \tilde{x} + \tilde{O}(r d^{-2} + d^{1/2} N^{-1/2}).
\end{aligned}
$$

$\square$

The remainder of this section (the proof of the divergence between the population and empirical gradients) follows exactly that of the single index model. In other words, we obtain the following theorem.

**Theorem D.9.** *Under Assumptions 3.4 and 3.1, 3.3, 3.4, 3.6 and 3.8, then for $k \geq 0$, when $\mathcal{D}_0^S = \{(\tilde{x}_m^{(0)}, \tilde{y}_m^{(0)})\}_{m \in [M_1]}$, where $\tilde{x}_m^{(0)} \sim U(S^{d-1})$, the first step of distillation gives $\tilde{x}_m^{(1)}$ such that*

$$\tilde{x}_m^{(1)} = -\eta_1^D \tilde{y}_m^{(0)} \left(c_d H \tilde{x}_m^{(0)} + \tilde{O}\left(r d^{-2} + d^{\frac{1}{2}} N^{-\frac{1}{2}} + d^{\frac{1}{2}} J^{*-\frac{1}{2}}\right)\right),$$

*where $c_d = \tilde{\Theta}\left(d^{-1}\right)$, $J^* = LJ/2$.*

We will also need the following corollary later.

**Corollary D.10.** *$\tilde{x}_m^{(1)}$ can be decomposed into $-\eta_1^D \tilde{y}_m^{(0)} c_d H \tilde{x}_m$ and an error term $\eta_1^D \tilde{y} \epsilon$ depending on $\tilde{x}$. Consider $M_1$ samples $\tilde{x}_m^{(0)} \sim U(S^{d-1})$, then with probability at least $1 - \delta$, $\max_m \|\epsilon(\tilde{x}_m^{(0)})\| = \tilde{O}(r d^{-2} + d^{\frac{1}{2}} N^{-\frac{1}{2}} + d^{\frac{1}{2}} J^{*-\frac{1}{2}})$. This is a high probability event in our definition.*

### D.4. $t = 2$ Teacher Training

Let us now consider the teacher training at $t = 2$. The updated weights, based on $\mathcal{D}_1^S$ can be written as follows:

$$w_i^{(1)} = -\eta_1^R a_i \frac{1}{M_1} \sum_m \tilde{y}_m^{(0)} \tilde{x}_m^{(1)} \sigma'(\langle w_i^{(0)}, \tilde{x}_m^{(1)} \rangle),$$

where $\tilde{x}_m^{(1)} = \eta_1^D \tilde{y}^{(0)} c_d H \tilde{x}^{(0)} + \eta_1^D \tilde{y}^{(0)} \epsilon_m$. We drop the indices such as $(0)$ and $(1)$ for simplicity sake and suppose that there are $L^*$ neurons that are not $0$. The goal of this subsection is to provide a similar proof flow as Damian et al. (2022) to

analyze the behavior of DD and show that the resulting distilled data provide high generalization performance at retraining. Moreover, by setting $(\tilde{y}^{(0)})^2 \sim \chi(d)$, we can absorb the randomness of $\tilde{y}^{(0)}$ into $\tilde{x}_m^{(1)}$. To summarize, the gradient of the teacher training using $M_1$ distilled data points can be defined with the property as follows:

**Lemma D.11.** *Under the assumptions of Theorem D.9, the gradient of step $t = 2$ of the teacher training is*

$$g_M(w) := \frac{\eta_1^D}{M} \sum_m \tilde{z}_m \sigma'(\langle w, \tilde{z}_m \rangle),$$

*where $\tilde{z}_m = c_d H \tilde{x}_m + \tilde{\epsilon}_m$, where $\tilde{x}_m \sim N(0, I_d)$, and $\tilde{\epsilon}_m = \tilde{O}(rd^{-\frac{3}{2}} + dN^{-\frac{1}{2}} + dJ^{*-\frac{1}{2}})$, with high probability.*

*Proof.* Note that

$$\frac{1}{M_1} \sum_m \tilde{y}_m^{(0)} \tilde{x}_m^{(1)} \sigma'(\langle w_i^{(0)}, \tilde{x}_m^{(1)} \rangle) = \frac{1}{M_1} \sum_m \tilde{y}_m^{(0)} \left( \eta_1^D \tilde{y}^{(0)} c_d H \tilde{x}^{(0)} + \eta_1^D \tilde{y}^{(0)} \epsilon_m \right) \sigma'(\langle w^{(0)}, \eta_1^D \tilde{y}^{(0)} c_d H \tilde{x}^{(0)} + \eta_1^D \tilde{y}^{(0)} \epsilon_m \rangle)$$

$$= \frac{\eta_1^D}{M_1} \sum_m \left( c_d H (\tilde{y}^{(0)})^2 \tilde{x}^{(0)} + (\tilde{y}^{(0)})^2 \epsilon_m \right) \sigma'(\langle w^{(0)}, c_d H (\tilde{y}^{(0)})^2 \tilde{x}^{(0)} + (\tilde{y}^{(0)})^2 \epsilon_m \rangle),$$

where we used that $\sigma'(t^2 z) = \sigma'(z)$ for all $t > 0$. Now since $(\tilde{y}^{(0)})^2 \tilde{x}^{(0)} \sim N(0, I_d)$, and with high probability, $(\tilde{y}^{(0)})^2 = \tilde{\Theta}(\sqrt{d})$ from Lemma A.2, we obtain the desired formulation by combining with D.10. $\square$

We now consider population gradient of $g_M$ with no perturbation $\tilde{\epsilon}_m$.

**Lemma D.12.** *Under $\tilde{x}_m \sim N(0, I_d)$,*

$$g(w) := \mathbb{E}_{\tilde{x}_1, \dots, \tilde{x}_M} \left[ \frac{\eta_1^D}{M_1} \sum_m c_d H \tilde{x}_m \sigma'(\langle w, c_d H \tilde{x}_m \rangle) \right] = \alpha_d \frac{H^2 w}{\|Hw\|},$$

*where $\alpha_d = \frac{\eta_1^D c_d}{\sqrt{2\pi}}$.*

*Proof.* Since $\tilde{x}_1, \dots, \tilde{x}_M$ are all i.i.d. and $H^\top = H$, we can consider one sample. Consequently,

$$g(w) = \eta_1^D c_d H \mathbb{E}_{\tilde{x}_1} [\tilde{x}_1 \sigma'(\langle Hw, \tilde{x}_1 \rangle)].$$

By rotational symmetry, the expectation has to align with $Hw$, which means the expectation can be simplified as

$$g(w) = \eta_1^D c_d H \mathbb{E}_Z [Z \sigma'(Z)] \frac{Hw}{\|Hw\|},$$

with $Z \sim N(0, 1)$. Since the expectation is $\frac{1}{\sqrt{2\pi}}$, we obtain the desired result. $\square$

With this $g(w)$, the following two inequalities hold. These constitute the ingredients to prove a lemma similar to Lemma 21 (Damian et al., 2022).

**Lemma D.13.** *Under the assumptions of Theorem D.9, for all $\|T\|_F = 1$, $k \geq 1$ and $i \in [k]$,*

$$\mathbb{E}_w[\langle T, g(w)^{\otimes k} \rangle^2] \gtrsim \alpha_d^{2k} r^{-k} \kappa^{-2k},$$

$$\mathbb{E}_w[\|T(g(w)^{\otimes k-i})\|_F^2] \lesssim \alpha_d^{-2i} r^i \kappa^{2k} \lambda_{\min}^{2i} \mathbb{E}_w[\langle T, g(w)^{\otimes k} \rangle^2],$$

*where $\kappa = \lambda_{\max} / \lambda_{\min}$.*

*Proof.* Before starting computing the expectation, we make the following observation. Let us consider the decomposition $H = D \Lambda D^\top$, where $\Lambda \in \mathbb{R}^{r \times r}$, $D \in \mathbb{R}^{d \times r}$ and $D^\top D = I_r$. Such a decomposition exists by Assumption 3.3. In other words,

$$g(w) = \alpha_d D \frac{\Lambda^2 D^\top w}{\|D \Lambda D^\top w\|} = \alpha_d D \frac{\Lambda^2 D^\top w}{\|\Lambda D^\top w\|}.$$

Now, by rotational symmetry and the fact that $g(w)$ is invariant to scaling of $w$, all expectations over $w$ that only include $g(w)$ are equivalent to the following $g(u)$ with $u \sim S^{r-1}$,[8]

$$g(u) = \alpha_d D \frac{\Lambda^2 u}{\|\Lambda u\|}.$$

Taking into account this, we obtain

$$\mathbb{E}_w[\langle T, g(w)^{\otimes k}\rangle^2] = \mathbb{E}_u[\langle T, g(u)^{\otimes k}\rangle^2] = \alpha_d^{2k} \mathbb{E}_u\left[\langle T, (D\Lambda^2 u)^{\otimes k}\rangle^2/\|\Lambda u\|^{2k}\right].$$

Since $\|\Lambda u\| \leq \lambda_{\max}$,

$$\mathbb{E}_w[\langle T, g(w)^{\otimes k}\rangle^2] \geq \alpha_d^{2k} \lambda_{\max}^{-2k} \mathbb{E}_u\left[\langle \hat{T}, u^{\otimes k}\rangle^2\right],$$

where $\hat{T}$ is defined by $\hat{T}(u_1, \ldots, u_k) = T(D\Lambda^2 u_1, \ldots, D\Lambda^2 u_k)$. As a result, by the same procedure as the proof of Lemma 21 of Damian et al. (2022),

$$\mathbb{E}_w[\langle T, g(w)^{\otimes k}\rangle^2] \gtrsim \alpha_d^{2k} \lambda_{\max}^{-2k} r^{-k} \|\hat{T}\|_F \geq \alpha_d^{2k} \lambda_{\max}^{-2k} r^{-k} \lambda_{\min}^{2k} \|T\|_F.$$

We now proof the second inequality of the statement.

$$\begin{aligned}
\mathbb{E}_w[\|T(g(w)^{\otimes k-i})\|_F^2] &= \left(\frac{\alpha_d}{\lambda_{\min}}\right)^{2(k-i)} \mathbb{E}_u[\|T((D\Lambda^2 u)^{\otimes k-i})\|_F^2] \\
&= \left(\frac{\alpha_d}{\lambda_{\min}}\right)^{2(k-i)} \mathbb{E}_u[\|\hat{T}(u^{\otimes k-i})\|_F^2] \\
&\lesssim \left(\frac{\alpha_d}{\lambda_{\min}}\right)^{2(k-i)} r^i \mathbb{E}_u[\langle \hat{T}, u^{\otimes k}\rangle^2] \\
&= \left(\frac{\alpha_d}{\lambda_{\min}}\right)^{2(k-i)} r^i \mathbb{E}_u\left[\langle T, (\alpha_d D\Lambda^2 u/\|\Lambda u\|)^{\otimes k}\rangle^2 \cdot \left(\frac{\|\Lambda u\|}{\alpha_d}\right)^{2k}\right] \\
&= \alpha_d^{-2i} r^i \left(\frac{\lambda_{\max}}{\lambda_{\min}}\right)^{2k} \lambda_{\min}^{2i} \mathbb{E}_u\left[\langle T, g(u)^{\otimes k}\rangle^2\right].
\end{aligned}$$

$\square$

**Definition D.14.** For $w \in S^{d-1}$, we define
$$r(w) := g_M(w) - g(w).$$

This error can be bounded as follows.

**Lemma D.15.** *With probability at least* $1 - \delta$*, we have for* $j \leq 4p$*, if* $d \gtrsim \frac{144\alpha^2 r^2}{\lambda_{\min}^2}$*,* $N \gtrsim \frac{144\alpha^2 d^4}{\lambda_{\min}^2}$ *and* $J^* \gtrsim \frac{144\alpha^2 d^4}{\lambda_{\min}^2}$*,*

$$\mathbb{E}_w[\|\Pi^* r(w)\|^j]^{1/j} \lesssim \lambda_{\max} \eta_1^D c_d \sqrt{\frac{r + \log(1/\delta)}{M_1}} + \frac{\lambda_{\min} \eta_1^D c_d}{4\alpha} + \lambda_{\max} \eta_1^D c_d \left(\sqrt{r} + \sqrt{\frac{\log 1/\delta}{M_1}}\right) \left(\frac{4\alpha\epsilon}{\lambda_{\min} c_d}\right)^{1/j} + \eta_1^D \epsilon,$$

*where* $\epsilon = \tilde{O}(rd^{-\frac{3}{2}} + dN^{-\frac{1}{2}} + dJ^{*-\frac{1}{2}})$*.*

*Proof.* Since

$$r(w) = \frac{\eta_1^D}{M_1} \sum_m \tilde{z}_m \sigma'(\langle w, \tilde{z}_m\rangle) - \eta_1^D \mathbb{E}_{\tilde{x}_1}\left[c_d H\tilde{x}_1 \sigma'(\langle Hw, c_d\tilde{x}_1\rangle)\right],$$

---

[8]We can first substitute $w$ with $g \sim N(0, I_d)$ by the scale invariance of $g$, then use the rotation invariance of the Gaussian distribution to change $D^\top g$ to $\tilde{u} \sim N(0, I_r)$, and finally by the scale invariance of $g$ again, replace $\tilde{u}$ by $u \sim S^{r-1}$.

where $\tilde{z}_m = c_d H \tilde{x}_m + \tilde{\epsilon}_m$, $r$ can be divided into the following three terms.

$$R_1 = \frac{\eta_1^D}{M_1} \sum_m c_d H \tilde{x}_m \sigma'(\langle w, c_d H \tilde{x}_m \rangle) - \eta_1^D \mathbb{E}_{\tilde{x}_1} \left[ c_d H \tilde{x}_1 \sigma'(\langle H w, c_d \tilde{x}_1 \rangle) \right]$$

$$R_2 = \frac{\eta_1^D}{M_1} \sum_m c_d H \tilde{x}_m \sigma'(\langle w, c_d H \tilde{x}_m + \tilde{\epsilon}_m \rangle) - \frac{\eta_1^D}{M_1} \sum_m c_d H \tilde{x}_m \sigma'(\langle w, c_d H \tilde{x}_m \rangle)$$

$$R_3 = \frac{\eta_1^D}{M_1} \sum_m \tilde{\epsilon}_m \sigma'(\langle w, c_d H \tilde{x}_m + \tilde{\epsilon}_m \rangle).$$

Consequently,

$$\mathbb{E}_w[\|\Pi^* r(w)\|^j]^{1/j} \leq \mathbb{E}_w[\|\Pi^* R_1(w)\|^j]^{1/j} + \mathbb{E}_w[\|\Pi^* R_2(w)\|^j]^{1/j} + \mathbb{E}_w[\|\Pi^* R_3(w)\|^j]^{1/j}.$$

We start by bounding $\mathbb{E}_w[\|\Pi^* R_1(w)\|^j]^{1/j}$. Since $\mathbb{E}_w[\|\Pi^* R_1(w)\|^j]^{1/j} \leq \sup_{w \in S^{d-1}} \|\Pi^* R_1(w)\|$, we will bound the right hand side.

$$\sup_{w \in S^{d-1}} \|\Pi^* R_1(w)\| = \sup_{w \in S^{d-1}} \sup_{z \in S^{d-1}} \langle z, \Pi^* R_1(w) \rangle.$$

We can thus proceed similarly to Theorem B.19, by observing that

$$\langle z, H \tilde{x}_m \rangle \sigma'(\langle w, H \tilde{x}_m \rangle) = \langle D^\top z, \Lambda D^\top \tilde{x}_m \rangle \sigma'(\langle D^\top w, \Lambda D^\top \tilde{x}_m \rangle),$$

and using rotational symmetry, all vectors are projected to a $r$-dimensional subspace. In other words,

$$\sup_{w \in S^{d-1}} \sup_{z \in S^{d-1}} \langle z, \Pi^* R_1(w) \rangle = \sup_{w \in S^{r-1}} \sup_{z \in S^{r-1}} \langle z, \Lambda b_m \rangle \sigma'(\langle w, \Lambda b_m \rangle),$$

with samples $b_m = D^\top \tilde{x}_m^{(0)} \in \mathbb{R}^r$. This is a $\lambda_{\max}$-sub-gaussian random variable in $\mathbb{R}^r$. This thus yields with probability at least $1 - \delta$ over data $\{x_m\}$

$$\sup_{w \in S^{d-1}} \|\Pi^* R_1(w)\| \lesssim \lambda_{\max} \eta_1^R c_d \sqrt{\frac{r + \log(1/\delta)}{M_1}}.$$

Let us now focus on $\mathbb{E}_w[\|\Pi^* R_2(w)\|^j]^{1/j}$. Define $\Delta_m := \sigma'(\langle w, c_d H \tilde{x}_m + \tilde{\epsilon}_m \rangle) - \sigma'(\langle w, c_d H \tilde{x}_m \rangle)$. Then,

$$\mathbb{E}_w[\|\Pi^* R_2(w)\|^j]^{1/j} = \mathbb{E}_w \left[ \left\| \frac{\eta_1^D c_d}{M_!} \sum_m H \tilde{x}_m \Delta_m \right\|^j \right]^{1/j} \leq \frac{\eta_1^D c_d}{M_1} \sum_m \|H \tilde{x}_m\| \mathbb{E}_w[\|\Delta_m\|^j]^{1/j}.$$

Since $\Delta_m$ only takes values in $\{-1, 0, 1\}$, $\mathbb{E}_w[\|\Delta_m\|^j]^{1/j} = P(\Delta_m \neq 0)^{1/j}$. Denote $\theta_m$ the angle between $c_d H \tilde{x}_m + \tilde{\epsilon}_m$ and $c_d H \tilde{x}_m$, and $s_m := c_d H \tilde{x}_m$, then $P(\Delta_m \neq 0)^{1/j} = (\theta/\pi)^{1/j}$, since $w$ is isotropic. If $\|\tilde{\epsilon}_m\| \leq \frac{1}{2} \|s_m\|$, then $\|s_m + \tilde{\epsilon}_m\| \geq \|s_m\| - \|\tilde{\epsilon}_m\| \geq \frac{1}{2} \|s_m\|$ and

$$\theta_m \leq \frac{\pi}{2} \sin(\theta_m) = \frac{\|(I - \mathrm{Proj}_z)(s_m + \epsilon_m)\|}{\|s_m + \tilde{\epsilon}_m\|} \leq \frac{\|\tilde{\epsilon}_m\|}{\|s_m + \tilde{\epsilon}_m\|} \leq 2 \frac{\|\tilde{\epsilon}_m\|}{\|s_m\|}.$$

Now, let us define a truncation $\tau := \frac{\lambda_{\min} c_d}{4\alpha}$ ($\alpha > 0$), $\mathcal{B} := \{m \mid \|s_m\| \leq \tau\}$ and $\mathcal{G} := \{m \mid \|s_m\| \geq \tau\}$. Particularly, for $m \in \mathcal{G}$, $\|\tilde{\epsilon}_m\| / \|s_m\| \leq \|\tilde{\epsilon}_m\| / \tau = 4\alpha \|\tilde{\epsilon}_m\| / (\lambda_{\min} c_d)$. Under the condition, $\max_m \|\tilde{\epsilon}_m\| \leq \tau/2$, we have

$$\begin{aligned}
\mathbb{E}_w[\|\Pi^* R_2(w)\|^j]^{1/j} &= \frac{\eta_1^D}{M_1} \sum_m \|s_m\| P(\Delta_m \neq 0)^{1/j} \\
&= \frac{\eta_1^D}{M_1} \sum_{m \in \mathcal{B}} \|s_m\| P(\Delta_m \neq 0)^{1/j} + \frac{\eta_1^D}{M_1} \sum_{m \in \mathcal{G}} \|s_m\| P(\Delta_m \neq 0)^{1/j} \\
&\lesssim \frac{\eta_1^D}{M_1} \sum_{m \in \mathcal{B}} \|s_m\| + \frac{\eta_1^D}{M_1} \sum_{m \in \mathcal{G}} \|s_m\| (\|\tilde{\epsilon}_m\| / \|s_m\|)^{1/j} \\
&\lesssim \eta_1^D \tau + \frac{\eta_1^D}{M_1} \sum_{m \in \mathcal{G}} \|s_m\| \left( \frac{4\alpha \max_m \|\tilde{\epsilon}_m\|}{\lambda_{\min} c_d} \right)^{1/j}.
\end{aligned}$$

For $\|H\tilde{x}_m\|$, we have $\mathrm{P}(\|H\tilde{x}_m\|/\lambda_{\max} \geq \sqrt{r} + t) \leq \mathrm{e}^{-t^2/2}$. As a result, with probability $1 - \delta'$,

$$\frac{1}{M_1}\sum_m \|H\tilde{x}_m\| \lesssim \left(\sqrt{r} + \sqrt{\frac{\log(1/\delta')}{M_1}}\right)\lambda_{\max}.$$

Combining these result with Corollary D.10, we obtain with probability $1 - \delta$,

$$\mathbb{E}_w[\|\Pi^* R_2(w)\|^j]^{1/j} \lesssim \frac{\eta_1^D c_d \lambda_{\min}}{4\alpha}$$
$$+ \eta_1^D c_d \lambda_{\max}^2 \left(\sqrt{r} + \sqrt{\frac{\log(2/\delta')}{M_1}}\right)\left(\frac{4\alpha(rd^{-\frac{3}{2}}\sqrt{\log(2M_1/\delta)} + dN^{-\frac{1}{2}} + dJ^{*-\frac{1}{2}})}{\lambda_{\min} c_d}\right)^{1/j}.$$

$\max_m \|\epsilon_m\| \leq \tau/2$ leads to $d \geq \frac{144\alpha^2 r^2 \log(2M/\delta)}{\lambda_{\min}^2}$, $N \geq \frac{144\alpha^2 d^4}{\lambda_{\min}^2}$ and $J^* \geq \frac{144\alpha^2 d^4}{\lambda_{\min}^2}$.

Finally, $\mathbb{E}_w[\|\Pi^* R_3(w)\|^j]^{1/j}$ is bounded by $\eta_1^D \cdot \tilde{O}(rd^{-\frac{3}{2}} + dN^{-\frac{1}{2}} + dJ^{*-\frac{1}{2}})$ since $\sigma' \leq 1$ and $\max\|\tilde{\epsilon}_m\| = \tilde{O}(rd^{-\frac{3}{2}} + dN^{-\frac{1}{2}} + dJ^{*-\frac{1}{2}})$.

$\square$

**Corollary D.16.** *With probability* $1 - \mathrm{e}^{-\iota}$, *if* $M_1 \gtrsim \left(\frac{\lambda_{\max} \eta_1^D c_d \sqrt{\iota}}{\delta}\right)^2 r$, $\alpha = \frac{\eta_1^D c_d \lambda_{\min}}{\delta}$ *and*

$$\epsilon \lesssim \left(\eta_1^D c_d \lambda_{\max}^2 (\sqrt{r} + \sqrt{\iota/M_1})/\delta\right)^{-j} c_d \lambda_{\min}/\alpha \wedge \delta/\eta_1^D,$$

*then*

$$\mathbb{E}_w[\|\Pi^* r(w)\|^j]^{1/j} \lesssim \delta.$$

We can now state the analogue of Lemma 21 (Damian et al., 2022).

**Lemma D.17** (Analogue of Lemma 21 (Damian et al., 2022)). *For any* $k \leq p$, *if* $\eta_1^D = \Theta(\sqrt{d})$, $M \geq \tilde{\Omega}(\hat{\kappa}_p^2 \lambda_{\min}^2 \lambda_{\max}^2 r^2)$, $d \geq \tilde{\Omega}(\hat{r}_p^2 r^2 \vee r^3 \lambda_{\min}^2 \hat{\kappa}_p^2)$, $N \geq \tilde{\Omega}(\hat{r}_p^2 d^4 \vee r \lambda_{\min}^2 \hat{\kappa}_p^2 d^4)$, $J^* \geq \tilde{\Omega}(\hat{r}_p^2 d^4 \vee r \lambda_{\min}^2 \hat{\kappa}_p^2 d^4)$, *where* $\hat{r}_p = r^{4p+1/2}\hat{\kappa}_p^{4p+1}\lambda_{\min}^{4p+1}\lambda_{\max}^{8p+1}$, $\hat{\kappa}_p = \max_{1 \leq i \leq 4k, 1 \leq k \leq p} \kappa^{k/i}$ *and* $\kappa = \lambda_{\max}/\lambda_{\min}$ *then, with high probability*

$$\mathrm{Mat}\left(\mathbb{E}_w\left[(\Pi^* g_M(w))^{\otimes 2k}\right]\right) \succeq (\alpha_d^{-2} r \kappa^2)^{-k}\Pi_{\mathrm{Sym}^k(S^*)},$$

*where* $\Pi_{\mathrm{Sym}^k(S^*)}$ *is the orthogonal projection onto symmetric $k$ tensors restricted to $S^*$, and* $\alpha_d^{-2} = \tilde{\Theta}((\eta_1^D c_d)^{-2}) = \tilde{\Theta}(d)$.

*Proof.* Following Damian et al. (2022), it suffices to prove for all symmetric $k$ tensor $T$ with $\|T\|_F^2 = 1$ that

$$\mathbb{E}_w[\langle T, (\Pi^* g_M(w))^{\otimes k}\rangle^2] \gtrsim (\alpha_d^{-2} r \kappa^2)^{-k}.$$

Since $g_M(w) = g(w) + r(w)$, the binomial theorem leads to

$$\mathbb{E}_w[\langle T, (\Pi^* g_M(w))^{\otimes k}\rangle^2] \geq \frac{1}{2}\mathbb{E}_w[\langle T, (g(w))^{\otimes k}\rangle^2] - \mathbb{E}_w[\delta(w)^2],$$

where we used Young's inequality, $\Pi^* g(w) = g(w)$ and $\delta \lesssim \sum_{i=1}^k \|T(g(w)^{\otimes k-i})\|_F \|\Pi^* r(w)\|^i$.

From Lemma D.13,

$$\mathbb{E}_w[\delta(w)^2] \lesssim \mathbb{E}_w[\langle T, (g(w))^{\otimes k}\rangle^2]\sum_{i=1}^k (\alpha_d^{-2} r \lambda_{\min}^2 \kappa^{2k/i}\mathbb{E}_w[\|\Pi^* r(w)\|^{4i}]^{1/2i})^i.$$

Therefore, if

$$\mathbb{E}_w[\|\Pi^* r(w)\|^j]^{1/j} \leq \frac{1}{4}\alpha_d/(\sqrt{r}\lambda_{\min}\hat{\kappa}_p) \leq \frac{1}{4}\alpha_d/(\sqrt{r}\lambda_{\min}\kappa^{k/i}), \quad \forall j \leq 4k,$$

where $\hat{\kappa}_p = \max_{1 \le i \le 4k, 1 \le k \le p} \kappa^{k/i}$, we obtain our result. This is satisfied by substituting $\delta = \frac{1}{4}\alpha_d/(\sqrt{r}\lambda_{\min}\hat{\kappa}_p)$ in Corollary D.16. This notably implies

$$M_1 \gtrsim \hat{\kappa}_p^2 \lambda_{\min}^2 \lambda_{\max}^2 r^2,$$

and

$$\epsilon \lesssim \frac{1}{\hat{r}_p d} \wedge \frac{c_d}{\sqrt{r}\lambda_{\min}\hat{\kappa}_p},$$

where $\hat{r}_p := r^{4p+1/2}\hat{\kappa}_p^{4p+1}\lambda_{\min}^{4p+1}\lambda_{\max}^{8p+1}$.[9] The last inequality is satisfied when

$$d \gtrsim \hat{r}_p^2 r^2 \iota, \quad N \gtrsim \hat{r}_p^2 d^4, \quad J^* \gtrsim \hat{r}_p^2 d^4.$$

As a result,

$$\mathbb{E}_w[\delta(w)^2] \lesssim \frac{1}{4}\mathbb{E}_w[\langle T, (g(w))^{\otimes k}\rangle^2],$$

and this leads to

$$\mathrm{Mat}\left(\mathbb{E}_w\left[(\Pi^* g_M(w))^{\otimes 2k}\right]\right) \succeq (\alpha_d^{-2} r \kappa^2)^{-k} \Pi_{\mathrm{Sym}^k(S^*)}.$$

Since $\alpha_d = \tilde{\Theta}(\eta_1^D c_d)$ and $c_d = \Theta(1/d)$, we can set $\eta_1^D = \Theta(\sqrt{d})$ so that $\alpha_d^{-2} = \Theta(d)$, which leads to the desired result. $\qquad\square$

The following corollary immediately follows from Corollary 22 of Damian et al. (2022).

**Corollary D.18** (Analogue of Corollary 22 (Damian et al., 2022))**.** *If $M \ge \tilde{\Omega}(\hat{\kappa}_p^2 \lambda_{\min}^2 \lambda_{\max}^2 r^2)$, $d \ge \tilde{\Omega}(\hat{r}_p^2 r^2 \vee r^3 \lambda_{\min}^2 \hat{\kappa}_p^2)$, $N \ge \tilde{\Omega}(\hat{r}_p^2 d^4 \vee r\lambda_{\min}^2 \hat{\kappa}_p^2 d^4)$, $J^* \ge \tilde{\Omega}(\hat{r}_p^2 d^4 \vee r\lambda_{\min}^2 \hat{\kappa}_p^2 d^4)$, where $\hat{r}_p = r^{4p+1/2}\hat{\kappa}_p^{4p+1}\lambda_{\min}^{4p+1}\lambda_{\max}^{8p+1}$ and $\hat{\kappa}_p = \max_{1 \le i \le 4k, 1 \le k \le p} \kappa^{k/i}$, then for any $k \le p$ and any symmetric $k$ tensor $T$ supported on $S^*$, there exists $z_T(w)$ such that*

$$\mathbb{E}_w[z_T(w)(\langle g_M(w), x\rangle^p] = \langle T, x^{\otimes k}\rangle,$$

*with $\mathbb{E}_w[z_T(w)^2] \lesssim (dr\kappa^2)^k \|T\|_F^2$ and $|z_T(w)| \lesssim (dr\kappa^2)^k \|T\|_F^2 \|g_M(w)\|^k$.*

Next, in order to provide the analogue of Lemma 23 (Damian et al., 2022), we observe the following statement.

**Lemma D.19.** *If $\eta_1^R = \tilde{\Theta}(\sqrt{d}/r)$, $N \ge \tilde{\Omega}(d^4)$ and $J^* \ge \tilde{\Omega}(d^4)$, with high probability, we have*

$$\eta_1^R \|g_M(w)\| \le 1, \quad 2\eta_1^R \langle g_M(w), x_i\rangle \le 1.$$

*Proof.* First of all,

$$
\begin{aligned}
\|g_M(w)\| &\le \left\|\eta_1^D \frac{1}{M}\sum_m (c_d H\tilde{x}_m^{(0)} + \epsilon_m)\sigma'(\langle w, c_d H\tilde{x}_m^{(0)} + \epsilon_m\rangle)\right\| \\
&\le \eta_1^D \frac{1}{M}\sum_m \left\|(c_d H\tilde{x}_m^{(0)} + \epsilon_m)\right\| \\
&\le \eta_1^D c_d \frac{1}{M}\sum_m \left\|H\tilde{x}_m^{(0)}\right\| + \eta_1^D \frac{1}{M}\sum_m \|\epsilon_m\| \\
&= \tilde{\Theta}\left(\sqrt{r\iota/(Md)}\right) + \sqrt{d}\max_m \|\epsilon_m\|.
\end{aligned}
$$

From Corollary D.10, the second term is sufficiently small for large $d$, $N$ and $J^*$. The first term can be made small by multiplying with a factor $\eta_1^R = \tilde{\Theta}(\sqrt{d}/\sqrt{r})$ with sufficiently small constant.

Note that $g_M(w)$ depends on the training data $x_i$, through $\{\tilde{x}_m^{(1)}\}$.

---

[9] Here, we assumed $\lambda_{\max}\lambda_{\min} \ge 1$.

$$\langle g_M(w), x_i \rangle = \left\langle \eta_1^D \frac{1}{M} \sum_m c_d H \tilde{x}_m^{(0)} \sigma'(\langle w, c_d H \tilde{x}_m^{(0)} + \epsilon_m \rangle), x_i \right\rangle + \left\langle \eta_1^D \frac{1}{M} \sum_m \epsilon_m^{(i)} \sigma'(\langle w, c_d H \tilde{x}_m^{(0)} + \epsilon_m^{(i)} \rangle), x_i \right\rangle$$

$$= \left\langle \eta_1^D \frac{1}{M} \sum_m c_d H \tilde{x}_m^{(0)} \sigma'(\langle w, c_d H \tilde{x}_m^{(0)} + \epsilon_m \rangle), \Pi^* x_i \right\rangle + \left\langle \eta_1^D \frac{1}{M} \sum_m \epsilon_m^{(i)} \sigma'(\langle w, c_d H \tilde{x}_m^{(0)} + \epsilon_m^{(i)} \rangle), x_i \right\rangle$$

$$\leq 2 \|\Pi^* x_i\| \eta_1^D c_d \frac{1}{M} \sum_m \|H \tilde{x}_m^{(0)}\| + \|x_i\| \eta_1^D \frac{1}{M} \sum_m \|\epsilon_m^{(i)}\|.$$

Now, since $\|x_i\| = \sqrt{d\iota}$, $\|\Pi^* x_i\| = \sqrt{r\iota}$, $\frac{1}{M} \sum_m \|H \tilde{x}_m^{(0)}\| = (\sqrt{r} + \sqrt{\iota/M}) \lambda_{\max}$ and $\frac{1}{M} \sum_m \|\epsilon_m^{(i)}\| = \tilde{O}(d^{-1} \sqrt{r \log(2M/\delta)} + dN^{-\frac{1}{2}} + dJ^{*-\frac{1}{2}})$ with high probability, as long as $N \geq \tilde{\Omega}(d^4)$, $J^* \geq \tilde{\Omega}(d^4)$, $\eta_1^R = \tilde{\Theta}(r^{-1}\sqrt{d})$ is sufficient to assure $2\eta_1^R \langle g_M(w), x_i \rangle \leq 1$. $\qquad\square$

Let us now state the analogue of Lemma 23 (Damian et al., 2022).

**Lemma D.20** (Analogue of Lemma 23 (Damian et al., 2022)). *If $\eta_1^D = \tilde{\Theta}(\sqrt{d})$, $\eta_1^R = \tilde{\Theta}(r^{-1}\sqrt{d})$, $M \geq \tilde{\Omega}(\hat{\kappa}_p^2 \lambda_{\min}^2 \lambda_{\max}^2 r^2)$, $d \geq \tilde{\Omega}(\hat{r}_p^2 r^2 \vee r^3 \lambda_{\min}^2 \hat{\kappa}_p^2)$, $N \geq \tilde{\Omega}(\hat{r}_p^2 d^4 \vee r \lambda_{\min}^2 \hat{\kappa}_p^2 d^4 \vee d^4)$, $J^* \geq \tilde{\Omega}(\hat{r}_p^2 d^4 \vee r \lambda_{\min}^2 \hat{\kappa}_p^2 d^4 \vee d^4)$, where $\hat{r}_p = r^{4p+1/2} \hat{\kappa}_p^{4p+1} \lambda_{\min}^{4p+1} \lambda_{\max}^{8p+1}$ and $\hat{\kappa}_p = \max_{1 \leq i \leq 4k, 1 \leq k \leq p} \kappa^{k/i}$, then for any $k \leq p$ and any symmetric $k$ tensor $T$ supported on $S^*$, there exists $h_T(a, w, b)$ such that if*

$$f_{h_T}(x) := \mathbb{E}_{a,w,b}[h_T(a, w, b) \sigma(\langle w^{(1)}, x \rangle + b)],$$

*we have $\frac{1}{N} \sum_n (f_{h_T}(x_n) - \langle T, x_n^{\otimes p} \rangle)^2 \lesssim 1/n$ with*

$$\mathbb{E}_{a,w,b}[h_T(a, w, b)^2] \lesssim r^{3k} \kappa^{2k} \iota^{3k} \|T\|_F^2,$$
$$\sup_w |h_T(a, w, b)| \lesssim r^{3k} \kappa^{2k} \iota^{6k} \|T\|_F^2.$$

*Remark* D.21. Note that the exponent of $r$ in the bounds of $\mathbb{E}_{a,w,b}[h_T(a, w, b)^2]$ and $\sup_w |h_T(a, w, b)|$ is $3k$, while in Damian et al. (2022) it was $k$.

To conclude, we obtain the following theorem for the teacher training.

**Theorem D.22.** *Under the assumptions of Theorem D.9, with parameters $\eta_1^D = \tilde{\Theta}(\sqrt{d})$, $\eta_1^R = \tilde{\Theta}(r^{-1}\sqrt{d})$, $\lambda_1^R = 1/\eta_1^R$, $\lambda_1^D = 1/\eta_1^D$, $M \geq \tilde{\Omega}(\hat{\kappa}_p^2 \lambda_{\min}^2 \lambda_{\max}^2 r^2)$, $d \geq \tilde{\Omega}(\hat{r}_p^2 r^2 \vee r^3 \lambda_{\min}^2 \hat{\kappa}_p^2)$, $N \geq \tilde{\Omega}(\hat{r}_p^2 d^4 \vee r \lambda_{\min}^2 \hat{\kappa}_p^2 d^4 \vee d^4)$, $J^* \geq \tilde{\Omega}(\hat{r}_p^2 d^4 \vee r \lambda_{\min}^2 \hat{\kappa}_p^2 d^4 \vee d^4)$, where $\hat{r}_p = r^{4p+1/2} \hat{\kappa}_p^{4p+1} \lambda_{\min}^{4p+1} \lambda_{\max}^{8p+1}$ and $\hat{\kappa}_p = \max_{1 \leq i \leq 4k, 1 \leq k \leq p} \kappa^{k/i}$, there exists $\lambda_2^{Tr}$ such that if $\eta_2^{Tr}$ is sufficiently small and $T = \tilde{\Theta}(\{\eta_2^{Tr} \lambda_2^{Tr}\}^{-1})$ so that the final iterate of the teacher training at $t = 2$ output a parameter $a^{(\xi_2^{Tr})}$ that satisfies with probability at least 0.99,*

$$\mathbb{E}_{x,y}[|f_{(a^{(\xi_2^{Tr})}, W^{(1)}, b^{(1)})}(x) - y|] - \zeta \leq \tilde{O}\left(\sqrt{\frac{dr^{3p}\kappa^{2p}}{N}} + \sqrt{\frac{r^{3p}\kappa^{2p}}{L^*}} + \frac{1}{N^{1/4}}\right).$$

## D.5. $t = 2$ Distillation and Retraining

Finally, the behavior of distillation and retraining at $t = 2$ is similar to the single index model case. Indeed, we prepare $\mathcal{D}_2^S = \{\hat{x}_m^{(0)}, \hat{y}_m^{(0)}\}_{m=1}^{M_2}$ so that it satisfies the regularity condition B.36. Please see Appendix B.8 for further details.

**Theorem D.23.** *Under the assumptions of Theorem D.22, if $\{\hat{x}_m^{(0)}\}$ satisfies the regularity condition B.36, $\eta_1^S = M_1 \eta_1^{Tr}$, then for a sufficiently small $\eta_2^D$ and $\xi_2^R = \tilde{\Theta}((\eta_2^D \sigma_{\min})^{-1})$, where $\sigma_{\min} > 0$ is the smallest eigenvalue of $\tilde{K}^\top \tilde{K}$, the one-step gradient matching (17) finds $M$ labels $\tilde{y}_m^{(1)}$ so that*

$$\mathbb{E}_{x,y}[|f_{(\tilde{a}^{(\xi_1^S)}, W^{(1)}, b^{(0)})}(x) - y|] - \zeta \leq \tilde{O}\left(\sqrt{\frac{dr^{3p}\kappa^{2p}}{N}} + \sqrt{\frac{r^{3p}\kappa^{2p}}{L^*}} + \frac{1}{N^{1/4}}\right),$$

*The overall memory cost is $\tilde{\Theta}(r^2 d + L)$.*

*Table 5.* The performance of dataset distillation and overlap with respect to the number of distilled images.

| Dataset | Metric | $M = 10$ | $M = 100$ | $M = 500$ |
|---------|--------|----------|-----------|-----------|
| MNIST | Acc. | 0.92 | 0.97 | 0.99 |
| | Overlap | 0.91 | 0.93 | 0.94 |
| CIFAR10 | Acc. | 0.28 | 0.44 | 0.54 |
| | Overlap | 0.27 | 0.51 | 0.66 |

**Theorem D.24.** *Under the assumptions of Theorem D.22, if $\{\hat{x}_m^{(0)}\}$ satisfies the regularity condition B.36, then one-step performance matching can find with high probability a distilled dataset at the second step of distillation so that*

$$\mathbb{E}_{x,y}[\|f_{(\tilde{a}^{(1)}, W^{(1)}, b^{(0)})}(x) - y\|] - \zeta \leq \tilde{O}\left(\sqrt{\frac{dr^{3p}\kappa^{2p}}{N}} + \sqrt{\frac{r^{3p}\kappa^{2p}}{L^*}} + \frac{1}{N^{1/4}}\right),$$

*where $\tilde{a}^{(1)}$ is the output of the retraining algorithm at $t = 2$ with $\tilde{a}^{(0)} = 0$. The overall memory usage is only $\tilde{\Theta}(r^2 d + L)$.*

### D.6. Summary of Algorithm

We first summarize the setting of each hyperparameter that was not provided in the assumptions.

- $\eta_1^D = \tilde{\Theta}(\sqrt{d})$, $\lambda_1^D = 1/\eta_1^D$,

- $\eta_1^R = \tilde{\Theta}(d)$ for single index models with $M_1 = 1$, and else $\eta_1^R = \tilde{\Theta}(r^{-1}\sqrt{d})$, $\lambda_1^R = 1/\eta_1^R$,

- $\xi_2^{Tr} = \tilde{\Theta}(1/(\eta_2^{Tr}\lambda_2^{Tr}))$, sufficiently small $\eta_2^{Tr}$, there exist a $\lambda_2^{Tr}$,

- $\eta_2^S = \eta_2^R$, $\lambda_2^S = 0$,

- for one-step PM, $\xi_2^D = \tilde{\Theta}(1/(\eta_2^D\lambda_2^D))$, sufficiently small $\eta_2^D$, there exist a $\lambda_2^D$, $\lambda_2^R = 0$,

- for one-step GM, $\eta_2^D = M_2\eta_2^{Tr}$, $\lambda_2^D = 0$, $\xi_2^R = \tilde{\Theta}(1/\eta_2^R)$, sufficiently small $\eta_2^R$, $\lambda_2^R = 0$.

In the following, Algorithm 2 provides the precise formulation of Algorithm 1 with Assumption 3.6 applied.

## E. Image Classification Tasks

In this appendix, we provide additional experiments with real world data and practical models such as convolutional networks to validate our theory. All the experiments follow the setup of Zhao et al. (2021): gradient matching with a 3-layer ReLU ConvNet with average pooling and group normalization. We use their default parameter setup unless stated otherwise.

We start by varying distilled set sizes $M$. We measured both distillation test accuracy (Acc.) and the overlap between the leading task-relevant feature subspace of the real data and that of the distilled data (Overlap), computed from penultimate-layer representations of a reference network. The reported accuracy values follow the setup of (Zhao et al., 2021). Results is shown in Table 5. For the overlap metric, 1 indicates perfect overlap and 0 indicates orthogonal subspaces. Stronger subspace preservation correlates with higher distilled performance, which supports our hypothesis that low-dimensional structure is encoded into the distilled data.

Next, our theoretical analysis predicts that increasing the training data size ($N$), network width ($L$), and distilled dataset size should improve DD performance. We report in Tables 6 and 7 the mean over 5 random seeds. Consistent with our theory, increasing $N$ and $L$ leads to improved DD performance.

---

**Algorithm 2** Analyzed Dataset Distillation Algorithm

---

**Input:** Training dataset $\mathcal{D}^{Tr}$, model $f_\theta$, loss function $\mathcal{L}$, fixed initial parameter $\theta^{(0)} = \left(a^{(0)}, W^{(0)}, b^{(0)}\right)$.

**Output:** Distilled data $\mathcal{D}^S$.

Initialize distilled data $\mathcal{D}_0^S$ randomly, $\alpha = \frac{1}{N} \sum_{n=1}^N y_n$, $\gamma = \frac{1}{N} \sum_{n=1}^N y_n x_n$, preprocess: $y_n \leftarrow y_n - \alpha - \langle \gamma, x_n \rangle$.

**for** $t = 1$ **do**

    Sample a batch of initial states $\{\theta_j^{(0)}\}_{j=1}^J$, and initialize distilled data $\mathcal{D}^S$ randomly.

    *I. Training Phase*

    **for** $j = 0$ **to** $J$ **do**

        **Teacher Training** : $W_j^{(1)} = W_j^{(0)} - \eta_1^{Tr} \left\{ \nabla_W \mathcal{L}(\theta_j^{(0)}, \mathcal{D}^{Tr}) + \lambda_1^{Tr} W_j^{(0)} \right\}$, where $\lambda_1^{Tr} = 1/\eta_1^{Tr}$.

        With teacher gradient : $g_{i,j}^{Tr} = \nabla_{w_i} \mathcal{L}(\theta_j^{(0)}, \mathcal{D}^{Tr})$. $w_{i,j}^{(0)}$ is the realization of the $j$-th initialization of $w_i$.

        **Student Training** : $\tilde{W}_j^{(1)} = W_j^{(0)} - \eta_1^S \left\{ \nabla_W \mathcal{L}(\theta_j^{(0)}, \mathcal{D}^S) + \lambda_1^S W_j^{(0)} \right\}$, where $\lambda_1^S = 1/\eta_1^S$.

        With student gradient: $g_{i,j}^S = \nabla_{w_i} \mathcal{L}(\theta_j^{(0)}, \mathcal{D}^S)$.

    **end for**

    *II. Distillation Phase*

    $\mathcal{D}_1^S = \mathcal{D}^S - \eta_1^D \left( \frac{1}{J} \sum_j \nabla_{\mathcal{D}^S} \left( 1 - \frac{1}{L} \sum_{i=1}^L \langle g_{i,j}^S, g_{i,j}^{Tr} \rangle \right) + \lambda_1^D \mathcal{D}^S \right)$, where $\lambda_1^D = 1/\eta_1^D$.

    *III. Retraining Phase*

    $W^{(1)} = W^{(0)} - \eta_1^R \left\{ \nabla_W \mathcal{L}(\theta^{(0)}, \mathcal{D}_1^S) + \lambda_1^R W^{(0)} \right\}$.

**end for**

Reinitialize $b_i \sim N(0, 1)$

**for** $t = 2$ **do**

    Sample a batch of initial states $\{\theta_j^{(1)}\}_{j=1}^J$ where $\theta_0^{(1)} = (a^{(0)}, W^{(1)}, b^{(0)})$, and a new distilled data $\mathcal{D}^S$.

    *I. Training Phase*

    **for** $j = 1$ **to** $J$ **do**

        **Teacher Training** : $a_j^{(\tau)} = a_j^{(\tau-1)} - \eta_2^{Tr} \left\{ \nabla_a \mathcal{L}((a_j^{(\tau-1)}, W^{(1)}, b^{(0)}), \mathcal{D}^{Tr}) + \lambda_2^{Tr} a_j^{(\tau-1)} \right\}$, from $\tau = 1$ to $\tau = \xi_2^{Tr}$.

        With teacher gradient: $G_{2,j}^{Tr} = \{g_{\tau,j}^{Tr}\}_{\tau=0}^{\xi_2^{Tr}-1} = \left\{ \nabla_a \mathcal{L}((a_j^{(\tau-1)}, W^{(1)}, b^{(0)}), \mathcal{D}^{Tr}) + \lambda_2^{Tr} a_j^{(\tau-1)} \right\}_{\tau=0}^{\xi_2^{Tr}-1}$.

        **Student Training**: $\tilde{a}_j^{(1)} = a_j^{(0)} - \eta_2^S \left\{ \nabla_a \mathcal{L}((a_j^{(0)}, W^{(1)}, b^{(0)}), \mathcal{D}^S) \right\}$.

        With student gradient: $\{g_{0,j}^S\} = \left\{ \nabla_a \mathcal{L}((a_j^{(0)}, W^{(1)}, b^{(0)}), \mathcal{D}^S) \right\}$.

    **end for**

    **if** One-Step Gradient Matching **then**

        *II. Distillation Phase*

        $\mathcal{D}_2^S = \mathcal{D}^S - \eta_2^S \frac{1}{J} \sum_{j=1}^J \nabla_{\mathcal{D}^S} \left( 1 - \left\langle \sum_{\tau=0}^{\xi_2^D-1} g_{\tau,j}^{Tr}, g_{0,j}^S \right\rangle \right)$.

        *III. Retraining Phase*

        $\tilde{a}^{(\tau)} = \tilde{a}^{(\tau-1)} - \eta_2^R \nabla_a \left( \mathcal{L}\left((a^{(\tau-1)}, W^{(1)}, b^{(0)}), \mathcal{D}_2^S\right) \right)$ from $\tau = 1$ to $\tau = \xi_2^R$.

    **end if**

    **if** One-Step Performance Matching **then**

        *II. Distillation Phase*

        **for** $\tau = 1$ **to** $\xi_2^R$ **do**

            $\mathcal{D}^S \leftarrow \mathcal{D}^S - \eta_2^R \left( \nabla_{\mathcal{D}^S} \mathcal{L}\left((\tilde{a}_0^{(1)}, W^{(1)}, b^{(0)}), \mathcal{D}^{Tr}\right) + \lambda_2^R \mathcal{D}^S \right)$, where $\tilde{a}_0^{(1)}$ is a function of $\mathcal{D}^S$.

        **end for**

        $\mathcal{D}_2^S = \mathcal{D}^S$.

        *III. Retraining Phase*

        $\tilde{a}^{(1)} = a^{(0)} - \eta_2^R \left\{ \nabla_a \mathcal{L}((a^{(0)}, W^{(1)}, b^{(0)}), \mathcal{D}_2^S) \right\}$.

    **end if**

**end for**

**return** $\mathcal{D}_1^S \cup \mathcal{D}_2^S$

---

*Table 6.* The performance of dataset distillation (testing accuracy) with respect to the number of training data in the teacher training. We created 10 distilled images per class.

| Dataset | $N = 10^2$ | $N = 10^3$ | $N = 10^4$ |
|---------|-----------|-----------|-----------|
| MNIST | 0.92 | 0.95 | 0.96 |
| SVHN | 0.70 | 0.75 | 0.75 |
| CIFAR10 | 0.27 | 0.39 | 0.43 |

*Table 7.* The performance of dataset distillation (testing accuracy) with respect to the network width. We created 10 distilled images per class.

| Dataset | $L = 32$ | $L = 64$ | $L = 128$ |
|---------|----------|----------|-----------|
| MNIST | 0.95 | 0.96 | 0.96 |
| SVHN | 0.71 | 0.74 | 0.75 |
| CIFAR10 | 0.39 | 0.43 | 0.44 |

