# OpenReview forum: "Dataset Distillation Efficiently Encodes Low-Dimensional Representations from Gradient-Based Learning of Non-Linear Tasks"
_ICML.cc/2026/Conference — ICML 2026 regular_

### Official Review · Reviewer_egzy · 2026-03-08

**Soundness:** 4
**Presentation:** 3
**Significance:** 3
**Originality:** 4
**Overall Recommendation:** 5
**Confidence:** 5

**Summary:**

This paper presents a theoretical study of dataset distillation in a nonlinear, gradient-based learning setting. The authors analyze two-layer ReLU networks on multi-index models, where the target task has a low intrinsic dimension. The main claim is that distilled data can efficiently encode the latent low-dimensional structure of the task. Based on this view, the paper develops a progressive two-phase interpretation of distillation and derives a memory complexity bound, which is significantly smaller than the storage cost of the original training process. The experiments mainly serve to illustrate the theoretical picture and a transfer-learning interpretation.

**Compliance With Llm Reviewing Policy:**

Affirmed.

**Final Justification:**

After reading the rebuttal and considering the other reviewers’ comments, I find that my concerns have been well addressed. The authors provided satisfactory clarifications, and I will therefore maintain my Accept recommendation.

**Key Questions For Authors:**

1. The first-stage latent structure encoding result is the most convincing part of the paper. How should readers interpret the second-stage result relative to it? Do the authors view the second-stage compact reconstruction as equally central, or as a more conditional extension that depends on stronger assumptions?

2. The paper gives a strong memory complexity result. Can the authors comment more explicitly on the computation-memory tradeoff in their construction?

3. How essential is the multi-index model assumption to the main phenomenon? Do the authors believe the same latent-structure encoding mechanism should extend to broader nonlinear tasks with low intrinsic dimension, or is the current proof strategy tightly coupled to this specific model class?

4. The experiments are framed as illustrations of the theory and of a transfer-learning perspective. It would help if the authors stated more explicitly which empirical observations they see as testing the theory most directly, and which ones are mainly qualitative demonstrations.

**Limitations:**

Yes

**Strengths And Weaknesses:**

Strengths:
This paper studies a meaningful theoretical question. Prior theory for dataset distillation has mostly focused on linear or kernel settings, or has not explicitly analyzed the role of the latent task dimension. In contrast, this paper studies gradient-based distillation for a nonlinear model and makes the intrinsic task structure central to the analysis. I find this direction important and timely. The paper also has a coherent technical narrative. The first-stage result, which shows that distilled data encode the principal subspace of the task, is the strongest part of the paper. This gives a concrete mechanism for why distillation can compress training information in structured tasks. The final memory bound is also interesting because it ties compression directly to the latent dimension rather than only the ambient dimension.

Weaknesses:
1. The analysis relies on two-layer ReLU networks, multi-index models, progressive distillation phases, and several carefully structured assumptions. These assumptions do not make the paper invalid, but they do narrow the scope of the conclusions. The paper convincingly explains dataset distillation in this controlled regime, but it does not yet show that the same mechanism extends to more general modern deep learning settings.

2. The first-stage latent structure encoding result is clear and compelling. By contrast, the second-stage reconstruction story depends on additional regularity conditions and a more delicate construction. Because of this, the paper’s strongest contribution is the explanation of how the principal subspace is encoded, while the full end-to-end reconstruction interpretation feels more conditional.

3. The memory complexity result is strong and interesting. However, the construction can still require substantial computation, especially through initialization and batch-size choices. As a result, the theory should be interpreted as explaining storage compression and representational efficiency, not as a full explanation of why dataset distillation is computationally efficient in practice.

4. The experiments mainly illustrate the theory rather than strongly validate it. This is acceptable for a theory paper, but it should be stated more explicitly. The experiments support the proposed interpretation, but they do not provide broad evidence for practical competitiveness or for the generality of the claimed mechanism beyond the analyzed setting.

---

> ### Author Rebuttal · Authors · 2026-03-30
>
> Thank you very much for spending your time on carefully reviewing our paper and for the thoughtful and supportive assessment.
>
> (W = Weakness)
> ### Q1
> The first-stage latent structure encoding result is indeed the conceptual core of the paper. The second-stage should be interpreted as follows: it illustrates how end-to-end distillation can succeed and explains how progressive dataset distillation (DD) can achieve compact overall memory complexity. We will revise the paper to make this hierarchy clearer.
> ### Q2 (and W3)
> We appreciate this important point. In terms of “efficiency”, the main goal and message of this paper is to explain the storage compression and representational efficiency of distilled points, rather than to provide a computationally optimized *construction* or to prove the computational efficiency of the full distillation pipeline, which usually requires substantial computation. In our theory, this appears through the initialization size $J$ and training data size $N$, which precisely reflects the broader computation-memory trade-off in DD: stronger summaries of the training process can reduce storage, but it requires more computation to construct them. While overall computational efficiency is an important research direction, it is equally important to understand what information can be distilled with practical DD algorithms and why such distilled representations are useful in nontrivial settings. This paper focuses on those questions and provides one of the first theoretical analyses in this direction. We hope that it can serve as a basis for subsequent work aimed both at deeper understanding of DD and at practical improvements. We will make this distinction more explicit in the revision.
> ### W2
> We understand the reviewer’s concern that the additional regularity condition for the second stage may appear more restrictive. Related to our two responses above, its purpose is to characterize when second-stage distillation can successfully yield a compact distilled set. We note that this condition should be naturally satisfied as, intuitively, increasing the number of rows cannot decrease the rank of $\tilde K$. Therefore, with sufficiently many distilled examples, the condition holds with high probability. For the initialization construction proposed in Appendix B.8, we plan to add additional experiments to illustrate this point: for widths ranging from $10$ to $10^5$, dataset sizes $M_2$ from $10$ to $10^2$, and intrinsic dimensions $r=3$ or $10$, the maximal attainable rank is achieved in most cases. Moreover, for completeness, we will add another construction that *provably* satisfies the regularity condition. In the notation of Lemma B.38, this construction is $\\{s_m\\}_m = \\{\pm \tau_i\\}_i$. The proof idea is to choose $\\{s_m\\}_m$ so that $\tilde K$ can be reconstructed to become diagonal with sufficiently many nonzero entries. If useful, we can provide a more detailed proof sketch of this argument in follow-up discussion.
> ### Q3
> Multi-index models are essential in the sense that they provide one of the cleanest regimes in which the latent-structure encoding can be isolated while retaining nontrivial nonlinear interactions of DD. Although the current proof strategy is tied to the multi-index setting, we believe that this underlying encoding mechanism is more general than this specific model class. One may certainly consider broader nonlinear tasks with low-dimensional structure, but this would make the analysis more difficult and would likely require moving to deeper neural networks, where tractable feature-learning analyses remain challenging. While this is an important future direction, it is currently unclear how best to proceed or what additional structural assumptions would be most informative. At present, we believe that the essential phenomenon of latent-subspace information encoding is already captured well by the multi-index model, which allows us to quantify the memory complexity of DD in terms of the internal structure of the task and thereby formalize a previously heuristic understanding.
>
> Related to this question, we also clarify several aspects of W1 in our response to **Reviewer QL9u’s Question 2**. Please refer to that response for further details.
> ### Q4 & W4
> We are grateful for this suggestion and will make the presentation more explicit. The experiments that most directly test the theory are those examining whether the scaling trends align with the predicted dependence in Figure 1. The transfer learning in Figure 2 is intended as an illustrative demonstration of the proposed interpretation, rather than as a direct test of the full theory.
>
> Please also refer to our responses to **Reviewer MJes’s Question 2** and **Reviewer Umuq’s Question 2** for further experiments and discussion of the practical implications of the theory.
>
> We hope these clarifications address the reviewer’s questions and concerns, and we would be happy to provide further detail if helpful.

---

> > ### Author Rebuttal · Reviewer_egzy · 2026-04-01
> >
> > After reading the rebuttal and considering the other reviewers’ comments, I find that my concerns have been well addressed. The authors provided satisfactory clarifications, and I will therefore maintain my Accept recommendation.

---

> > > ### Author Response · Authors · 2026-04-03
> > >
> > > We thank the reviewer for the positive feedback and for acknowledging that the concerns were well addressed in the rebuttal. We appreciate the careful reading of our work and the constructive comments.

---

### Official Review · Reviewer_Umuq · 2026-03-11

**Soundness:** 3
**Presentation:** 3
**Significance:** 4
**Originality:** 3
**Overall Recommendation:** 4
**Confidence:** 5

**Summary:**

This paper provides a theoretical characterization of dataset distillation (DD) in a setting commonly used in the feature-learning theory literature, namely the multi-index model with a standard two-layer ReLU neural network. It mainly analyzes two classic DD paradigms: gradient matching and performance matching. This work explains how low-dimensional intrinsic structure of real data is encoded into the distilled data during gradient-based training. It also studies how the final-layer directional parameter can be reconstructed from the distilled set in the second stage. The paper derives corresponding memory-complexity and generalization-error upper bounds under certain conditions, and presents synthetic experiments whose trends are consistent with the theory.

**Compliance With Llm Reviewing Policy:**

Affirmed.

**Final Justification:**

The rebuttal is thoughtful and sufficiently addresses my concern within the intended scope of the paper, so I increase the significance score to 4.

**Key Questions For Authors:**

Can the authors clarify to what extent the current framework can cover distribution matching, trajectory matching, and diffusion-model-based prototype distillation?

The empirical evaluation should include real datasets, deeper networks, and other GM-based distillation methods, to verify whether the qualitative claim that ``low-dimensional structure is encoded into the distilled data'' holds in realistic settings.

Can the proposed analysis be extended to study the cross-architecture transferability of distilled datasets?

**Limitations:**

yes

**Strengths And Weaknesses:**

## Strengths:

1. By placing DD in a carefully controlled, analyzable regime, the paper offers a principled answer to the question “what signal is actually distilled from the original dataset into the synthetic set.” It formalizes an optimization chain in which low-dimensional structure is captured through gradients, mapped onto the principal subspace, and then distilled via reconstruction, which is valuable for understanding DD’s learning behavior.

2. The theory of this work is well-developed: it provides a coherent set of results around representation encoding, memory complexity, and generalization error bounds.

## Weaknesses:

1. The methodological coverage is limited. The analysis focuses on the traditional GM/PM formulations and does not speak to more widely used approaches in current DD work, such as distribution matching or trajectory matching. It also does not explain why more recent diffusion-based generative prototype dataset distillation methods can work.

2. The assumptions are somewhat idealized. The Gaussian input and polynomial target in Assumption 3.1 are not typical in real applications, and the network is shallow. In practice, DD is usually performed with deeper nonconvex networks, and real data may contain both task-relevant low-dimensional structure and task-irrelevant (yet still low-dimensional) structure, possibly coupled. The current theory does not distinguish how these different structures are distilled or which one dominates the outcome.

3. The experiments are overly simplistic. They are limited to synthetic data and lack validation on real datasets and realistic distillation settings (deeper networks, commonly used DD algorithms, standard benchmarks), making it hard to assess generality and practical relevance.

4. A key goal of DD is that distilled data should retain utility across different models/architectures. However, the core theory of this work is model-dependent (two-layer ReLU with specific training dynamics), and does not explain cross-architecture transfer of distilled sets.

5. While the paper characterizes how low-dimensional signals are learned and provides complexity and error bounds, it offers limited direct guidance for practice, e.g., how to design better DD algorithms, choose hyperparameters, decide stage schedules, or set initialization strategies.

---

> ### Author Rebuttal · Authors · 2026-03-30
>
> Thank you very much for spending your time on carefully reviewing our paper and for recognizing the value of our feature-learning analysis.
>
> ### Q1 and Weakness 1
>
> We agree that the present paper does not provide a unified theory of all modern DD algorithms. Methods such as distribution matching, trajectory matching, and diffusion-based prototype distillation optimize substantially different objects (i.e., respectively, feature/distribution alignment, path-level training dynamics, and a diffusion objective in addition to the matching loss) so extending the current proof techniques to them is nontrivial and outside the scope of this work. Our focus is instead to provide a first rigorous analysis of practical gradient-based DD mechanisms in a regime where one can mathematically trace how task-relevant low-dimensional structure is encoded into synthetic data, which is why we study GM and PM. These objectives also serve as core building blocks in several more recent approaches (e.g., trajectory matching or diffusion-based DD), and we view our work as a foundational starting point for studying more involved distillation techniques. That said, we believe the representation-level picture identified here may still be informative beyond GM/PM. If these other methods also succeed by preserving task-relevant low-dimensional structure, then our results suggest one possible mechanism for why distilled data can remain informative across paradigms, even though we do not currently prove this. We will clarify this scope more explicitly in the revision.
>
> ### Q2
>
> We appreciate this suggestion, which indeed strengthens the paper. In the table below, we report a real-data experiment using a 3-layer ReLU ConvNet architecture (with average pooling and group normalization) on MNIST and CIFAR10. For varying distilled set sizes $M$, we measured both distillation test accuracy (Acc.) and the overlap between the leading task-relevant feature subspace of the real data and that of the distilled data (Overlap), computed from penultimate-layer representations of a reference network. We used the experimental framework of [1]; the reported accuracy values follow their setup. For the overlap metric, 1 indicates perfect overlap and 0 indicates orthogonal subspaces. Our hypothesis is that stronger subspace preservation should correlate with higher distilled performance, which is indeed what we observe. We will add this result to the revised version of the paper.
>
> | Dataset | Metric | $M=10$ | $M=100$ | $M=500$ |
> |---|---|---:|---:|---:|
> | MNIST | Acc. | 0.917 | 0.974 | 0.988 |
> | MNIST | Overlap | 0.906 | 0.928 | 0.939 |
> | CIFAR10 | Acc. | 0.283 | 0.449 | 0.539 |
> | CIFAR10 | Overlap | 0.273 | 0.512 | 0.660 |
>
> ### Q3 and Weakness 4
>
> We very much appreciate this point. A natural extension toward cross-architecture transferability would be to ask when two architectures induce compatible task-relevant representations, in which case a distilled set encoding those shared directions might transfer across them. In this sense, our theory suggests one possible route to transferability: if different architectures recover similar principal task-aligned subspaces, then distilled data that encode those subspaces may generalize across architectures. Formalizing this would require a comparative representation analysis across model classes, which is beyond the current paper. Here, we study how a distilled set preserves task-relevant structure relative to a specified architecture and training dynamic. This controlled setting is precisely what makes a first rigorous analysis tractable. We agree that extending the theory toward cross-architecture transfer is an important future direction, and one that we would like to pursue.
>
> ### Weakness 2
>
> Please refer to our response to **Reviewer QL9u’s Question 2** for the main clarification on this point.
>
> We would like to add the following remark. We view the simultaneous presence of task-relevant and task-irrelevant low-dimensional structure as a substantial next step beyond the current analysis. Addressing this would require a model in which gradients respond differently to discriminative and nondiscriminative low-dimensional directions, followed by an analysis of which components are preferentially retained by the distillation objective. We believe this is a promising direction for understanding more realistic settings, including questions related to transfer and continual learning.
>
> ### Weaknesses 3 and 5
>
> We appreciate this concern. To better connect the theory to practice, we are also considering adding empirical analyses on real-world datasets and deeper architectures. Please refer to our response to **Reviewer MJes’s Question 2** for further details.
>
> We hope these clarifications address the reviewer’s questions and concerns, and we would be happy to provide further detail if helpful.
>
> [1] Zhao et al., Dataset Condensation with Differentiable Siamese Augmentation, *ICML2021*.

---

> > ### Author Rebuttal · Reviewer_Umuq · 2026-04-01
> >
> > The rebuttal has addressed most of my questions and clarified the intended scope of the paper. In particular, the authors are clear that the current theory focuses on GM/PM in a controlled feature-learning setting, and the added real-data evidence is helpful in supporting the paper’s main qualitative claim that task-relevant low-dimensional structure is preserved in the distilled data. While some broader questions, such as extensions to more modern DD methods and cross-architecture transfer, remain open, I think the rebuttal is overall satisfactory and consistent with the paper’s scope. I therefore choose to keep my score unchanged.

---

> > > ### Author Response · Authors · 2026-04-03
> > >
> > > We thank the reviewer for further recognizing the intended scope of our theoretical framework and our additional experiments in realistic scenarios.
> > >
> > > We agree that extensions to modern dataset distillation methods and cross-architecture transfer are important and promising directions. Although a full treatment is beyond the scope of the present work as noted in our rebuttal, we would like to highlight that we have taken initial steps to connect our framework to deeper architectures and cross-architecture transfer, and we are happy to concisely report this here as it further supports the generality of our theory.
> > >
> > > In particular, our analysis can be interpreted in the context of deeper neural networks by viewing the model as consisting of a feature-learning layer followed by an output function comprising the remaining layers. Under this perspective, the first distilled dataset of Section 4.1 can be used to retrain the feature-learning layer, while the analysis in Section 4.2 extends to the subsequent layers. This suggests that similar distilled datasets capturing the low-dimensionality of the task exist that can be reused across different deeper architectures, expanding our theory to deeper networks and providing a theoretical explanation for cross-architecture transfer as well.
> > >
> > > We will incorporate this clarification into the paper. We do not consider this a full treatment of deeper networks as it remains unclear what fundamentally new theoretical insights would arise beyond the core mechanism already identified in our paper when moving to deeper architectures, which would go beyond the scope of the work. Indeed, the crux of the above discussion is to decompose deep neural networks into two components analogously to two-layer neural networks. Nevertheless, this observation supports our main claim that one of the key mechanisms underlying dataset distillation is the preservation of task-relevant low-dimensional structure, which is not inherently restricted to the specific setting studied in this paper.

---

### Official Review · Reviewer_QL9u · 2026-03-12

**Soundness:** 3
**Presentation:** 2
**Significance:** 2
**Originality:** 3
**Overall Recommendation:** 4
**Confidence:** 3

**Summary:**

This paper provides a rigorous theoretical investigation into the mechanisms of dataset distillation, focusing on a two-layer ReLU neural networks when learning tasks with low-dimensional structures. A core contribution is the demonstration that gradient-based distillation algorithms can effectively extract the "essential" features of complex tasks.

**Compliance With Llm Reviewing Policy:**

Affirmed.

**Final Justification:**

Overall, the rebuttal has addressed most of my technical concerns. The paper provides a solid theoretical contribution to understanding dataset distillation beyond the kernel regime. Therefore, I will maintain my recommandation of weak accept.

**Key Questions For Authors:**

please refer weakness

**Limitations:**

yes

**Strengths And Weaknesses:**

Strengths:

- This is one of the first works to provide a rigorous analysis of DD dynamics that simultaneously accounts for task structure, non-linear model architectures, and finite-time gradient-based optimization.
- It successfully characterizes "feature learning" within the distillation process, proving that synthetic points effectively capture and project the task's principal subspace.
- It provides concrete memory complexity bounds that reflect task difficulty, explaining why simpler tasks require fewer distilled images

Weakness and Questions:

Q1. I have a concern on the last assumption (Assumption 3.8) on the surrogate function of ReLU. While softplus is indeed twice differentiable at 0, many modern networks adopts the non-differentiable ReLU form for purpose of the sparsity property. Therefore, such assumption may limit the generalization of such theoretical framework.

Q2. Also, there is a doubt on the Assumption 3.1 of the iid data distribution. It would be too ideal to assume the real world data have iid data distribution.

Q3. A potential drawback is the increase in the initial batch size would face the scalibility issues, especially when DD is hard to scale.

Overall, I believe it is an interesting paper.

---

> ### Author Rebuttal · Authors · 2026-03-30
>
> Thank you very much for spending your time on carefully reviewing our paper and for the positive assessment. We appreciate the reviewer’s recognition of the novelty of combining task structure, nonlinear models and finite-time gradient-based optimization.
> ### Q1
> We agree that most practical networks use ReLU rather than smooth surrogates such as softplus. The key issue behind Assumption 3.8 is that, in gradient-based DD, distillation involves the computation of the *second derivative of ReLU*, which is not well-defined. This ill-posedness is therefore inherent to gradient-based DD itself, rather than a limitation specific to our analysis. Our intent is not to claim that softplus is the practically dominant or most effective activation function, but rather to work with a well-defined and analyzable operation. In the paper, we address this issue in two ways. The first is to replace ReLU only in the student training at $t=1$ with an arbitrary smooth surrogate satisfying mild conditions. This corresponds to Assumption 3.8. The second is to keep ReLU and instead define a well-posed distillation update directly for the ReLU case. This is treated formally in Appendix C. Importantly, both approaches lead to the same qualitative conclusions in the main theorems. The second option is particularly motivated by the fact that practical frameworks such as PyTorch adopt a specific convention for the ReLU second derivative, which effectively makes DD well-defined in practice. Under this type of ReLU-defined update, we show that the same result continues to hold. In this sense, Appendix C also provides a possible explanation for why DD continues to work well empirically even when ReLU is used. We will revise the manuscript to make this point clearer.
> ### Q2
> Thank you for raising this point. Since related concerns were raised across multiple reviews, we include here a consolidated response to address this issue clearly and consistently.
>
> Our analysis indeed uses several assumptions on the dataset and task structure. We would first like to clarify that we do not claim that real datasets satisfy them exactly, nor do we believe that they do. For example, extending the analysis beyond the i.i.d. setting is certainly an important direction, since real datasets are not perfectly i.i.d. in a strict sense. Our goal is instead to isolate a regime in which task structure is well-defined and governed by task-relevant intrinsic dimension rather than ambient dimension to explain the mechanism and efficiency of DD.
>
> More generally, assumptions of this type on the data (e.g., i.i.d. Gaussian inputs), task structure, and architecture (e.g., two-layer neural networks) are standard in the theory of feature learning rather than being introduced *ad hoc* for this paper (see, e.g., Damian et al., 2022; Abbe et al., 2022; Oko et al., 2024a). This is especially natural here, since our work is, to the best of our knowledge, among the first to formalize and analyze feature learning in the context of DD. Relaxing such assumptions is also an active research direction. For example, related single-index analyses have been extended beyond Gaussian data [1].
>
> Notably, the feature-learning analysis in DD is more involved than in standard supervised training, since DD contains several interacting stages (teacher training, student training, distillation, and retraining) and each stage has its own optimization dynamics rather than a single end-to-end training process as we tried to keep the algorithmic procedure as close as possible to practice using gradient-based formulation. Analyzing feature learning across these coupled stages is therefore substantially more complex, and constituted one of the main technical challenges of the paper. Overall, we believe that the present setting is appropriate for providing one of the first principled explanations of why DD can achieve substantial compression, while also serving as a basis for future theory beyond the current controlled regime. We will revise the paper to make these points clearer.
> ### Q3
> We appreciate this concern. We believe this is the price to pay to obtain strong summaries of the training process (i.e., no free lunch), but tightening the dependence on $d$ of the initialization and training data sizes ($J$ and $N$) is an important problem. More importantly, these quantities should be interpreted as sufficient conditions used to establish the guarantee, rather than as prescriptions for practical DD. Please refer to our response to **Reviewer egzy’s Question 2** for further clarification on this point.
>
> Please also refer to our response to **Reviewer MJes’s Question 2**  where we connect our theory to practice. We add empirical analyses on real-world datasets and deeper architectures.
>
> We hope these clarifications address the reviewer’s questions and concerns, and we would be happy to provide further detail if helpful.
>
> [1] Zweig et al., On Single-Index Models beyond Gaussian Data,*NeurIPS2023*.

---

> > ### Author Rebuttal · Reviewer_QL9u · 2026-04-02
> >
> > Thank you for the detailed rebuttal and for providing additional clarifications.
> >
> > Regarding my concern on Assumption 3.8 (ReLU surrogate), the authors' explanation about the inherent ill-posedness of second derivatives in gradient-based DD and the supplementary analysis in Appendix C for the ReLU case are satisfactory.
> >
> > Regarding Assumption 3.1 (i.i.d. distribution), I acknowledge that this is a standard setup in feature learning theory to isolate the core mechanisms, and the authors have clarified that the theory's goal is to explain the efficiency and mechanism rather than to perfectly mirror real-world data complexity.
> >
> > While the scalability issue (Q3) remains a practical limitation, the authors' response appropriately frames it as a computation-memory trade-off inherent to current DD methods. The additional experiments on real-world datasets and deeper architectures provided in the rebuttal further strengthen the paper's qualitative claims.
> >
> > Overall, the rebuttal has addressed most of my technical concerns. The paper provides a solid theoretical contribution to understanding dataset distillation beyond the kernel regime. Therefore, I will maintain my original score of 4.

---

> > > ### Author Response · Authors · 2026-04-03
> > >
> > > We thank the reviewer for recognizing the theoretical contribution of this work towards understanding dataset distillation beyond simple kernel regimes, as well as for acknowledging our additional experiments in more realistic settings.
> > >
> > > Regarding the remaining partially resolved points, such as the scalability issue of dataset distillation, we agree that these are important and interesting directions. We view scalability primarily as an algorithmic and computational challenge. In this sense, we do not view scalability as a limitation of the theory itself, but rather as an important research direction that our theoretical framework helps motivate and formalize.

---

### Official Review · Reviewer_MJes · 2026-03-13

**Soundness:** 3
**Presentation:** 3
**Significance:** 3
**Originality:** 4
**Overall Recommendation:** 4
**Confidence:** 3

**Summary:**

The paper provides a theoretical analysis of gradient-based dataset distillation techniques, specifically gradient matching and performance matching algorithms. Interestingly, the authors leverage the manifold hypothesis and show that the intrinsic dimension serves as a bound on the size of the distilled set, where the dimension is tailored per training task. Finally, experiments are provided to strengthen the above claims.

**Compliance With Llm Reviewing Policy:**

Affirmed.

**Final Justification:**

The rebuttal addresses my main concerns reasonably well and is consistent with the paper's scope. I remain cautiously positive and keep my score.

**Key Questions For Authors:**

1) Can your analysis be extended to multi-layer neural networks? If not, what would be the obstacles in pursuing this?
2) The theoretical results are compelling, but it remains unclear how these results translate to real-world datasets. For example, do the structural insights e.g., principal subspace encoding, lead to measurable improvements on real-world datasets when using PM and GM? Can this also aid us in understanding when and when not to use PM/GM dataset distillation techniques?
3) The authors in their analysis did not reuse the distilled set, as opposed to what the PM original does. Would resuing change the theoretical guarantees? If so, wouldn't it make the bounds tighter?
4) Can all the assumptions used for the analysis hold when using real-world datasets?

**Limitations:**

The authors adequately discussed the limitations and potential negative societal impact of their work.

**Strengths And Weaknesses:**

**Strengths**
* The authors show that the task of dataset distillation for gradient-based algorithms encodes the intrinsic low-dimensional structure that is tailored for the training task into the distilled set itself, which is very intriguing.
* Novel theoretical contribution that serves to be the first theoretical analysis of gradient-based dataset distillation approaches.
* For small $r$ and $d$, which are the related task intrinsic dimension, the result is favorable upon kernel-based DD theoretical analysis, resulting in smaller distilled sets, when the kernel dimension is high, or the input dimension is high.


**Weaknesses**
* The paper only targets two-layer ReLU neural networks. It is unclear whether the theoretical analysis can be extended to deeper or more realistic applications.
* The paper uses a simplified version of the progressive dataset distillation gradient-based approaches, by not reusing the distilled dataset, which is not reflective of the practical dataset distillation pipelines.
* The analysis relies on many technical assumptions, which may limit applicability.

---

> ### Author Rebuttal · Authors · 2026-03-30
>
> Thank you very much for spending your time on carefully reviewing our paper and for recognizing the originality of the paper.
> ### Q1
> This is indeed an important next step, but not straightforward. In deeper networks, the learned representation evolves jointly across multiple layers. This is one of the main obstacles: it makes it difficult to separate intrinsic dimensions from trajectory-dependent effects and to retain the low-dimensional characterization, while also capturing what is genuinely new in deeper networks. We will clarify this point in the revision. Below, we link our analysis to experiments with deeper networks.
> ### Q2
> We appreciate the reviewer’s question regarding the practical relevance of the theory. We bridge our theory to practice by providing 1. practical implications suggested by the theory and 2. additional experiments on real-world datasets and practical architectures.
>
> 1) Our theory suggests that the success of PM/GM depends less on the ambient input dimension and more on the task-relevant low-dimensional structure. This prediction is consistent with existing empirical observations. For example, [1] estimates the intrinsic dimensions of MNIST, SVHN, and CIFAR10 to be approximately 13, 19, and 26, respectively. In parallel, the dataset distillation (DD) results reported in [2] with 500 distilled images decrease from MNIST to SVHN to CIFAR10 (roughly 98\%, 82\%, and 54\%) for convolutional networks (ConvNet). Our analysis also suggests that the necessary distilled size should scale with task complexity rather than ambient dimension alone. This provides a concrete theory-supported perspective on how the distilled dataset size should scale in practice.
> 2) Our work predicts that increasing the training data size ($N$), network width ($L$), and distilled dataset size should improve DD performance. The latter trend is already well documented empirically (e.g., [2]). We conducted additional experiments on real-world datasets using the setup of [2]: gradient matching with a 3-layer ReLU ConvNet with average pooling and group normalization. We report in tables below the mean over 5 random seeds. Consistent with our theory, increasing $N$ and $L$ leads to improved DD performance.
> |Dataset|$N=10^2$|$N=10^3$|$N=10^4$|
> |---|---:|---:|---:|
> |MNIST|0.9178|0.9542|0.9605|
> |SVHN |0.7016|0.7519|0.7540|
> |CIFAR10|0.2709|0.3971|0.4363|
>
> |Dataset|$L=32$|$L=64$|$L=128$|
> |---|---:|---:|---:|
> |MNIST|0.9480|0.9558|0.9591|
> |SVHN| 0.7062 |0.7390|0.7536|
> |CIFAR10|0.3897|0.4300|0.4406|
>
> We will revise the paper to include these points. Please also see our response to **Reviewer Umuq’s Question 2** for further evidence that low-dimensional structure is encoded into the distilled data in more realistic settings.
>
> To conclude this response, we would like to emphasize that the goal of this work is to provide a principled analysis of gradient-based DD, which several reviewers identified as a core strength of the paper. Accordingly, the contribution is not to directly propose an improved practical algorithm, but to identify and rigorously explain, in a nonlinear and nontrivial setting, a core mechanism underlying *practical* gradient-based DD that had previously only been observed heuristically. We view this as a foundational step toward understanding DD beyond kernel and heuristic regimes, and as an analyzable theoretical framework which can initiate future work both for deeper understanding and for practical improvements.
> ### Q3 and Weakness 2
> We would like to clarify an important distinction. In progressive DD, during retraining, both $\mathcal{D}_1^S$ and $\mathcal{D}_2^S$ would be jointly replayed. When we wrote that we simplified this procedure, we specifically meant that we do not use such joint replay, not that the first-phase distilled set $\mathcal{D}_1^S$ is discarded when constructing the second-phase set $\mathcal{D}_2^S$. $\mathcal{D}_1^S$ is explicitly *reused* in the initialization of $\mathcal{D}_2^S$, as described in Appendix B.8, and this reuse is precisely what helps reduce the memory complexity of the second phase. In this sense, the essential progressive structure is retained. Now, we do not expect this explicit replay of $\mathcal{D}_1^S$ to alter our main conclusion. However, whether it would lead to tighter guarantees is subtle: while replay can help optimization in practice, it also introduces additional dependencies that may complicate worst-case analysis rather than automatically improving the bounds. We will revise the paper to make this explicit.
>
> ### Q4
> Please refer to our response to **Reviewer QL9u’s Question 2**. This discussion also helps clarify our response to Q1.
>
> We hope these clarifications address the reviewer’s questions and concerns, and we would be happy to provide further detail if helpful.
>
> [1] Pope et al., The intrinsic dimension of images and its impact on learning, *ICLR2021*.
>
> [2] Zhao et al., Dataset condensation with differentiable siamese augmentation, *ICML2021*.

---

> > ### Author Rebuttal · Reviewer_MJes · 2026-04-02
> >
> > The rebuttal addresses my main concerns reasonably well and is consistent with the paper's scope. I remain cautiously positive and keep my score.

---

> > > ### Author Response · Authors · 2026-04-03
> > >
> > > We thank the reviewer for recognizing the intended scope of our paper and for acknowledging that the rebuttal was able to address the main concerns reasonably well.
> > >
> > > Concerning extensions to deeper neural networks and more realistic applications, we agree that these are important and promising directions. Although a full treatment is beyond the scope of the present work as noted in our rebuttal, we would like to share that we have taken initial steps to connect our framework to deeper architectures, and we are happy to concisely report this here as it further supports the generality of our theory.
> > >
> > > In particular, our analysis can be interpreted in the context of  deeper neural networks by viewing the model as consisting of a feature-learning layer followed by an output function comprising the remaining layers. Under this perspective, the first distilled dataset of Section 4.1 can be used to retrain the feature-learning layer, while the analysis in Section 4.2 extends to the subsequent layers. This suggests that similar distilled datasets capturing the low-dimensionality of the task exist that can be reused across different deeper architectures, expanding our theory to deeper networks and, at the same time, providing a theoretical explanation for cross-architecture transfer as well.
> > >
> > > We will incorporate this clarification into the paper. We do not consider this a full treatment of deeper networks, as it remains unclear what fundamentally new theoretical insights would arise beyond the core mechanism already identified in our paper when moving to deeper architectures, which would go beyond the scope of the work. Indeed, the crux of the above discussion is to decompose deep neural networks into two components analogously to two-layer neural networks. Nevertheless, this observation supports our main claim that one of the key mechanisms underlying dataset distillation is the preservation of task-relevant low-dimensional structure, which is not inherently restricted to the specific setting studied in this paper.

---

### Decision · Program_Chairs · 2026-04-30

**Decision:**

Accept (regular)

**Comment:**

This paper provides a theoretical exploration of Dataset Distillation (DD), a technique used to compress large datasets into a small set of synthetic "distilled" points. While DD is widely used to reduce the costs of data storage and model optimization, its underlying mechanisms have remained largely empirical.

Reviewers commended the paper for providing much-needed theoretical rigor to a field that has been dominated by heuristic and empirical results.
The analysis of how latent representations are captured during the distillation process was recognized as a valuable contribution to understanding neural network dynamics. By establishing bounds on memory complexity, the work offers a principled way to evaluate the efficiency of different distillation algorithms.

Reviewers initially pointed out that the paper does not provide a "unified theory" for all modern DD algorithms, particularly those using distribution matching, trajectory matching, or diffusion-based distillation.
The authors clarified that these methods optimize fundamentally different objectives (feature alignment vs. training dynamics). They argued that providing a single unified theory for such diverse objectives is currently unfeasible, and their focus on gradient-based learning in two-layer networks serves as a critical foundational step.

Some concerns were raised regarding the breadth of the empirical evaluation. The authors provided a thoughtful response that successfully addressed the concerns within the intended theoretical scope of the paper. This led to a significant increase in confidence from the reviewers, with one increasing their significance score.

The consensus among reviewers is positive. The paper's contribution to the theoretical understanding of dataset distillation is considered substantial and well-timed. While it does not cover every possible distillation variant, the depth of the analysis for multi-index models and two-layer networks provides a strong framework for future research. The authors’ rebuttal was thorough and clarified the limitations of the current theoretical scope, leading to a recommendation for acceptance.